# Learning Actionable Counterfactual Explanations in Large State Spaces

**Keziah Naggita**                                                      *knaggita@ttic.edu*
*Toyota Technological Institute at Chicago*

**Matthew R. Walter**                                                   *mwalter@ttic.edu*
*Toyota Technological Institute at Chicago*

**Avrim Blum**                                                          *avrim@ttic.edu*
*Toyota Technological Institute at Chicago*

**Reviewed on OpenReview:** *https://openreview.net/forum?id=tXnVRpRlR8&noteId=am8pGPvsEE*

## Abstract

Recourse generators provide actionable insights, often through feature-based counterfactual explanations (CFEs), to help negatively classified individuals understand how to adjust their input features to achieve a positive classification. These feature-based CFEs, which we refer to as *low-level* CFEs, are overly specific (e.g., coding experience: $4 \rightarrow 5+$ years) and often recommended in a feature space that doesn't straightforwardly align with real-world actions. To bridge this gap, we introduce three novel recourse types grounded in real-world actions: high-level continuous (*hl-continuous*), high-level discrete (*hl-discrete*), and high-level ID (*hl-id*) CFEs.

We formulate single-agent CFE generation methods for hl-discrete and hl-continuous CFEs. For the hl-discrete CFE, we cast the task as a weighted set cover problem that selects the least cost set of hl-discrete actions that satisfy the eligibility of features, and model the hl-continuous CFE as a solution to an integer linear program that identifies the least cost set of hl-continuous actions capable of favorably altering the prediction of a linear classifier. Since these methods require costly optimization per agent, we propose data-driven CFE generation approaches that, given instances of agents and their optimal CFEs, learn a CFE generator that quickly provides optimal CFEs for new agents. This approach, also viewed as one of learning an optimal policy in a family of large but deterministic MDPs, considers several problem formulations, including formulations in which the actions and their effects are unknown, and therefore addresses informational and computational challenges.

We conduct extensive empirical evaluations using publicly available healthcare datasets (BRFSS, Foods, and NHANES) and fully-synthetic data. For negatively classified agents identified by linear and threshold-based binary classifiers, we compare the proposed forms of recourse to low-level CFEs, which suggest how the agent can transition from state $\mathbf{x}$ to a new state $\mathbf{x}'$ where the model prediction is desirable. We also extensively evaluate the effectiveness of our neural network-based, data-driven CFE generation approaches. Empirical results show that the proposed data-driven CFE generators are accurate and resource-efficient, and the proposed forms of recourse offer various advantages over the low-level CFEs.

## 1 Introduction

Machine learning models are increasingly being used to guide high-stakes decision-making processes. Given the potential impact on individuals' livelihoods, society demands transparency and the right to an explanation, as outlined in Articles 13–15 of the European Parliament and Council of the EU (2016) General

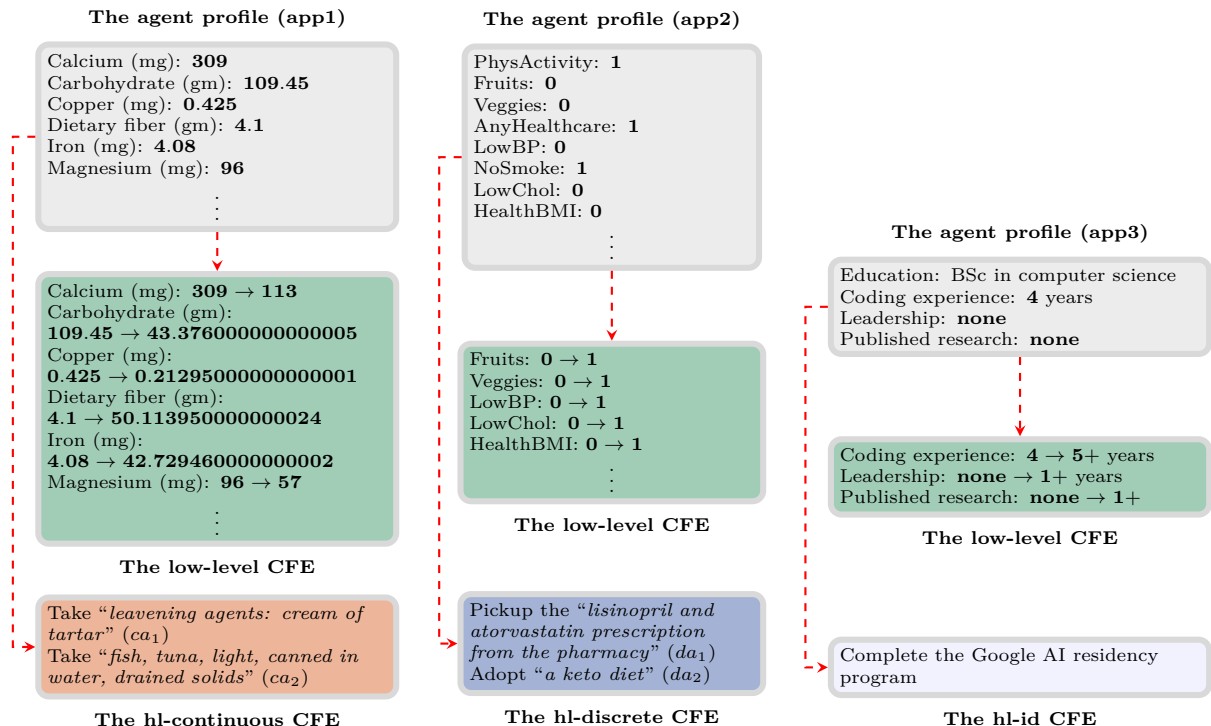

Figure 1: A comparative analysis of low-level CFEs with three high-level types: hl-continuous, hl-discrete, and hl-id, offering actionable insights to help negatively classified agents (with initial profiles/states in grey) achieve positive outcomes from binary classifiers: dietary changes to achieve a healthy waist-to-hip ratio (app1), guide an agent (app2) to meet wellness check criteria, and help an agent (app3) qualify for an AI junior research engineer role. In all cases, the low-level CFE precisely specifies which features to modify and by how much. The hl-continuous CFE adjusts multiple features numerically, e.g., corresponding to features in the profile (app3), $ca_1 = [8.0, 61.5, 0.195, 0.2, 3.72, 2.0, \cdots], ca_2 = [17.0, 0.0, 0.05, 0.0, 1.63, 23.0, \cdots]$ but the agent does not need to understand the exact changes to follow the CFE. The hl-discrete CFE ensures features meet specific eligibility thresholds without the agent needing to know the exact adjustments to take the CFE, e.g., $da_1 = [0, 0, 0, 0, 1, 0, 1, 0, \cdots], da_2 = [0, 1, 1, 0, 0, 0, 0, 1, \cdots]$. The hl-id CFE provides a single overarching high-level action that favorably modifies all the features it should. This figure illustrates how different types of CFEs might impact the agent's ability to interpret and act on given recourse.

Data Protection Regulation and Article 13 of the European Union (2025) AI Act. A critical aspect of this transparency is understanding how individuals (agents) can modify their input features to achieve a desired outcome, such as a positive label in a binary classification setting. Recourse or counterfactual explanation (CFE) generators that provide actionable insights offer one such solution (Wachter et al., 2018; Ustun et al., 2019; Joshi et al., 2019; Dandl et al., 2020; Mothilal et al., 2020; Karimi et al., 2021; 2022).[1]

A popular form of recourse generation, actionable recourse (Ustun et al., 2019), generates feature-based CFEs, specifying precise adjustments to features (state) to ensure that the new features collectively result in a positive classification. For comparison with our work, we refer to these as *low-level* CFEs. While helpful, low-level CFEs are overly specific and might be challenging to translate into real-world-like actions. To address this limitation, we introduce three novel forms of recourse that align with real-world actions: high-level continuous (*hl-continuous*), high-level discrete (*hl-discrete*), and high-level ID (*hl-id*) CFEs.

Figure 1 presents an illustrative example of hypothetical recourse across three binary classification tasks, highlighting the distinctions between low- and high-level CFEs. Specifically, the hl-continuous CFEs involve

---

[1]Following prior work (Pawelczyk et al., 2022; Rasouli & Chieh Yu, 2024; Jiang et al., 2024; Verma et al., 2024), we use the terms recourse and CFE interchangeably. For a detailed discussion and a nuanced differentiation of these terms, we refer the reader to Section 2 of Karimi et al. (2022) and Section 3.3 of Verma et al. (2024).

general and predefined real-world-like actions that modify multiple features simultaneously through numerical adjustments. Similarly, hl-discrete CFEs also stem from real-world actions, each of which might affect several features, with the effect on each feature assumed to be known. However, unlike hl-continuous CFEs, which adjust feature values, hl-discrete CFEs modify feature eligibility. The hl-discrete CFEs use binary vector actions to ensure all features meet a predefined threshold, reducing decisions to unit simple yes/no questions. It is particularly efficient in scenarios where feature satisfiability depends on a set threshold, such as level-one decision-making in wellness checks (see Figure 1 and Appendix A). Lastly, the hl-id CFE encapsulates either hl-continuous or hl-discrete CFEs into a single overarching action, encompassing all necessary changes without detailing specific feature modifications. This level of abstraction relies on domain knowledge and often conveys significant implicit information.

**Our contributions.** First, we introduce three forms of recourse: hl-continuous, hl-discrete, and hl-id CFEs, that bridge the gap between feature-based and real-world actions (Sections 1 and 2). Second, we propose single-agent CFE generation methods that leverage predefined, real-world-like actions to generate optimal CFEs. Specifically, we formulate single-agent hl-discrete CFE generation as a weighted set cover problem and single-agent hl-continuous CFE generation as an integer linear programming (ILP) problem (Section 3).

Third, we propose data-driven approaches that, given instances of agents and their optimal CFEs (agent–CFE dataset), learn a CFE generator that will quickly provide optimal hl-continuous, hl-discrete, and hl-id CFEs for new agents. Unlike the expensive single-agent CFE generation approach, the data-driven methods are more computationally friendly. Additionally, these methods are especially favorable when historical instances of agents and their optimal CFE data are available or aggregatable, recourse generators can operate independently of decision-makers, query access to the classifier is restricted or unavailable, and the actions along with their costs and explicit effects on features are unknown (Section 4). To the best of our knowledge, these alternative forms of recourse and the data-driven approach for generating CFEs are novel contributions.

Finally, we conduct extensive experiments on 30 agent–CFE datasets derived from real-world healthcare datasets: BRFSS, Foods, and NHANES. We chose these datasets due to the availability of a wealth of publicly accessible data, which allows us to effectively demonstrate the benefits of incorporating real-world-like actions in CFE generation and sufficiently explore the impact of a larger action space (or action grid according to Ustun et al. (2019)), enabled by the high number of actionable features with broad value ranges. Alongside these real-world datasets, we also include 34 fully-synthetic agent–CFE datasets for comparison. We extensively compare the low-level CFEs with the hl-continuous and hl-discrete CFEs, and provide an in-depth analysis of the performance of the data-driven hl-continuous, hl-discrete, and hl-id CFE generators under various settings (Sections 5 and 6). Our code can be accessed here.

## 2 Background

We consider a binary classification setting, where an agent in the state $\mathbf{x} \in \mathcal{X}$ receives either a positive (desirable) or negative (undesirable) outcome under a model $f(\mathbf{x})$. The state space $\mathcal{X}$ consists of all valid agent states, each represented by a feature vector capturing attributes such as age and calcium(mg). The model operates over this space, and the CFE generator searches within it to identify alternative states with favorable model outcomes. Although we focus on binary classification, our data-driven CFE framework generalizes to other settings. Given an undesirable outcome, the CFE generator suggests actionable changes to move the agent at state $\mathbf{x}$ to a new state $\mathbf{x}' \in \mathcal{X}$ where the model prediction is desirable. Actionable recourse (Ustun et al., 2019) provides low-level CFEs that specify feature-level changes needed to reach such a state.

**The low-level CFE generator.** Ustun et al. (2019) proposed an ILP-based low-level CFE generator (Equation 1) that generates a low-level CFE to help an agent change an undesirable model outcome to a desirable one.

$$
\begin{aligned}
\min \quad & \text{cost}(\mathbf{a}; \mathbf{x}) \\
\text{s.t.} \quad & f(\mathbf{x} + \mathbf{a}) = \hat{y}^\star \\
& \mathbf{a} \in A(\mathbf{x}),
\end{aligned}
\tag{1}
$$

where $\hat{y}^{\star}$ is the desired model outcome, $A(\mathbf{x})$ denotes the set of feasible actions given the input $\mathbf{x}$, and the function $\text{cost}(\cdot; \mathbf{x}) : A(\mathbf{x}) \to \mathbb{R}_+$ encodes the preferences between these actions. When Equation 1 is feasible, the optimal actions that modify the features (i.e., $\mathbf{x} + \mathbf{a}$) and lead to a desirable model outcome are recommended to the agent (cf. Figure 1). We refer the reader to Ustun et al. (2019) for a more detailed description and to Appendix B.2.1 for dataset-specific experimental setup and supplementary examples of low-level CFEs.

**Shortcomings of low-level CFEs and their generators.** We note two limitations of low-level CFEs in comparison to our proposed forms of recourse (hl-continuous, hl-discrete, and hl-id CFE), and two, as we compare the low-level CFE generator to the proposed data-driven CFE generation approach (Barocas et al., 2020; Karimi et al., 2022; Verma et al., 2024).

First, low-level CFEs are feature-based and highly specific (e.g., in Figure 1, app1, calcium (mg): $309 \to 113$), which may overwhelm agents and introduce additional costs to translate the CFE into implementable steps. In contrast, our proposed CFEs are better aligned with real-world scenarios, offering *what you see is what you get* actionable insights (see Figure 1). Furthermore, with low-level CFEs, details about the actions the agent implements (the number of them needed, which features they would simultaneously modify, and costs to incur) are often unknown beforehand, potentially leading to a misleading price of recourse and related metrics such as sparsity (few modified features) and proximity (closeness of final state to initial state) (Barocas et al., 2020).

Second, although a CFE (e.g., complete the Google AI residency program in Figure 1) could have been optimal for several agents with different but close profiles (e.g., one has coding experience: 4 and another 3), the low-level CFE being too specific, would give the agents different CFEs (coding experience: $4 \to 5+$ years and coding experience: $3 \to 5+$ years). In contrast, our proposed recourse generation approaches are both agent-specific (tailored to an agent's initial state) and generalizable (providing similar recommendations to agents with comparable profiles).

Lastly, while most low-level CFE generators operate on a single-agent basis (Karimi et al., 2022; Verma et al., 2024), recent work by Pedapati et al. (2020); Rawal & Lakkaraju (2020); Kanamori et al. (2022); Ley et al. (2023) and Carrizosa et al. (2024) propose global low-level CFE generators capable of producing CFEs for multiple agents. However, these methods generate feature-based CFEs and require, at a minimum, query access to the classifier. In contrast, we propose data-driven CFE generation approaches that, given historical mappings between agents and their optimal CFEs, such as healthcare intervention records or high school counselors' past successful college recommendations, can learn to generate CFEs with real-world-like actions for multiple new agents without requiring re-optimization. Additionally, the proposed data-driven approaches work well in settings where access to critical information, such as sufficient classification training data, classifier, or a comprehensive list of actions and their costs, is restricted or inaccessible.

## 3 The Proposed Single-agent CFE Generators

This section outlines the single-agent CFE generators for the proposed hl-continuous and hl-discrete CFEs. Each generator relies on predefined, real-world-like actions to solve optimization problems for CFE generation: the weighted set cover problem for hl-discrete CFEs and ILP for hl-continuous CFEs.

### 3.1 The Single-agent hl-continuous CFE Generation

Below, we formally define hl-continuous actions and the single-agent hl-continuous CFE generation process for outcomes predicted by linear classification models.

**Definition 1.** (hl-continuous action): An hl-continuous action is a signed ($\pm$) and predefined real-world action whose cost and varied effects on an agent's input features are predefined and known. For example, in Figure 1, the hl-continuous action: $ca_1$: take "*leavening agents: cream of tartar*" modifies 6+ features by a known amount and the agent incurs a cost (e.g., estimated average price in USD) that is known apriori.

**The singe-agent hl-continuous CFE generator.** This generator produces an hl-continuous CFE by solving an integer linear program (ILP). Given the profile of a negatively classified agent $\mathbf{x}$ and a set of

hl-continuous actions with known costs (defined above), the objective is to identify the lowest-cost subset of hl-continuous actions that, when taken, modify the agent's features to achieve a positive classification. The ILP is of the form:

$$\text{minimize} \quad \sum_{j \in J} \text{cost}_j a_j$$
$$\text{s.t.} \quad \mathbf{c}^T \sum_{j \in J} a_j \cdot (2\epsilon_j - 1) \cdot \mathbf{v}_j \geq -(\mathbf{c}^T \mathbf{x} + b) + \delta \tag{2}$$
$$\epsilon_j \in \{0, 1\}, \quad a_j \in \{0, 1\}, \quad \forall j \in J$$

where $J$ denotes the indices of the hl-continuous actions, with each action represented by a vector $\mathbf{v}_j$ and with a predefined cost, $\text{cost}_j \in \mathbb{R}_+$. The boolean variable $a_j$ indicates the inclusion ($a_j = 1$) or exclusion ($a_j = 0$) of the $j^{\text{th}}$ hl-continuous action, while $\epsilon_j$ encodes the sign of this action, representing addition ($\epsilon_j = 1$) or subtraction ($\epsilon_j = 0$). The coefficients $\mathbf{c}$ and intercept $b$ are the parameters of the linear classifier, and $\delta$ is a small positive value that ensures strict inequality.

## 3.2 The Single-agent hl-discrete CFE Generation

Below, we formally define hl-discrete actions and the single-agent hl-discrete CFE generation process based on a threshold classifier that determines feature eligibility.

**Definition 2.** (hl-discrete action): An hl-discrete action represents a binary vector that adds capabilities to specific features to meet the eligibility threshold. For example, consider the agent state $\mathbf{x} = [0, 0, 0, 0, 1]$ and the hl-discrete action $\mathbf{v}_j = [1, 1, 0, 0, 0]$. When taken, the hl-discrete action adds capabilities to features 1 and 2 of $\mathbf{x}$, transforming it to a new state $[1, 1, 0, 0, 1]$. Although we focus on binary actions, the formulation is extensible to more general cases.

**The singe-agent hl-discrete CFE generator.** This generator produces an hl-discrete CFE by solving a weighted set cover problem. Specifically, it identifies the lowest-cost subset of hl-discrete actions, each with a predefined cost, that a negatively classified agent $\mathbf{x} \in \{0, 1\}^n$ (e.g., someone deemed a health risk) can take to achieve a desirable classification (e.g., no longer classified as a health risk). The problem can be formally defined as follows:

$$\text{minimize} \quad \sum_{j \in J} \text{cost}_j a_j$$
$$\text{s.t.} \quad \sum_{j \in J} d_{ji} a_j + x_i \geq t_i, \ \forall i \in [n], \tag{3}$$
$$a_j \in \{0, 1\}, \ d_{ji} \in \{0, 1\},$$

where $J$ are the indices of the hl-discrete actions, each represented by a vector $\mathbf{v}_j$ and with a predefined cost: $\text{cost}_j \in \mathbb{R}_+$. The threshold classifier $\mathbf{t} = \{t_1, t_2, \cdots, t_n\}$ over $n$ features classifies an agent state $\mathbf{x}$ positive if $x_i \geq t_i, \ \forall i \in [n]$, and negative otherwise. The binary variable $a_j$ denotes inclusion ($a_j = 1$) or exclusion ($a_j = 0$) of the $j^{\text{th}}$ hl-discrete action, while $d_{ji}$ indicates whether the $j^{\text{th}}$ hl-discrete action transforms (adds capabilities to) the feature $i$ of the agent state $\mathbf{x}$, i.e., when performed, the new agent state $\mathbf{x} + \mathbf{v}_j = \mathbf{x}'$ is such that $x'_i > x_i$ and $x'_i \geq t_i$.

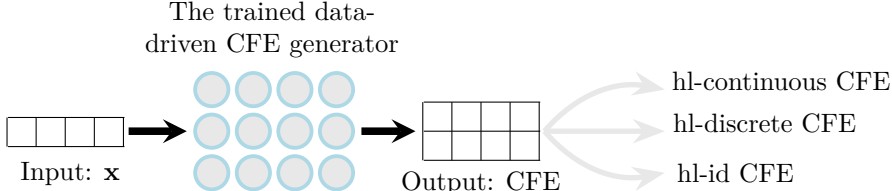

Figure 2: Given an agent state (profile) $\mathbf{x}$, a data-driven CFE generator trained on instances of agents and their optimal CFEs (agent–CFE dataset) generates a high-level CFE (hl-continuous, hl-discrete or hl-id) for the agent without the need for generator re-optimization or access to the decision-making classifier.

# 4    The Proposed Data-driven CFE Generators

This section, supplemented by Appendix C, details the three proposed data-driven CFE generators:  hl-continuous, hl-discrete, and hl-id (see Figure 2).  Each generator learns from instances of agents and respective optimal CFEs, defined as the least cost CFE that leads to a favorable model outcome.  Once trained, these generators can produce optimal CFEs for new agents without re-optimization.  Empirical results demonstrate that even shallow deep-learning architectures perform strongly at this task, that is, generate *correct* CFEs, the least cost CFEs that favorably flip the model outcome.

The data-driven approaches are computationally more efficient than single-agent CFE generation approaches, which require optimization for each new agent.  Furthermore, they are particularly favorable when agent–CFE data is available or aggregatable from various sources.  They allow recourse generation to operate independently of decision-makers, function without direct access to the classifier, and handle scenarios where action costs and their explicit effects on features are unknown.

## 4.1    The Data-driven hl-continuous CFE Generator

We develop a data-driven hl-continuous CFE generator trained on agent–hl-continuous CFE training dataset, consisting of agents and their corresponding optimal hl-continuous CFEs.  Each CFE defines a set of hl-continuous actions along with their associated costs.  For instance, a CFE might include actions {action-a, action-b, action-c} with corresponding costs {cost-a, cost-b, cost-c}.  The generator learns from this data to produce hl-continuous CFEs for new agents without requiring generator re-optimization.  Below are further details of the generator architecture.

Given the agent–hl-continuous CFE training dataset, we train a neural network model to learn to generate hl-continuous CFEs for testing set agents.  Specifically, the generator is a neural network model with three hidden layers, each containing $2,000$ neurons.  The model incorporates $\ell_2$ regularization, dropout, and batch normalization.  The training process uses the Adam optimizer (Kingma & Ba, 2015) with early stopping, restoring the best weights after a patience level of $300$.  The model trains with a batch size of $6,000$ for an average of $5,000$ epochs.  To ensure accurate data-driven hl-continuous CFE generation for both training and testing set agents, we optimize the model loss function $\mathcal{L}_{\text{HC}}$ given by:

$$\mathcal{L}_{\text{HC}} = -\frac{1}{M} \sum_{m=1}^{M} \sum_{j=1}^{J} [a_{jm} \log(\hat{a}_{jm}) + (1 - a_{jm}) \log(1 - \hat{a}_{jm})] \tag{4}$$

where $\hat{a}_{jm}$ is the predicted probability and $a_{jm}$ is the true indication of a presence (1) or absence (0) of the $j^{\text{th}}$ hl-continuous action in agent $m$'s hl-continuous CFE.  There are $J$ possible hl-continuous actions and $M$ agents in the agent–hl-continuous CFE training dataset.

## 4.2    The Data-driven hl-discrete CFE Generator

We propose a data-driven hl-discrete CFE generator, trained using the agent–CFE training dataset and evaluated on the agent–CFE testing dataset.  Each agent–CFE dataset comprises instances of agents and their optimal hl-discrete CFEs that specify a set of hl-discrete actions and associated costs.

Given the agent–hl-discrete CFE training dataset, we design a sequential encoder-decoder model to generate hl-discrete CFEs for new agents without generator re-optimization.  The model configuration was dependent on the experimental setting.  On average, we used 500 training epochs with a batch size of 128, a dropout rate of 0.4, a learning rate of 0.0005, and either the mean squared error or binary cross-entropy loss as the objective function.  The models, on average, consisted of three layers, each using ReLU activation functions.

## 4.3    The Data-driven hl-id CFE Generator

The data-driven hl-id CFE generator is a supervised learning model trained on agent–hl-id CFE dataset of agent–CFE pairs consisting of agents' initial states (profiles) and their corresponding optimal hl-id CFEs and associated costs.  Each hl-id CFE, as introduced in Section 1 and illustrated in Figure 1, is a single high-level

action encapsulating all required changes without specifying individual feature modifications. When trained, the generator learns to generate CFEs for new agents without requiring re-optimization. Below, we provide details of the generator architecture.

Given the agent–hl-id CFEs training dataset, we design the data-driven hl-id CFE generator as a neural network model with an average of two hidden layers, each consisting of 2000 neurons, $\ell_2$ regularization, dropout, and batch normalization. We used the Adam optimizer (Kingma & Ba, 2015) and implemented early stopping and restoration of the best weights after a patience level of 360. On average, we set the batch size to 2000 and the number of epochs set to 3000. To ensure that the data-driven hl-id CFE generator performs well on the training dataset and accurately generates hl-id CFEs for agents in the testing set, we optimize the model loss function $\mathcal{L}_{\text{HiD}}$ given by:

$$\mathcal{L}_{\text{HiD}} = -\frac{1}{M} \sum_{m=1}^{M} \sum_{k=1}^{K} [a_{km} \log(\hat{a}_{km})] \tag{5}$$

where $\hat{a}_{km}$ is the predicted probability and $a_{km}$ is the true indication of the $k^{\text{th}}$ CFE being the hl-id CFE (1) or not (0) for the $m^{\text{th}}$ agent. There are $K$ possible hl-id CFEs and $M$ agents in the training dataset.

## 5 Experimental Setup

This section provides a detailed description of the experimental setup, including the evaluation metrics employed and the methodology for generating the 34 fully-synthetic agent–CFE datasets and the 30 semi-synthetic agent–CFE datasets from real-world healthcare data sources: BRFSS, Foods, and NHANES.

### 5.1 Real-world Datasets

Below, we outline the extraction and preprocessing of the real-world datasets used in our experiments, including their statistical descriptions. We split all datasets into an 80/20 ratio for training and testing.

**The Foods, BMI, and WHR datasets.** We extracted the Foods dataset from USDA, Agricultural Research Service, Nutrient Data Laboratory (2016); Awram (2024) and the BMI (body mass index) and WHR (waist-to-hip ratio) datasets from NHANES body measurement surveys (CDC, 1999; ICPSR at the University of Michigan, 2024), covering the years 1999 to pre-pandemic 2020.

To ensure commonality in actionability features between the (Foods, BMI) and the (Foods, WHR) dataset pairs, we selected intersectional nutritional intake features: *protein (gm), carbohydrate (gm), dietary fiber (gm), calcium (mg), iron (mg), magnesium (mg), phosphorus (mg), potassium (mg), sodium (mg), zinc (mg), copper (mg), selenium (mcg), vitamin C (mg), niacin (mg), vitamin B6 (mg), total folate (mcg), vitamin B12 (mcg), total saturated fatty acids (gm), total monounsaturated fatty acids (gm)*, and *total polyunsaturated fatty acids (gm)*.

After preprocessing, for example, removing missing data and ensuring that selected nutritional intake features were a subset of the intersectional ones, the Foods dataset contained 3901 food items. Each item includes nutritional composition. We added two cost attributes: Monetary cost (in USD, obtained via web scraping) and Caloric cost (reflecting total caloric content, sourced from (Caputo, 2023)). In our experiments, Foods+costs serves as the **hl-continuous action space**, where food items represent actions, and the cost attributes define the cost constraints. Based on cost type, we define two forms of hl-continuous actions: (i) Foods+monetary cost and (ii) Foods+caloric cost.

The BMI dataset after preprocessing contained 50918 agents, each with 3 demographic features and 19 nutrient intake features, classified as either healthy (1) or unhealthy (0) BMI. On the other hand, the WHR dataset contained 9120 agents, each with 3 demographic features and 20 nutrient intake features, classified as either healthy (1) or unhealthy (0) WHR. For additional preprocessing details, see Appendix B.1.1.

**The BRFSS dataset.** We extracted the Behavioral Risk Factor Surveillance System (BRFSS) dataset from Teboul (2024); Centers for Disease Control and Prevention (2024). After preprocessing, e.g., removing missing data, the dataset was reduced to $13,799$ agents, each represented by 16 binary health risk features.

These include: *LowBP, LowChol, HealthBMI, NoSmoke, NoStroke, NoCHD, PhysActivity, Fruits, Veggies, LightAlcoholConsump, AnyHealthcare, DocbcCost, GoodGenHlth, GoodMentHlth, GoodPhysHlth* and *NoDiffWalk*. For additional details, see Appendix B.1.2.

## 5.2 Single-agent CFE Generation

Here and in Appendices B.2.1, B.2.2, and B.2.3, we describe the generation of low-level, hl-continuous, and hl-discrete CFEs using single-agent CFE generators (i.e., Equations 1, 2, 3). Since the agents and their computed CFEs will also be used to train and evaluate data-driven CFE generators, we generate CFEs for negatively classified agents in the BMI, WHR, and BRFSS training and testing datasets.

For BMI and WHR datasets, only intersectional nutritional features are considered actionable, whereas all features are actionable for the BRFSS dataset. We trained binary classifiers to identify agents requiring CFEs and identify classifier parameters to use in single-agent CFE generators. Fine-tuned logistic regression models for BMI and WHR achieved test accuracies of 72.78% and 85.18%, respectively. For the BRFSS dataset, which focuses on wellness checks, a threshold classifier $\mathbf{t} = \mathbf{1}_{16}$ achieved 100% accuracy.

**The single-agent low-level CFE generation.** We generated a low-level CFE for each negatively classified agent in the BMI, WHR, and BRFSS training/testing datasets. We accomplished this by using the agent's initial state and the parameters of the trained binary linear decision-making classifiers for each dataset, along with the ILP framework defined in Equation 1.

**The single-agent hl-continuous CFE generation.** For the BMI and WHR training/testing datasets, we use the negatively classified agents alongside two types of hl-continuous actions: Foods+monetary costs and Foods+caloric costs to create hl-continuous CFEs. Using the ILP framework defined in Equation 2, we generate two distinct forms of hl-continuous CFEs for each agent: the optimal set of food items with minimal monetary cost and the optimal set of food items with minimal caloric cost.

**The single-agent hl-discrete CFE generation.** Lastly, using the BRFSS training/testing set agents, the threshold classifier ($\mathbf{t} = \mathbf{1}_{16}$), and 100 synthetically generated hl-discrete actions (each of length 16) with associated costs, we applied Equation 3 to generate a hl-discrete CFE for each agent. Each CFE represents an optimal set of hl-discrete actions with minimal costs for each agent.

## 5.3 Data-driven CFE Generation

This section, along with Appendices B.2, B.3, and C, describe the creation of agent–CFE datasets (where the CFE is either hl-continuous, hl-discrete, or hl-id), and their role in data-driven CFE generation.

**The semi-synthetic agent–CFE datasets.** Using agent states and their optimal CFEs from Section 5.2, we construct training and testing agent–CFE datasets. First, we generate: 2 agent–hl-continuous CFE train/test datasets for BMI, 2 agent–hl-continuous CFE train/test datasets for WHR, and 1 agent–hl-discrete CFE train/test datasets for BRFSS.

Then, given the following agent–CFE datasets: the agent–hl-continuous CFE train/test datasets for BMI, created using Foods+monetary costs as hl-continuous actions; the agent–hl-continuous CFE train/test datasets for WHR, created using Foods+caloric costs as hl-continuous actions; and the agent–hl-discrete CFE train/test datasets for BRFSS, we generate 3 agent–hl-id CFE datasets. Specifically, for each agent–CFE dataset, we create a unique identifier for the CFE that denotes the single overarching action, resulting in an agent–hl-id dataset corresponding to instances of agents and their hl-id CFE.

Lastly, for each of the agent–CFE datasets described above, we generated three variations based on the frequency of CFEs in the dataset: `all` (includes all data), `>10` (CFEs with more than 10 agents), and `>40` (CFEs with more than 40 agents) varied frequency of CFEs agent–CFE datasets.

**The fully-synthetic agent–CFE datasets.** We use the ILP defined in Equation 3 to generate five variants of the agent–hl-discrete CFE datasets: varied dimensionality, frequency of CFEs, information access, feature satisfiability, and actions access. Below, we briefly describe the varied dimensionality and frequency of CFEs datasets and include more details about these and other variants in Appendix B.3.

For varied dimensions agent–CFE datasets, we generated datasets with 20, 50, and 100 dimensions (actionable features), where we set the agent's feature to 1 with a probability $p_f$, and each discrete action can add capabilities to a feature with a probability $p_a$. The cost of each action depends on the features it transforms. Lastly, we created three varied frequency of CFEs datasets: `all`, `>10`, and `>40`, and agent–hl-id CFE datasets for each varied dimensions agent–CFE dataset, using a similar approach as in the semi-synthetic agent–CFE datasets described above.

**Data-driven CFE generators.** Given the semi-synthetic and fully-synthetic training agent–CFE datasets, we train the corresponding data-driven generators described in Section 4 and evaluate their effectiveness on the testing agent–CFE datasets. See Appendix C for supplemental details.

### 5.4 Evaluation and Comparative Analysis Metrics

We compare single-agent generated low-level CFEs to both hl-continuous and hl-discrete CFEs. Additionally, we assess the performance of data-driven CFE generators. The metrics used for comparison and evaluation are detailed below and in Appendix D.

**Accuracy of data-driven generators.** To assess the accuracy of the proposed data-driven CFE generators, we use zero-one loss (see Equation 6), which checks if the generated CFE $\hat{I}$ matches the true CFE $I$, defined as the least cost CFE that favorably flips the model outcome.

$$\mathcal{L}_{\text{eval}}(I, \hat{I}) = \begin{cases} 0 & \text{if } I = \hat{I} \\ 1 & \text{if } I \neq \hat{I} \end{cases} \tag{6}$$

**Comparison metrics.** We analyze various factors related to the use of CFEs, including the average number of actions taken, the number of modified features, the proportion of agents sharing the same optimal CFE, and the overall improvement measured as the distance between an agent's initial state and its final state after following a CFE. Assuming CFEs encourage truthful responses, we refer to this as agent *improvement*. We compare these factors when agents follow a low-level CFE versus a high-level CFE, either hl-continuous or hl-discrete. The comparative analysis focuses on CFEs generated by single-agent CFE generators for negatively classified agents in the training sets of three datasets: BMI, WHR, and BRFSS. To ensure a fair comparison, we include only agent–CFE pairs where both low- and high-level CFEs are available, as the low-level CFE generator (cf. Equation 1) occasionally fails to produce a CFE.

To assess how much each variable, e.g., number of modified features varies across groups, for example, between male and female agents, we compute the coefficient of variations (Equation 7), a normalized measure of dispersion calculated as the ratio of the standard deviation to the mean of the variable $v$.

$$\text{coefficient of variation}(v) = \frac{\text{standard deviation}_v}{\text{mean}_v} \times 100 \tag{7}$$

## 6 Experimental Results

In this section, we provide comprehensive empirical evidence showcasing the strong performance of our data-driven CFE generators and their advantages over single-agent CFE generators. Furthermore, we highlight the advantages of hl-continuous and hl-discrete CFEs, which offer actionable insights that closely align with real-world action spaces, over feature-based low-level CFEs.

### 6.1 Comparison of low-level CFEs to the hl-continuous and hl-discrete CFEs

Below and in Appendices E.1 and E.2, we provide empirical evidence to show that, compared to low-level CFEs, both hl-continuous and hl-discrete CFEs involve fewer actions but lead to more improvement involving more modified features, are easier to personalize, and simplify the design and interrogation of CFE generators for fairness issues.

**Fewer actions but higher improvement and more modified features.** Our results indicate that hl-continuous and hl-discrete CFEs require fewer actions while yielding higher improvements and modi-

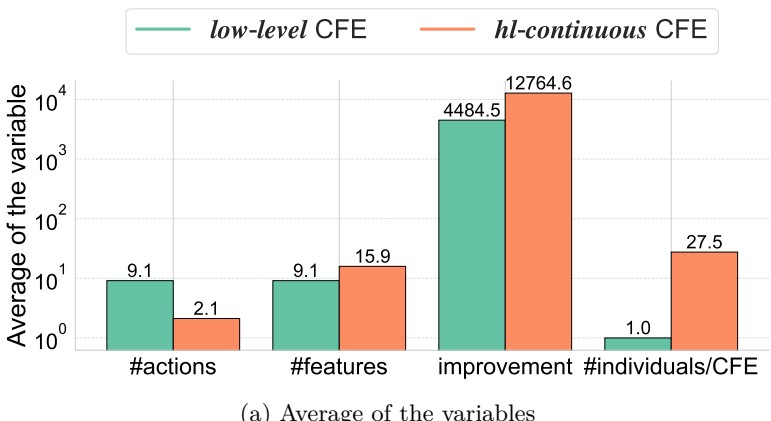

(a) Average of the variables

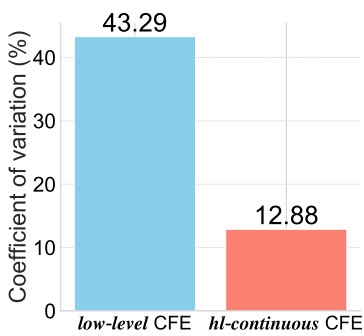

(b) Coefficient of variation in number of modified features across intersectional (race, age, gender) groups

Figure 3: On the WHR dataset, (a) compared to low-level CFEs, to take hl-continuous CFEs require fewer actions but modify a significant number of features and result in higher improvement. On average, each hl-continuous CFE is optimal for multiple agents, with a high frequency of 27.5 per CFE, unlike the low-level CFEs with 1.0. Additionally, as shown in (b), there is greater variability in the number of features modified across sensitive groups when using low-level CFEs, suggesting lower fairness than with hl-continuous CFEs. For more details on (a), see Appendix Figures 9, 10, and 11; for (b), see Appendix Figure 13(c) and 13(d).

fying more features than low-level CFEs. In contrast, low-level CFEs involve more actions but result in lower improvements despite modifying a high number of features. For instance, while on average, on the WHR dataset, the hl-continuous CFEs require only 2 actions yet achieve a significantly higher improvement (12, 765), the low-level CFEs involve 9 actions but yield a much lower improvement (4, 484.5) (see Figure 3(a)).

In low-level CFE generation, sparsity (small number of modified features) and proximity (new agent state after taking the CFE close to the initial state) are often a primary goal due to actionable insights being part of the feature space (Ustun et al., 2019; Verma et al., 2024). We observe a perfect positive correlation between the number of modified features and actions taken (Kendall's $\tau = 1.0$, $p\text{-}value = 0.0$) and a positive correlation between actions taken and improvement achieved in low-level CFEs (Kendall $\tau = 0.368$, $p\text{-}value = 5.41e\text{-}227$). However, for hl-continuous CFEs, the correlation between the number of modified features and actions taken is positive but weaker (Kendall's $\tau = 0.722$, $p\text{-}value = 0.0$) and there is almost no relationship between the number of actions taken and improvement achieved (Kendall $\tau = 0.0625$, $p\text{-}value = 3.21e\text{-}06$). Despite requiring fewer actions (2), hl-continuous CFEs modify significantly more features (16) compared to low-level CFEs, which involve 9 actions and modify 9 features (Figures 1 and 3(a)).

Consequently, unlike taking low-level CFEs, agents using hl-continuous or hl-discrete CFEs tend to become more "positive" or "qualified" after taking fewer actions. That is, our proposed form of CFEs require fewer actions but result in greater improvement, broader feature modifications, and lower costs in both interpretation and execution (see Figures 1 and 3(a), and Appendix E.1 and Figures 9, 10, 11 and 12).

**Personalization and fairness.** Since both hl-discrete actions and hl-continuous actions are predefined and real-world-like, it is easier and more transparent to examine the hl-discrete and hl-continuous CFE generators and generated CFEs for potential fairness issues, and to tailor the CFE generation to agents' needs. For example, our data-driven hl-continuous CFE generators can produce CFEs for agents who place greater importance on monetary costs over caloric costs.

Moreover, agents across the intersectional sensitive (race, age, and gender) groups (e.g., Hispanic, 21-40, Female) take a comparable number of actions, modifying a closely similar number of features, achieving comparable improvements, and incurring closely similar costs when using hl-continuous CFEs, as indicated by the low coefficient of variation in Figure 3(b) and Appendix Figures 13 and 14. In contrast, taking low-level CFEs results in higher variability across the groups in all these variables (see Figure 3(b) and Appendices E.2.2 and E.2.1). Thus, high-level CFEs yield fairer outcomes than low-level CFEs.

| | Accuracy of CFE generators | | | | | Effect of frequency of CFEs | | |
|---|---|---|---|---|---|---|---|---|
| | hl-continuous | hl-discrete | hl-id | | | all | >10 | >40 |
| BMI | $0.92 \pm 0.0053$ | | $0.94 \pm 0.0045$ | | 20-dim | $0.84 \pm 0.0060$ | $0.89 \pm 0.0052$ | $0.94 \pm 0.0042$ |
| WHR | $0.92 \pm 0.0176$ | | $0.97 \pm 0.0107$ | | 20-dim$\star$ | $0.97 \pm 0.0028$ | $0.98 \pm 0.0021$ | $0.99 \pm 0.0014$ |
| BRFSS | | $0.98 \pm 0.0102$ | $0.99 \pm 0.0050$ | | BMI | $0.90 \pm 0.0057$ | $0.91 \pm 0.0055$ | $0.92 \pm 0.0053$ |
| 20-dim | | $0.94 \pm 0.0042$ | $0.99 \pm 0.0014$ | | BRFSS | $0.70 \pm 0.0182$ | $0.86 \pm 0.0158$ | $0.98 \pm 0.0102$ |

Table 1: (**left**) The accuracy of the hl-continuous, hl-discrete, and hl-id data-driven CFE generators on the testing set agents for >40, BMI, BRFSS, WHR, and fully-synthetic (20-dim): 20-dimensional, datasets. (**right**) The data-driven CFE generators' accuracy decreases with a decrease in the frequency of CFEs (number of agents for whom a given CFE is optimal) in the agent–CFE training set, regardless of the dataset type. Specifically, training and testing on the (20-dim): the 20-dimensional agent–hl-discrete CFE dataset, (20-dim)$\star$: the 20-dimensional agent–hl-id dataset, (BMI): the BMI agent–hl-continuous CFE dataset, and (BRFSS): the BRFSS agent–hl-discrete CFE dataset all show this trend. The data-driven CFE generators are accurate (**left**) and their accuracy improves as the frequency of CFEs increases, with those trained highest CFE frequency dataset (>40) performing best (**right**).

Lastly, our data-driven hl-discrete CFE generators effectively generate CFEs for all agents, regardless of action restrictions or feature satisfiability variations. This strong performance holds even without explicit knowledge of these variations (see Appendices E.2.3 and E.2.4).

## 6.2 The Data-driven CFE generators are Accurate and Resource-efficient

Our results show that the proposed data-driven CFE generators are resource-efficient in terms of both limited information access and low computational overhead. These generators, without requiring re-optimization, accurately and efficiently generate CFEs for new agents after being trained under restrictive constraints such as no query access to the classifier or, in the case of the data-driven hl-id CFE generator, without knowledge of the cost and impact of actions on agent states. Refer to Table 1(left) and Appendices E.3, E.4, and E.5).

In contrast to the overly specific low-level CFEs, which are generally unique to each agent, hl-continuous and hl-discrete CFEs are often optimal for a broad range of agents (refer to Figures 1 and 3(a) and Appendix Figure 18). The removal of the need for re-optimization for each new agent, combined with the general applicability of the actions to agents, enhances the scalability of our proposed CFE generators compared to low-level generators. Additionally, because the actions in the hl-continuous and hl-discrete CFEs are both general and predefined, they are more transparent and easier to interpret (see Figure 1), making them cheaper and more desirable than the overly specific and unique low-level CFEs.

However, our results show that the accuracy of the proposed data-driven generators declines with the low frequency of CFEs (see Table 1 (right)) and the generalizability of CFE generation decreases with an increase in the number of actionable features. We observed that this is due to the growing uniqueness of CFEs to agents (see Table 1 (right)) and Section E.6). Data augmentation mitigates the negative effects of low CFE frequency. For instance, on the `all` 20-dimensional dataset, data augmentation improves accuracy from 0.969 to 0.982.

Lastly, the performance of data-driven CFE generators improves as model complexity increases. For example, on a discrete agent–hl-id dataset, the neural network model outperforms the Hamming distance method (see Appendix Figure 20). Future research could explore more advanced data-driven models for CFE generation and techniques like federated learning to enable CFE generation under limited access to agent–CFE data and privacy constraints.

## 7 Limitations and Ethical Considerations

The decision-maker must have access to data on instances of agents and their corresponding optimal CFEs to train the proposed data-driven CFE generators. Although this level of access mitigates some information access challenges, such as needing at least query access to the classifier and representative prediction training

data or having an exhaustive list of actions and the associated costs, obtaining a historical agent–CFE dataset may still pose significant challenges. Future research could investigate techniques like federated learning and secure multi-party computation to facilitate collaborative training of robust CFE generators under varied privacy and data access constraints.

While our proposed data-driven CFE generators are agnostic to the underlying classification model, the proposed single-agent hl-continuous CFE generators rely on linear classifiers. Although the linear models may not always be optimal, they can outperform non-linear models in some contexts (Wainer, 2016). When non-linear classifiers are preferred, one practical alternative is to approximate them by linear models (e.g., (Bshouty & Long, 2012; Li, 2015; Shalizi, 2020; Liu et al., 2021)), enabling CFE generation with the proposed single-agent CFE generator. Extending the generator to produce exact CFEs for non-linear models in a scalable and computationally efficient manner remains a challenging but promising avenue for future research.

Additionally, our formulations of hl-continuous and hl-discrete CFEs restrict them to being defined as a set of actions. More generally, one could consider settings where the order of actions matters, such as where a CFE corresponds to an optimal policy for an agent in a deterministic Markov decision process (MDP). Further, one could consider actions whose effects are stochastic, and a CFE then corresponds to an optimal policy for the agent in a general MDP.

Since the proposed approaches to data-driven CFE generation are closely related to data-driven algorithm design, ethical concerns related to data-driven algorithms, e.g., potentially propagating and exacerbating biases in historical agent–CFE data and the potential for flawed resource allocation, might apply to our proposed CFE generators. Future research should investigate these ethical implications in greater depth.

Although our experiments primarily use healthcare datasets, our data-driven CFE generation approach generalizes to a broad spectrum of real-world scenarios, such as college admissions, loan applications, judicial systems, and other settings. Future works could expand our setup to other data settings and informational access challenges. Lastly, we caution readers that the experimentally generated CFEs from our empirical analyses are intended solely for illustrative purposes, and readers should not use them for self-treatment.

## 8 Related Work

The proposed single-agent hl-continuous and hl-discrete CFE generation approaches are in principle, similar to search-based optimization CFE generation frameworks (Ramakrishnan et al., 2020), single-agent ILP recourse generation approaches (Cui et al., 2015; Gupta et al., 2019; Ustun et al., 2019), and CFE generation methods based on logic and answer-set programming (Bertossi, 2020; Liu & Lorini, 2023; Marques-Silva, 2024). However, unlike these approaches, ours uses predefined real-world-like actions (see Figure 1), resulting in CFEs that involve fewer actions but modify more features and lead to more improvement.

Although most low-level CFE generators operate on a single-agent basis (Karimi et al., 2022; Verma et al., 2024), recent studies (Pedapati et al., 2020; Rawal & Lakkaraju, 2020; Kanamori et al., 2022; Ley et al., 2023; Carrizosa et al., 2024) have introduced approaches that can produce CFEs for multiple agents. Most closely related to our work is the approach by Kanamori et al. (2022), which learns a decision-tree-based global CFE generator that, once learned, can generate CFEs for multiple agents. However, unlike Kanamori et al. (2022), our data-driven CFE generators generate CFEs with real-world-like actions. Moreover, solving the mixed-integer linear programs and the overall Counterfactual Explanation Tree can be computationally prohibitive for scenarios with large action spaces, making our supervised learning approach a more scalable and efficient alternative.

Unlike low-level CFE generators that require, at a minimum, query access to the classifier and knowledge of the cost and impact of each action on state features (Shavit & Moses, 2019; Pedapati et al., 2020; Rawal & Lakkaraju, 2020; Naumann & Ntoutsi, 2021; Verma et al., 2022; Kanamori et al., 2022; De Toni et al., 2023; Ley et al., 2023; Carrizosa et al., 2024), without explicit access to this information, our data-driven CFE generators leverage access to agents and their optimal CFEs to generate CFEs described by real-world-like actions and costs.

While in some ways, the proposed data-driven CFE generators are similar to reinforcement learning-based CFE generation tools (Shavit & Moses, 2019; Naumann & Ntoutsi, 2021; De Toni et al., 2023), our proposed approach offers a more resource-efficient and exact solution alternative to the often high computational and approximate solutions. Notably, our approach is closest to that of Verma et al. (2022). While our method is akin to learning an optimal policy in a large but deterministic family of Markov decision processes (MDPs), Verma et al. (2022) focuses on learning optimal policies within smaller, stochastic MDP settings.

Finally, our work also relates to data-driven algorithm design (Gupta & Roughgarden, 2016; Balcan et al., 2018; Balcan, 2020), where models learn from training data instances to generalize to the testing data. We introduce novel data-driven CFE generators that address the question: *Can we, by learning from training agent–CFE data (i.e., instances of agents and their optimal CFEs), develop a CFE generator that quickly provides optimal CFEs for new agents?* Our proposed approach excels in generating CFEs for new agents, is computationally efficient and scalable, and functions effectively under varied informational access settings.

## 9 Conclusion

In this work, we propose three forms of recourse where actionable insights align closely with real-world actions and investigate settings where CFEs can be generated by analyzing the similarities between negatively classified agents using data-driven approaches. Our findings show that compared to low-level CFEs, both hl-continuous and hl-discrete CFEs require fewer actions, modify more features, and result in higher improvements. Additionally, the CFEs are fairer across sensitive groups and are easier to examine, compare, and personalize than low-level CFEs. Lastly, we empirically show that the proposed data-driven CFE generators are accurate, resource-efficient, and perform effectively under various information access constraints, including limited or restricted access to classifier parameters and training data.

### Acknowledgments

We are deeply grateful to Alexandra DeLucia for their insightful and comprehensive feedback on all drafts of the manuscript. We would also like to thank Harvineet Singh and the anonymous reviewers for their valuable feedback on our work. This work was supported in part by the National Science Foundation under grants CCF-2212968 and ECCS-2216899, and by the Simons Foundation under the Simons Collaboration on the Theory of Algorithmic Fairness.

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

# A  The hl-discrete CFE vs. hl-continuous CFE: Supplementary Details

We further elaborate on the distinction between hl-continuous and hl-discrete CFEs and a potentially interesting future direction.

**Differentiation between hl-discrete and hl-continuous CFEs.**  The key distinction between hl-discrete and hl-continuous CFEs is how they affect features, where the former caters to feature eligibility and the latter numerically modifies feature values. For instance, consider the feature *healthBMI* in App2, Figure 1. While hl-discrete CFE would ensure BMI is past a desired threshold, thus ensuring *healthBMI* = 1, the hl-continuous CFE would affect the actual feature values, such as reducing BMI from 30 to 22.3.

It is important to note that a threshold defining feature eligibility can stem from any classifier type. Continuing with the *healthBMI* example, the binary eligibility ($\{0, 1\}$) is determined based on whether the feature crosses a predefined threshold of what qualifies as a healthy BMI, as dictated by the classification model's set threshold.

These two forms of CFEs are suited for different contexts. The hl-discrete CFEs are particularly effective in rule-based systems, such as eligibility checks, wellness evaluations, or quality and safety assessments. In contrast, hl-continuous CFEs are better suited for scenarios where fine-grained numerical adjustments are meaningful, typically in settings where low-level CFEs are effective.

While it might be easier to go from hl-continuous CFEs to hl-discrete CFEs, the reverse is less trivial and may require additional considerations.

**Integration of high-level actions into existing CFE generation methods.**  While it may be feasible to incorporate the hl-continuous or hl-discrete actions into existing CFE generation methods workflows, it's currently unclear which methods are best suited, how readily they can be adapted, or what challenges might emerge. For example, in evolutionary algorithm-based approaches, generating exact CFEs under this new paradigm could be computationally intensive, particularly in large action spaces, thus potentially only generating approximate or Pareto-optimal CFEs. We hope future research explores how to effectively adapt existing CFE generation methods to this new paradigm and uncover any associated interesting challenges.

# B  Datasets: Supplemental Details

This section describes the supplemental details about the datasets used in the experiments. We conducted all experiments on a laptop with a CPU featuring the following hardware specifications: a 2.6 GHz 6-Core Intel Core i7 processor, 16 GB of 2400 MHz DDR4 RAM, and an Intel UHD Graphics 630 with 1536 MB of video memory. In all cases where we implement Equations 1, 2 and 3, we use the `CVXPY` Python package (Diamond & Boyd, 2016; Agrawal et al., 2018).

## B.1  Real-world Datasets Extraction and Preprocessing

First, we describe the extraction and preprocessing of real-world datasets: Foods, BMI, WHR, and BRFSS. Then, we describe the creation of semi-synthetic agent–hl-continuous CFE, agent–hl-discrete CFE, and agent–hl-id CFE datasets.

### B.1.1  Foods, Body Mass Index (BMI), and Waist-to-Hip Ratio (WHR) Datasets

**Intersectional nutritional features.** After extracting the datasets for Foods, BMI, and WHR and removing features with missing values in the Foods dataset, we selected an intersectional subset of nutritional value features in the Foods and BMI datasets and the Foods and WHR datasets. This subset consisted of 20 features, including: *'protein (gm)', 'carbohydrate (gm)', 'dietary fiber (gm)', 'calcium (mg)', 'iron (mg)', 'magnesium (mg)', 'phosphorus (mg)', 'potassium (mg)', 'sodium (mg)', 'zinc (mg)', 'copper (mg)', 'selenium (mcg)', 'vitamin C (mg)', 'niacin (mg)', 'vitamin B6 (mg)', 'total folate (mcg)', 'vitamin B12 (mcg)', 'total saturated fatty acids (gm)', 'total monounsaturated fatty acids (gm)'*, and *'total polyunsaturated fatty acids (gm)'*.

**Foods dataset preprocessing.** The Foods dataset from Awram (2024) initially contained 53 features. After finding the intersectional subset of nutritional value features and removing datapoints with missing values, the dataset had 27 features. These included the following: *'NDB_No', 'Shrt_Desc', 'GmWt_1', 'GmWt_Desc1', 'GmWt_2', 'GmWt_Desc2'*, and *'Refuse_Pct'*, along with the 20 nutritional features described above. To add costs to the dataset, we web-scraped the average USD prices and extracted caloric prices for each food item given their name specified in the *'Shrt_Desc'* feature. Out of 3901 food items, we successfully extracted USD prices for 3871 food items and caloric prices for 3125 food items. Therefore, when using USD prices as costs, there were 3871 possible hl-continuous actions, while using caloric prices meant 3125 possible hl-continuous actions.

**BMI dataset preprocessing.** The body mass index (BMI) dataset originally had 57 features. After removal of features with at least 20% null values and selecting the above nutritional features, except the feature *'total folate (mcg)'*, we had 23 features including: *'gender', 'age', 'race'*, and   *'body mass index (kg/m**2)'*. We selected agents whose age was greater than or equal to 20 at the time of surveys. Using the features *'body mass index (kg/m**2)'* and *'age'*, we computed the target label (binary class variable) for each agent as either healthy (1) BMI or unhealthy (0)  (WebMD, 2024). We then removed the feature *'body mass index (kg/m**2)'* and all the duplicates datapoints. At the end of data preprocessing, we did the 80/20 train/test data split resulting in 40734 data points in the predictive training set and 10184 in the predictive testing set.

**WHR dataset preprocessing.** Unlike the BMI dataset, there were fewer datapoints with 'waist-to-hip ratio' (WHR) information among the NHANES body measurement surveys (for years 1999 to prepandemic 2020) we scraped. First, we removed all features with at least 20% null values. Then using the features *'waist circumference (cm)', 'hip circumference (cm)'* and *'gender'*, we created the binary class variable *whr-class* (Wikipedia contributors, 2024), indicating healthy (1) or unhealthy (0) WHR. After preprocessing, we had 23 features, including the 20 nutritional features described above and the demographic features: *'gender', 'age'*, and *'race'*. Lastly, we removed the duplicates and split the dataset 80/20, creating 7296 data points in the predictive training set and 1824 in the predictive testing set.

### B.1.2 Behavioral Risk Factor Surveillance System (BRFSS) Dataset

The initial BRFSS dataset comprised 253680 rows and 22 features, each detailing various health and demographic attributes of agents (Teboul, 2024).

First, we removed all data points where '$Age$' = 1 denoting an age range of 18-24 because computation a new variable which relied on age being equal to or above 20 years, which reduced the dataset to $247,980$ rows. The new variable was called '$HealthBMI$,' an adult health BMI classification value (WebMD, 2024) from the feature '$BMI$.' Next, we transformed the existing features, which were predominantly binary, into new features where the 1 represents a desirable condition and 0 otherwise. We focused particularly on features we deemed actionable and renamed them to enhance their intuitiveness, specific to satisfiability. For instance, we renamed the feature '$HighBP$', which indicated high blood pressure (0 = no, 1 = yes), to '$LowBP$': {1 = yes (lowBP), 0 = no (highBP)}. Additionally, we removed six features '$CholCheck$,' $Diabetes\_012$,' '$Sex$,' '$Age$,' '$Education$,' and '$Income$,' and remained with 16 features.

These final 16 binary features included the following: '$LowBP$': {1 = yes (lowBP), 0 = no (highBP)}, '$LowChol$': {1 = yes (lowChol), 0 = no (highChol)}. The feature '$HealthBMI$': {1 =yes (healthy), 0 = no (unhealthy), '$NoSmoke$': {1 = yes, 0 = no}, '$NoStroke$': {1 = yes, 0 = no}, '$NoCHD$': {1 = yes, 0 = no}, '$PhysActivity$': {1 = yes, 0 = no}, '$Fruits$': {1 = yes, 0 = no}, '$Veggies$': {1 = yes, 0 = no}, '$LightAlcoholConsump$': {1 = yes, 0 = no}, '$AnyHealthcare$': {1 = yes, 0 = no}, '$DocbcCost$': {1 = yes, 0 = no}, '$GoodGenHlth$': {1 = excellent (1,2,3), 0 = bad (4,5)}, '$GoodMentHlth$': {1 = {1 = good ($< 2$), 0 = bad ($\geq 2$)}, '$GoodPhysHlth$': {1 = good ($< 2$), 0 = bad ($\geq 2$)}, and '$NoDiffWalk$': {1 = yes, 0 = no}.

Since we consider the setting where $\mathbf{t} = \mathbf{1}_{16}$, of the remaining data points, 8392 were considered to have a desirable outcome (no health risk) because all their features met the respective feature thresholds. Lastly, after removing the duplicate health risk agents and splitting the whole dataset 80/20, we had 11039 data points in the predictive training set and 2760 in the predictive testing set.

### B.2 Single-agent CFE Generation and Semi-synthetic agent–CFE Datasets

For all datasets, to determine which agents require CFEs (negatively classified agents), we use the classification models. Specifically, we trained logistic regression models on the BMI and WHR training sets, tuning the *solver* and *max_iter* hyperparameters using *GridSearchCV*. The best-performing models achieved test accuracies of 72.78% on the BMI dataset and 85.18% on the WHR dataset. For the BFRSS dataset, the threshold classifier $\mathbf{t} = \mathbf{1}_{16}$ achieved 100% test accuracy. We then used these models to determine the classifier parameters needed for single-agent CFE generation (see Equations 1, 2, and 3) and the specific agents requiring CFEs in the BMI, WHR, and BRFSS train/test datasets.

Below are details on the actionable features for each of the datasets. Refer to Appendix B.1.1 and Appendix B.1.2 for a detailed description of the meaning of the features.

**BMI actionable features.** For BMI agents states, we considered the following **19** actionable features: '$protein~(gm)$', '$carbohydrate~(gm)$', '$dietary~fiber~(gm)$', '$calcium~(mg)$', '$iron~(mg)$', '$magnesium~(mg)$', '$phosphorus~(mg)$', '$potassium~(mg)$', '$sodium~(mg)$', '$zinc~(mg)$', '$copper~(mg)$', '$selenium~(mcg)$', '$vitamin~C~(mg)$', '$niacin~(mg)$', '$vitamin~B6~(mg)$', '$vitamin~B12~(mcg)$', '$total~saturated~fatty~acids~(gm)$', '$total~monounsaturated~fatty~acids~(gm)$', and '$total~polyunsaturated~fatty~acids~(gm)$'.

**WHR actionable features.** For the generation of recourse for WHR agents, we use the following **20** actionable features: '$protein~(gm)$', '$carbohydrate~(gm)$', '$dietary~fiber~(gm)$', '$calcium~(mg)$', '$iron~(mg)$', '$magnesium~(mg)$', '$phosphorus~(mg)$', '$potassium~(mg)$', '$sodium~(mg)$', '$zinc~(mg)$', '$copper~(mg)$', '$selenium~(mcg)$', '$vitamin~C~(mg)$', '$niacin~(mg)$', '$vitamin~B6~(mg)$', '$total~folate~(mcg)$', '$vitamin~B12~(mcg)$', '$total~saturated~fatty~acids~(gm)$', '$total~monounsaturated~fatty~acids~(gm)$', and '$total~polyunsaturated~fatty~acids~(gm)$'.

**BRFSS actionable features.** Lastly, for the BRFSS agent states, we considered the following **16** actionable features: '$PhysActivity$', '$Fruits$', '$Veggies$', '$AnyHealthcare$', '$LowBP$', '$NoSmoke$', '$LowChol$', '$HealthBMI$', '$NoStroke$', '$NoCHD$', '$LightAlcoholConsump$', '$DocbcCost$', '$GoodGenHlth$', '$GoodMentHlth$', '$GoodPhysHlth$', and '$NoDiffWalk$'.

### B.2.1 The Low-level CFEs

Given negatively classified agents in the training and testing BMI, WHR and BRFSS datasets, and the actionable features for the corresponding datasets, we generate low-level CFEs using the low-level CFE generator (actionable recourse) (Ustun et al., 2019) described in Equation 1. Figure 4 illustrates examples of the generated low-level CFEs for the BMI, WHR and BRFSS datasets.

| Features to Change | Current Value | to | Required Value |
|---|---|---|---|
| Protein (gm) | 253.51 | → | 14.639999999999986 |
| Calcium (mg) | 1327 | → | 116 |
| Iron (mg) | 29.61 | → | 34.842000000000006 |
| Potassium (mg) | 6163 | → | 6370.618584999997 |
| Selenium (mcg) | 275.1 | → | 313.9095759999997 |
| Total monounsaturated fatty acids (gm) | 154.24 | → | 88.88112600000001 |

(a) for a BMI agent state

| Features to Change | Current Value | to | Required Value |
|---|---|---|---|
| Selenium (mcg) | 45 | → | 327.7319 |
| Total monounsaturated fatty acids (gm) | 12.392 | → | 89.34236700000017 |
| Total saturated fatty acids (gm) | 10.077 | → | 2.6004500000000004 |
| Vitamin B12 (mcg) | 1.21 | → | 0.1200000000000001 |
| Total folate (mcg) | 172 | → | 1179.7380000000003 |
| Vitamin B6 (mg) | 0.482 | → | 0.21794999999999998 |
| Niacin (mg) | 8.755 | → | 85.10721500000001 |
| Vitamin C (mg) | 35.7 | → | 0.10000000000000142 |
| Copper (mg) | 0.425 | → | 0.212950000000001 |
| Zinc (mg) | 2.61 | → | 1.3895 |
| Sodium (mg) | 1326 | → | 626.65 |
| Potassium (mg) | 994 | → | 6520.550000000004 |
| Phosphorus (mg) | 488 | → | 217 |
| Magnesium (mg) | 96 | → | 57 |
| Iron (mg) | 4.08 | → | 42.72946000000002 |
| Calcium (mg) | 309 | → | 113 |
| Total polyunsaturated fatty acids (gm) | 13.999 | → | 4.40896 |
| Carbohydrate (gm) | 109.45 | → | 43.376000000000005 |
| Dietary fiber (gm) | 4.1 | → | 50.113950000000024 |

(b) for a WHR agent state

| Features to Change | Current Value | to | Required Value |
|---|---|---|---|
| PhysActivity | 0 | → | 1 |
| Fruits | 0 | → | 1 |
| Veggies | 0 | → | 1 |
| LowBP | 0 | → | 1 |
| NoSmoke | 0 | → | 1 |
| LowChol | 0 | → | 1 |
| HealthBMI | 0 | → | 1 |
| NoStroke | 0 | → | 1 |
| GoodPhysHlth | 0 | → | 1 |
| NoDiffWalk | 0 | → | 1 |

(c) for a BRFSS agent state

Figure 4: Given a negatively classified BMI agent with actionable features presented in the order specified in Appendix B.2 and values [253.51, 352.76, 48.2, 1327., 29.61, 1204., 3966., 6163., 5890.0, 44.19, 7.903, 275.1, 30., 109.198, 3.492, 2.3, 59.686, 154.24, 113.429], the low-level CFE generator (cf. Equation 1) generates CFE (a) to help them become positive. On the other hand, given a negatively classified WHR agent with actionable features [29.03, 109.45, 4.1, 309., 4.08, 96., 488., 994., 1326., 2.61, 0.425, 45., 35.7, 8.755, 0.482, 172., 1.21, 10.077, 12.392, 13.999] in the order as described in Appendix B.2, the low-level CFE generator generates CFE (b). Lastly, the low-level CFE generator generates CFE (c) or an agent negatively classified based on their BRFSS features, with values [0, 0, 0, 1, 0, 0, 0, 0, 0, 1, 1, 1, 1, 1, 0, 0]. All the low-level CFEs ((a), (b) and (c)) are feature-based and precisely describe which features to change and by how much.

### B.2.2 The hl-continuous CFEs and agent–hl-continuous CFE Datasets

Below, we describe the single-agent hl-continuous CFE generation and the creation of the four semi-synthetic, agent–hl-continuous CFE datasets from the training/testing BMI and WHR datasets and the 2 forms of hl-continuous actions: Food+monetary costs and Food+caloric costs actions. Figure 6 shows examples of the generated hl-continuous CFEs for BMI and WHR agents.

For each negatively classified BMI and WHR train/test set agent, we generated hl-continuous CFEs using two types of actions: Foods+monetary costs and Foods+caloric costs. Leveraging the respective classifier parameters and the ILP formulation (Equation 2), we computed two distinct CFEs for each agent, one optimized for monetary cost and the other for caloric cost, each specifying an optimal set of food items.

As a result, we produced four unique agent–hl-continuous CFE datasets. We generated two CFEs for each training/testing BMI agent, one for each form of hl-continuous actions, yielding 40692 agent–CFE pairs for training and 10167 pairs for testing in each case. Similarly, for the WHR dataset, we generated 6387 training and 1603 testing agent–CFE pairs for both Foods+monetary and Foods+caloric costs.

### B.2.3 The hl-discrete CFEs and agent–hl-discrete CFE Datasets

Here, we outline the single-agent hl-discrete CFE generation and the process of generating the agent–hl-discrete CFE dataset, which consists of negatively classified BRFSS training/testing agents and their corresponding optimal synthetic hl-discrete CFEs. Figure 6 shows an example of the generated hl-discrete CFE for a BRFSS agent.

First, we generated 100 synthetic 16-dimensional, binary hl-discrete actions. The probability $p_a$ of an action satisfying feature eligibility was set to 0.5. The cost of satisfying feature eligibility was randomly predefined and remained uniform across all actions and agents. The total cost of an action was the sum of the costs associated with satisfying each feature's eligibility.

Next, we employed a threshold classifier with $\mathbf{t} = \mathbf{1}_{16}$ where an agent $\mathbf{x}$ is classified as not at health risk if $x_i \geq t_i$, $\forall i \in [n]$, and as a health risk otherwise. This classifier achieved perfect test accuracy $100.00\%$ on the BRFSS dataset, allowing us to accurately identify agents requiring CFEs in the training and testing datasets.

Using the identified agents, the synthetic hl-discrete actions, and the threshold classifier $\mathbf{t} = \mathbf{1}_n$, we applied Equation 3 to generate the agent–hl-discrete CFE dataset. As a result, we obtained $11,039$ agent–CFE pairs in the training dataset and $2,760$ in the testing agent–hl-discrete CFE dataset, where each CFE comprised optimal hl-discrete synthetic actions.

### B.2.4 The agent–hl-id CFE Datasets and other Variants

After creating the agent–hl-continuous CFE, agent–hl-discrete CFE, training/testing datasets, we generate other agent–CFE dataset variants from them.

**The agent–hl-id CFE datasets.** Given the agent–hl-continuous CFE and agent–hl-discrete CFE datasets described in Sections B.2.2 and B.2.3, we created corresponding agent–hl-id CFE datasets. This process involves encoding each CFE in the agent–CFE dataset with a unique identifier that distinguishes it from all other possible CFEs in that dataset. For example, given instances of agents–hl-discrete CFEs, we generate unique identifiers for all the hl-discrete CFEs to generate corresponding hl-id CFEs. At the end, we had 5 agent–hl-id CFE training/testing datasets.

**The semi-synthetic varied frequency of CFEs agent–CFE datasets.** For each of the generated agent–hl-continuous CFE, agent–hl-discrete CFE, and the agent–hl-id CFE training/testing datasets described above, we generate three frequency of CFE dataset variants: `all` (including all data), `>10` (more than 10 agents per CFE), and `>40` (more than 40 agents per CFE).

### B.3 The Fully-synthetic agent–CFE Datasets

We created five kinds of fully-synthetic agent–hl-discrete CFE datasets: varied dimension, frequency of CFEs, information access, feature satisfiability, and actions access. We provide statistical detailed information about the five variations of the agent–hl-discrete CFE datasets in Table 2 and Figure 5.

#### B.3.1 Varied Dimensions agent–CFE Datasets

We created 20-, 50- and 100-dimensional agent states datasets by varying the number of actionable features ($n = 20, 50, 100$) and keeping $p_f = 0.68$ the same for all datasets. We consider a unit vector threshold of length $n$. The cost associated with satisfying a feature's eligibility was predefined randomly and the same across all actions and agents. Each action was of length $n$, $p_a$ was 0.5, and action cost was the sum of the cost for each features the action fulfills. To create the 20-, 50- and 100-dimensional agent–hl-discrete CFE datasets, we computed the hl-discrete CFEs for each varied dimensional agent states datasets using the information above and the ILP defined in Equation 3.

#### B.3.2 Varied Frequency of CFEs agent–CFE Datasets

To investigate the effect of frequency of CFEs in the agent–CFE training set on the performance of the data-driven CFE generator, we create three varied frequency of CFEs agent–CFE datasets. For each of the varied dimensions agent–hl-discrete CFE datasets described in Appendix B.3.1, before the train/test split, we created three frequency-based agent–CFE datasets: `all`, where all data is included, `>10`, where we ensure a frequency of more than 10 agents per hl-discrete CFE, and `>40` with insurance of a frequency of more than 40 agents per hl-discrete CFE.

#### B.3.3 Varied Information Access agent–CFE Datasets

We construct varied information access agent–CFE datasets for synthetically generated agents and their corresponding fully-synthetic hl-discrete CFEs. That is, for each of the 20-, 50- and 100-dimensional agent–hl-discrete CFE datasets and their corresponding frequency-based datasets (`all`, `>10`, and `>40`), we created three varied information access datasets: agent–hl-discrete CFE dataset where the original hl-discrete CFE remains unchanged, the agent–hl-discrete-named CFE dataset where a unique name encodes each hl-discrete action in the hl-discrete CFE, and the agent–hl-discrete-id CFE dataset where a unique identifier denotes the entire hl-discrete CFE. For example, consider an agent $\mathbf{x} = [0, 0, 0, 0, 1]$ and their corresponding hl-discrete CFE given by $\{[0, 0, 1, 1, 0], [0, 1, 0, 0, 0], [1, 0, 0, 0, 0]\}$. The hl-discrete-named CFE $\{a, b, c\}$ where each hl-discrete action has a name (e.g., $a$) that uniquely identifies a specific hl-discrete action (e.g., $[0, 0, 1, 1, 0]$) among all hl-discrete actions. On the other hand, a unique name, say $z$, denotes the hl-discrete-id CFE, where $z$ uniquely represents this specific hl-discrete CFE among all the hl-discrete CFEs.

| Dataset name | Dataset size | One-action CFEs | Two-action CFEs | Three-action CFEs |
|---|---|---|---|---|
| 20-dimensional dataset | 71125 | 23687 | 44858 | 2576 |
| 50-dimensional dataset | 98966 | 1262 | 96770 | 934 |
| 100-dimensional dataset | 99728 | 0 | 45515 | 54213 |
| `manual groups` | 73484 | 13480 | 56653 | 3351 |
| `probabilistic groups` | 70226 | 44661 | 20258 | 5307 |
| `First10` | 74524 | 61794 | 12046 | 39 |
| `First5` | 74594 | 60656 | 6005 | 0 |
| `Last10` | 74401 | 53822 | 19952 | 1 |
| `Last5` | 74565 | 66068 | 644 | 0 |
| `Mid5` | 74594 | 63530 | 3010 | 0 |

Table 2: Statistics of some of the fully-synthetic agent–hl-discrete CFE datasets used in the experiments. Each hl-discrete CFE for each agent in all datasets has atmost 3 hl-discrete actions.

This setting aims to study the effectiveness of the data-driven CFE generators under various information access constraints within an agent–CFE training set, for example, (1) full access to hl-discrete actions and their effects on features (hl-discrete CFE), (2) access only to the names of hl-discrete actions without any information on how each action affects features (hl-discrete-named CFE), and (3) minimal information access, where only hl-discrete-id CFEs are known, with no explicit knowledge of the corresponding hl-discrete actions or their impact on features.

Given the agent–hl-discrete CFE varied information access datasets, we use the data-driven hl-continuous CFE generator to generate hl-discrete CFEs, data-driven hl-continuous CFE generator for hl-discrete-named CFEs, and data-driven hl-id CFE generators for hl-discrete-id CFEs.

### B.3.4  Varied Feature Satisfiability agent–CFE Datasets

Using the ILP formulation defined in Equation 3 with $n = 20$, and following the same agent and hl-discrete generation approach as in Appendix B.3.1 while varying the feature satisfiability for the threshold-based binary classifier (differing in which features are classifier-active (non-zero)), we generated five agent–hl-discrete CFE datasets. For the dataset `Last5`, the threshold vector is set as $\mathbf{t} = [15 \text{ zeros}, 5 \text{ ones}]$, while for the dataset `First5`, it is set as $\mathbf{t} = [5 \text{ ones}, 15 \text{ zeros}]$. The third dataset, `First10`, has a threshold vector of $\mathbf{t} = [10 \text{ ones}, 5 \text{ zeros}]$, and the dataset `Last10` has $\mathbf{t} = [10 \text{ zeros}, 10 \text{ ones}]$. Finally, the dataset `Mid5` has all features set to zero except for the five middle features set to one.

These varied feature satisfiability agent–hl-discrete CFE datasets are specifically created to investigate the effect of feature satisfiability on the nature of the hl-discrete CFEs and the effectiveness of the data-driven hl-continuous CFE generator at generating CFEs for new agents.

### B.3.5  Varied Access to Actions agent–CFE Datasets

Lastly, we consider two settings where grouped agents have restricted access to a set of actions: 1) `manual groups` where actions generated with the same probability $p_a = 0.5$ and agents are randomly assigned a restricted subset of actions; and 2) `probabilistic groups` where agents are assigned to groups and each group has its actions generated by different probabilities $p_a = [0.4, 0.5, 0.6, 0.7, 0.8]$. See Figure 5 for the statistics of the datasets.

We designed the varied access to actions agent–hl-discrete CFE datasets to empirically investigate fairness in CFE generation. Specifically, we examine the impact of restricting access of a group of agents to some actions on the nature of hl-discrete CFEs, such as CFE costs and the variations in accuracy of data-driven hl-continuous CFE generators across different groups.

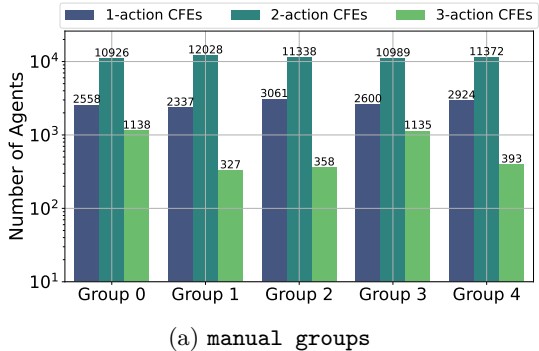

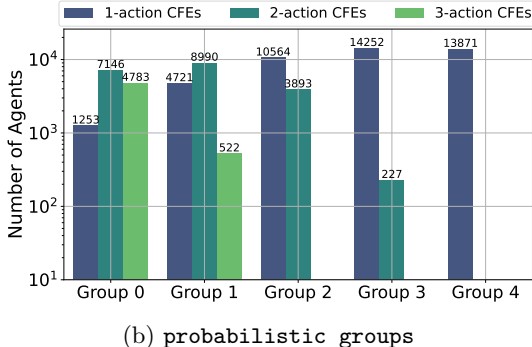

(a) `manual groups`        (b) `probabilistic groups`

Figure 5: Statistics on the varied access to actions agent–hl-discrete CFE datasets for `manual groups` and `probabilistic groups`. In the `probabilistic groups`, the action probability $p_a$ varies as follows: Group 0 ($p_a = 0.4$), Group 1 ($p_a = 0.5$), Group 2 ($p_a = 0.6$), Group 3 ($p_a = 0.7$), and Group 4 ($p_a = 0.8$). While the `manual groups` exhibit a more balanced distribution in terms of the number of actions taken by agents, the `probabilistic groups` introduce disparities where agents in certain groups have access only to more expensive and limited hl-discrete actions compared to others.

| | action-1 | action-2 |
|---|---|---|
| **PhysActivity** | 0 | 1 |
| **Fruits** | 0 | 1 |
| **Veggies** | 0 | 1 |
| **AnyHealthcare** | 0 | 0 |
| **LowBP** | 1 | 1 |
| **NoSmoke** | 0 | 1 |
| **LowChol** | 1 | 0 |
| **HealthBMI** | 0 | 1 |
| **NoStroke** | 1 | 0 |
| **NoCHD** | 0 | 0 |
| **LightAlcoholConsump** | 0 | 1 |
| **DocbcCost** | 1 | 0 |
| **GoodGenHlth** | 0 | 0 |
| **GoodMentHlth** | 1 | 0 |
| **GoodPhysHlth** | 1 | 0 |
| **NoDiffWalk** | 1 | 0 |

(a) for a BRFSS agent state

| | action-1 | action-2 | action-3 |
|---|---|---|---|
| **Protein (gm)** | 1.800 | 0.000 | 0.400 |
| **Carbohydrate (gm)** | 3.740 | 61.500 | 0.100 |
| **Dietary fiber (gm)** | 1.600 | 0.200 | 0.000 |
| **Calcium (mg)** | 51.000 | 8.000 | 13.000 |
| **Iron (mg)** | 1.800 | 3.720 | 0.300 |
| **Magnesium (mg)** | 81.000 | 2.000 | 11.000 |
| **Phosphorus (mg)** | 46.000 | 5.000 | 114.000 |
| **Potassium (mg)** | 379.000 | 16500.000 | 149.000 |
| **Sodium (mg)** | 213.000 | 52.000 | 215.000 |
| **Zinc (mg)** | 0.360 | 0.420 | 0.100 |
| **Copper (mg)** | 0.179 | 0.195 | 0.389 |
| **Selenium (mcg)** | 0.900 | 0.200 | 4.100 |
| **Vitamin C (mg)** | 30.000 | 0.000 | 1.000 |
| **Niacin (mg)** | 0.400 | 0.000 | 0.180 |
| **Vitamin B6 (mg)** | 0.099 | 0.000 | 0.010 |
| **Vitamin B12 (mcg)** | 0.000 | 0.000 | 5.000 |
| **Total saturated fatty acids (gm)** | 0.030 | 0.000 | 0.002 |
| **Total monounsaturated fatty acids (gm)** | 0.040 | 0.000 | 0.002 |
| **Total polyunsaturated fatty acids (gm)** | 0.070 | 0.000 | 0.006 |

(b) for a BMI agent state

| | action-1 | action-2 |
|---|---|---|
| **Protein (gm)** | 0.000 | 19.440 |
| **Carbohydrate (gm)** | 61.500 | 0.000 |
| **Dietary fiber (gm)** | 0.200 | 0.000 |
| **Calcium (mg)** | 8.000 | 17.000 |
| **Iron (mg)** | 3.720 | 1.630 |
| **Magnesium (mg)** | 2.000 | 23.000 |
| **Phosphorus (mg)** | 5.000 | 139.000 |
| **Potassium (mg)** | 16500.000 | 179.000 |
| **Sodium (mg)** | 52.000 | 247.000 |
| **Zinc (mg)** | 0.420 | 0.690 |
| **Copper (mg)** | 0.195 | 0.050 |
| **Selenium (mcg)** | 0.200 | 70.600 |
| **Vitamin C (mg)** | 0.000 | 0.000 |
| **Niacin (mg)** | 0.000 | 10.136 |
| **Vitamin B6 (mg)** | 0.000 | 0.319 |
| **Total folate (mcg)** | 0.000 | 4.000 |
| **Vitamin B12 (mcg)** | 0.000 | 2.550 |
| **Total saturated fatty acids (gm)** | 0.000 | 0.211 |
| **Total monounsaturated fatty acids (gm)** | 0.000 | 0.107 |
| **Total polyunsaturated fatty acids (gm)** | 0.000 | 0.277 |

(c) for a WHR agent state

Figure 6: For an agent negatively classified based on their BRFSS features, with values $[0, 0, 0, 1, 0, 0, 0, 0, 0, 1, 1, 1, 1, 1, 0, 0]$ in order similar to (a), the hl-discrete CFE generator recommends hl-discrete CFE(a) with hl-discrete actions, **action-1** and **action-2**. Additionally, for a negatively classified BMI agent, given their actionable features with values $[253.51, 352.76, 48.2, 1327., 29.61, 1204., 3966., 6163., 5890.0, 44.19, 7.903, 275.1, 30., 109.198, 3.492, 2.3, 59.686, 154.24, 113.429]$, arranged in the same order as features shown in (b), the hl-continuous CFE generator recommends CFE(b) containing the following hl-continuous actions: **action-1**: *take Swiss chard, raw*, **action-2**: *take leavening agents: cream of tartar*), and **action-3**: *take clams, mixed species, canned, in liquid.* Similarly, for a negatively classified WHR agent with actionable feature values ordered as features in (c) $[29.03, 109.45, 4.1, 309., 4.08, 96., 488., 994., 1326., 2.61, 0.425, 45., 35.7, 8.755, 0.482, 172., 1.21, 10.077, 12.392, 13.999]$ the hl-continuous CFE generator recommends CFE(c) with the following hl-continuous actions: **action-1**: *take leavening agents: cream of tartar* and **action-2**: *take fish, tuna, light, canned in water, drained solids.*

## C  Data-driven CFE Generators: Supplemental Details

This section includes supplemental details about the architectures of the data-driven CFE generators, details about other baseline models, and future works. Although we do not explicitly create a separate validation set during the initial 80/20 data split for training and testing, we use "*validation_split*" when training all the generator models.

### C.1  The Data-driven hl-continuous CFE Generator

The neural-network hl-continuous CFE generator we use in these experiments is susceptible to imbalance and overfitting. Therefore, we weight and regularize the loss function $\mathcal{L}_{\mathrm{HC}}$ in Equation 4 as follows:

$$\mathcal{L}_{\mathrm{HC}}^{w} = p_w \mathcal{L}_{\mathrm{HC}} + \alpha \frac{1}{M} \sum_{m=1}^{M} ||\hat{a}_m - a_m||_1 \tag{8}$$

The weighting factor $p_w$ weights $\mathcal{L}_{\mathrm{HC}}$ by scaling the contribution of each agent to the loss function. The term $\alpha \frac{1}{M} \sum_{m=1}^{M} ||\hat{a}_m - a_m||_1$ regularizes the model, thus preventing overfitting by nudging the model towards producing hl-continuous CFEs closer to $a_m$'s distribution. We, on average chose the values of $\alpha$ from the set $\{0.05, 0.1, 0.07\}$ and $p_w$ from $\{0.05, 0.1, 0.07\}$.

### C.2  The Data-driven hl-discrete CFE Generator

Below is the architecture of the neural-network based data-driven hl-discrete CFE generator described in the main paper.

### C.3  The Hamming Distance Data-driven CFE Generator

To produce hl-discrete-id CFEs (refer to Appendix B.3.3) for new agents, we mainly used the data-driven hl-id CFE generator. However, we wanted to investigate the effect of model complexity on the accuracy of CFE generation. Therefore, we compare the more complex data-driven hl-id CFE generator (refer to Section 4.3) with a basic model, e.g., Hamming distance-based CFE generator, whose choice is due to the agent features being binary for this setting. Below is a description of the Hamming distance hl-discrete-id CFE generator.

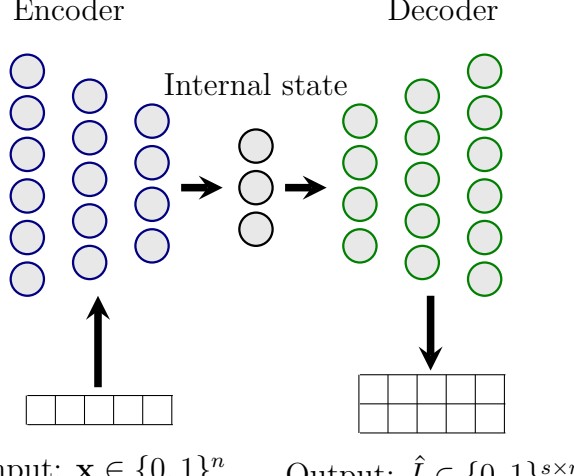

Figure 7: An encoder-decoder data-driven hl-discrete CFE generator, where $n$ is the data dimension and $s$ is the number of hl-discrete actions in the generated hl-discrete CFE $\hat{I}$.

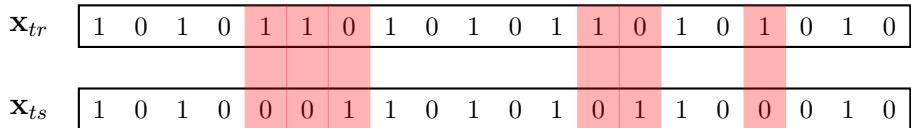

$$\text{Hamming Distance: } (\mathbf{x}_{tr}, \; \mathbf{x}_{ts}) = 6$$

Figure 8: Hamming distance between the agent–CFE training set agent $\mathbf{x}_{tr}$ and a testing set agent $\mathbf{x}_{ts}$.

Given a negatively classified new agent $\mathbf{x}_{ts}$, we compute the Hamming distance (see Figure 8) between them and each of the agents $\mathbf{x}_{tr}$ in the agent–hl-discrete-id CFE training set. Then, based on these distances, we choose the $k$ nearest training set agents and their associated hl-discrete-id CFEs. We then use the most common hl-discrete-id CFE as the hl-discrete-id CFE for the new agent $\mathbf{x}_{ts}$. We experimented with varied number of nearest neighbors: $5, 10$ and $15$, for the 20-, 50- and 100-dimensional agent–hl-discrete-id CFE datasets, respectively.

### C.4 Extensions of Data-driven CFE Generation

**Robust data-driven CFE generators.** CFE generators, whether single-agent, global, or data-driven, can be explicitly or implicitly compromised by changes in the underlying classifier, especially if that classifier is inaccurate or evolves with time. Low-level, feature-based CFEs, such as those proposed by Ustun et al. (2019), are particularly vulnerable: their specificity makes them highly sensitive to even minor modifications in the classification model. Data-driven CFE generators are also at risk of model drift and classification errors, primarily due to their reliance on agent–CFE datasets, implicitly or explicitly shaped by aggregation methods that depend on the current classifier.

To address issues related to classification model errors and changes, recent robust CFE generation approaches (see (Jiang et al., 2024)) account for model and distribution drift during the CFE generation process, which would improve the overall performance of all CFE generators, whether single-agent, global, or data-driven.

A key direction for future research is to investigate the robustness of data-driven CFE generators under model drift. In particular, it would be valuable to assess whether high-level CFEs offer improved resilience to model changes or errors compared to feature-based, low-level CFEs. Furthermore, future work could focus on valuing agent–CFE data instances and developing robust aggregation and data-sharing strategies (e.g., federated learning) to enhance the overall performance of data-driven CFE generators.

**Extension to multiple valid CFEs generation.** In the agent–CFE dataset, each agent is associated with a single valid CFE of minimal cost. The probability of agents possessing multiple unique valid CFEs with identical minimal costs was negligible because we assigned unique feature eligibility costs, resulting in varied overall hl-discrete action costs and uniquely optimal CFEs. In the rare cases where such ties occurred, we excluded the corresponding agents from the dataset.

Although we ensure each agent has a unique optimal CFE in the agent–CFE dataset, other valid CFEs that flip the prediction may exist but incur marginally higher costs. This scenario is more prevalent when generating sets of actions, whether hl-discrete or hl-continuous, as different combinations may achieve the same classification outcome at a higher cost.

Our current evaluation function (Equation 6) is overly strict, disproportionately penalizing valid CFEs that incur higher costs. A promising avenue for future work involves extending data-driven CFE generators to produce sets of valid CFEs rather than single optimal ones. Instead of training the generators on agent–CFE pairs (each agent mapped to a unique minimal-cost CFE), train the generators on agent–validCFE sets, where each agent is associated with multiple valid CFEs. During inference, evaluate each generated CFE for prediction flip and relative cost. This approach broadens the CFE search space, potentially enhancing robustness of the CFE generators.

# D  Evaluation and Comparative Analysis Metrics: Supplemental Details

Here, we provide additional details on the evaluation metrics used to compare low-level CFEs with both hl-continuous and hl-discrete CFEs, as well as to assess the effectiveness of data-driven CFE generators.

## D.1  Comparison Metrics

We evaluate each key variable ($v$), such as the number of actions taken, features modified, agents using the same CFE, and agent improvement, when agents follow a low-level CFE vs. a high-level CFE (hl-continuous, or hl-discrete). Specifically, using the general Equation 9, we define comparison metric $\delta_v$ that compares each variables when an agent follows a low-level CFE versus an hl-continuous or hl-discrete CFE.

$$\delta_v(P, Q) = P_v - Q_v \tag{9}$$

Where $P$ and $Q$ denote two CFEs under consideration, e.g., $P$ may correspond to a low-level CFE and $Q$ to an hl-discrete CFE. The terms $P_v$ and $Q_v$ denote the variable value, such as the number of actions taken, following the execution of each CFE.

For all variables, a positive $\delta_v$ indicates that the low-level CFE variable value is higher than the compared CFE, while a negative $\delta_v$ indicates the opposite. The magnitude of $\delta_v$ reflects the extent of this difference. Below are the specific $\delta_v$ metrics.

**Difference in number of actions taken.** The metric $\delta_{\text{actions}}(\cdot, \cdot)$ (Equation 10) quantifies the difference in the number of actions taken when an agent executes a low-level CFE versus an hl-continuous or hl-discrete CFE.

$$\delta_{\text{actions}}(P, Q) = P_{\text{actions}} - Q_{\text{actions}} \tag{10}$$

Here, $P$ and $Q$ represent the low-level CFE and the hl-continuous or hl-discrete CFE, respectively, while $P_{\text{actions}}$ and $Q_{\text{actions}}$ denote the number of actions taken when executing each.

**Difference in agent improvement.** The metric $\delta_{\text{improvement}}(\cdot, \cdot)$ (Equation 11) quantifies the difference in the agent improvement earned when an agent executes a low-level CFE versus an hl-continuous or hl-discrete CFE.

$$\delta_{\text{improvement}}(P, Q) = P_{\text{improvement}} - Q_{\text{improvement}} \tag{11}$$

Here, $P$ and $Q$ represent the low-level CFE and the hl-continuous or hl-discrete CFE, respectively, while $P_{\text{improvement}}$ and $Q_{\text{improvement}}$ denote the agent improvement earned when executing each. Specifically,

$$P_{\text{improvement}} = \|\mathbf{x}' - \mathbf{x}\| \tag{12}$$

where $P$ is the CFE taken and $\mathbf{x}'$ is the resultant agent state after taking the CFE from $\mathbf{x}$, which is the initial agent state. Ideally high improvement (less proximate), that is, $\mathbf{x}'$ more distant from $\mathbf{x}$ is preferred.

**Difference in number of features modified.** The metric $\delta_{\text{features}}(\cdot, \cdot)$ (Equation 13) quantifies the difference in the number features modified when an agent executes a low-level CFE versus an hl-continuous or hl-discrete CFE.

$$\delta_{\text{features}}(P, Q) = P_{\text{features}} - Q_{\text{features}} \tag{13}$$

Here, $P$ and $Q$ represent the low-level CFE and the hl-continuous or hl-discrete CFE, respectively, while $P_{\text{features}}$ and $Q_{\text{features}}$ denote the number of features modified when executing each.

## D.2  Statistical Significance between Variables

Given the different variables, e.g., list of the number of actions taken, number of modified features, and improvement achieved with each CFE: hl-continuous, hl-discrete, and low-level, we compute the statistical significance of the differences. We use the Scipy stats tool (Developers, 2023) to compute the Kendall tau and $p$-value to assess the statistical significance of the relationship between the two variables at a time.

# E   Experimental Results: Supplemental Details

In this section, we provide additional and thorough empirical evidence demonstrating the strong performance of the proposed data-driven CFE generators in producing optimal CFEs for new agents. We also highlight the strong and desirable characteristics of the hl-continuous and hl-discrete CFEs over the low-level CFEs. Lastly, we analyze how various constraints, such as varied data dimensions, the frequency of CFEs, decision-makers information access, feature satisfiability, and restrictions on agents' access to actions, affect the agent–CFE data distribution and the effectiveness of data-driven CFE generators.

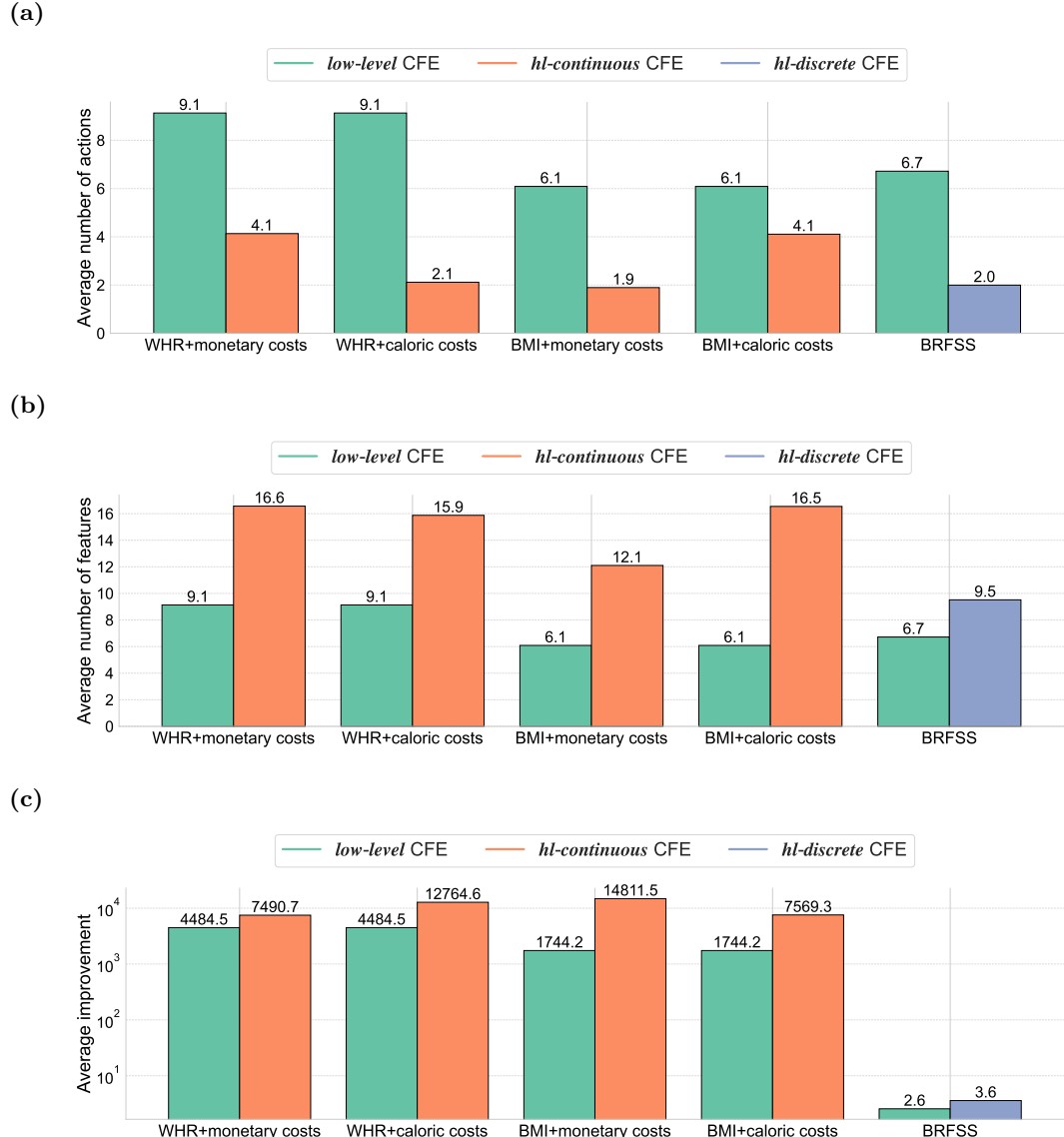

Figure 9: A comparison of hl-continuous CFEs consisting of a set of Food+monetary or Food+caloric cost hl-continuous actions for WHR and BMI datasets, alongside hl-discrete CFEs on the BRFSS dataset, evaluated against low-level CFEs for their respective datasets. All annotations up to one decimal place, low-level CFEs require (a) more actions but lead to (b) fewer feature modifications and (c) result in less improvement (i.e., closer resultant agent states) compared to hl-discrete and hl-continuous CFEs.

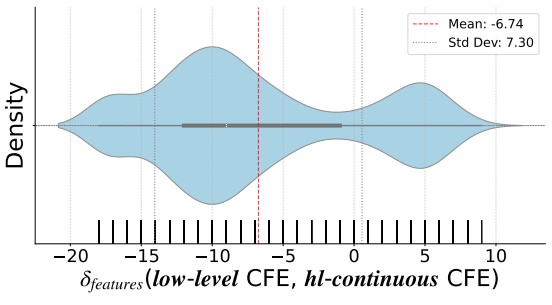 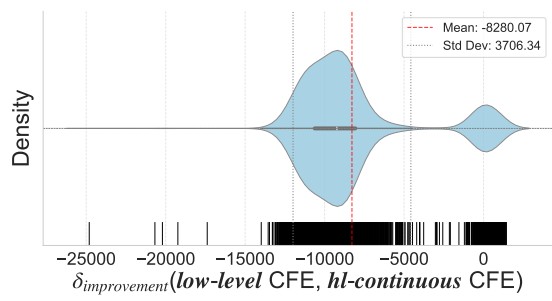

(a) Difference in number of modified features

(b) Difference in agent improvement

Figure 10: Given WHR negatively classified agents and the low-level and hl-continuous CFEs they took, a computation of $\delta_{\mathrm{improvement}}(P, Q)$ (Equation 11)} and $\delta_{\mathrm{features}}(P, Q)$ (Equation 13) where $P$ denotes taking a low-level CFE and $Q$ denotes taking an hl-continuous CFE, shows that most of the density is negative implying that when agents take hl-continuous CFEs, a higher number of their features is modified (a) and resultant improvement is significantly higher (b) than if they took low-level CFEs.

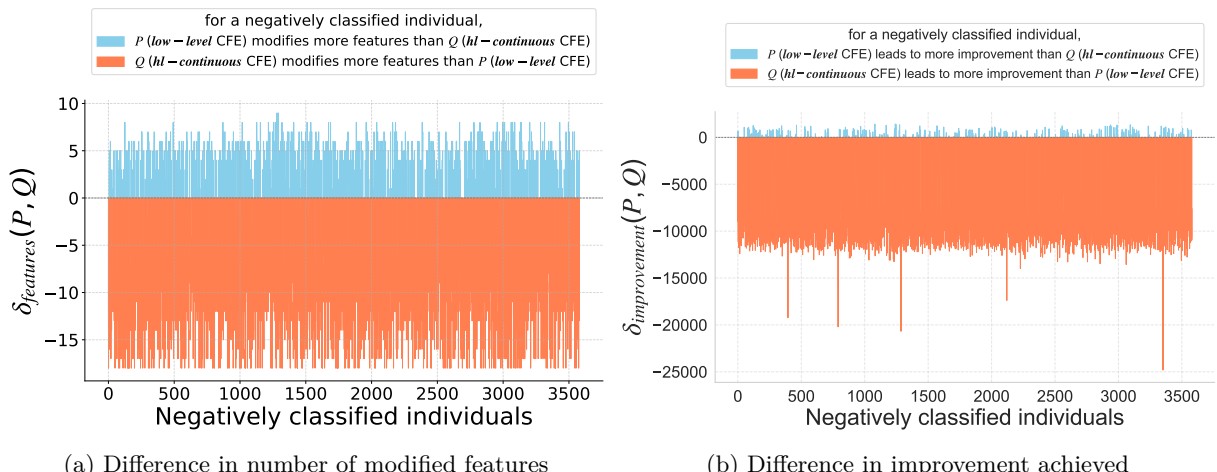

(a) Difference in number of modified features

(b) Difference in improvement achieved

Figure 11: A comparative analysis of the (a) difference in number of modified features ($\delta_{\mathrm{features}}(P, Q)$ (Equation 13)) and (b) difference in agent improvement ($\delta_{\mathrm{improvement}}(P, Q)$ (Equation 11)) when each negatively classified WHR agent takes a low-level CFE ($\mathbf{P}$) versus an hl-continuous CFE ($\mathbf{Q}$), shows that hl-continuous CFEs modify more features (a) and lead to significantly higher improvement (b) than low-level CFEs.

### E.1 High-level CFEs Result in Higher Improvement and More Feature Modifications

Unlike low-level CFEs, high-level CFEs (hl-continuous and hl-discrete CFEs) involve fewer actions on average (see Figure 9(a)) and results in higher improvements (Figures 9(c), 10(b), and 11(b)) and simultaneously modify multiple features (see Figures 9(b), 10(a), and 11(a)).

While low-level CFEs exhibit a perfect correlation between the number of actions taken and the number of features modified, hl-continuous and hl-discrete CFEs show a positive but weaker relationship (Figure 12(b)). Additionally, hl-discrete and low-level CFEs have a strong positive correlation ($\tau = 0.708$) in the number of modified features (see BRFSS dataset in Figure 12(a)). In contrast, hl-continuous CFEs show a weak negative correlation with low-level CFEs in the number of modified features and actions taken (see BMI and WHR datasets in Figure 12(a), $\tau = -0.2684$ and $\tau = -0.233$ respectively).

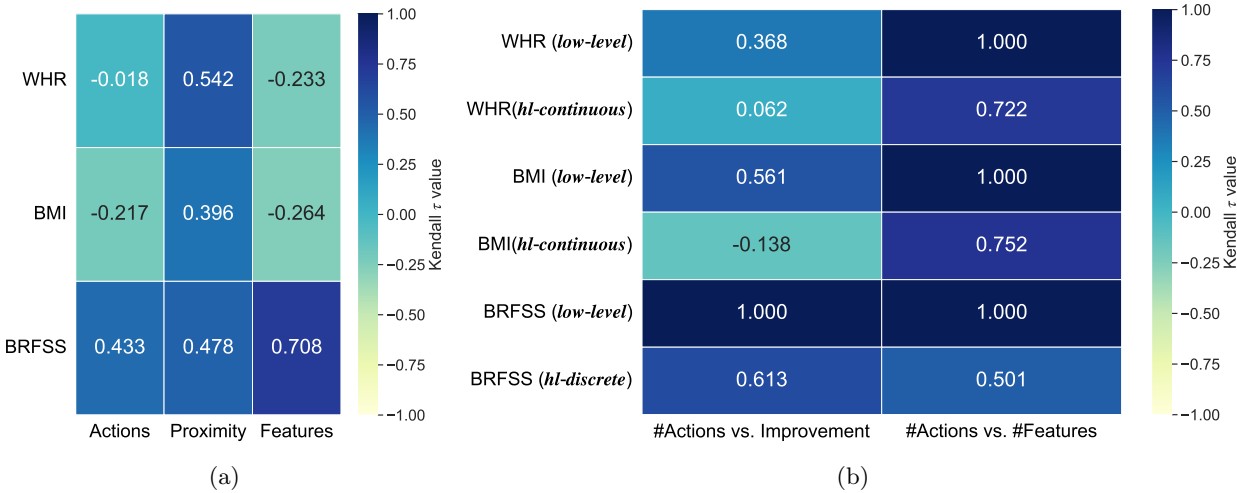

Figure 12: In (a), we illustrate the correlations for three different aspects: (1) between the number of actions taken with CFEs P and Q, (2) between the number of features modified with CFEs P and Q, and (3) between the improvement achieved after taking CFEs P and Q. For the BMI and WHR datasets, P and Q represent low-level and hl-continuous CFEs, respectively. For the BRFSS dataset, P and Q denote low-level and hl-discrete CFEs, respectively. On the other hand, (b) shows the correlation between the number of actions taken and the number of modified features and between the number of actions taken and improvement achieved for each CFE and dataset. In general, low-level CFEs have a perfect positive relationship between the number of actions and modified features

## E.2    High-level CFEs are Easier to Personalize and Lead to Fairer Outcomes

Fairness in CFE generation has primarily been studied along the dimension of equalizing the recourse costs across different groups (e.g., (Gupta et al., 2019)). In this work, we extend the analysis by exploring several dimensions of fairness in CFE generation.

First, we investigate how agents across sensitive groups using the same CFE generator (same kind of CFEs) experience differences in how much they improve, the number of actions taken, the number of modified features, and the costs incurred. Second, we explore the effects of limiting agents to a subset of actions (varied access to actions) on the distribution of agent–CFE datasets and the accuracy of data-driven CFE generators across groups. Lastly, we examine variations in feature satisfiability (differences in what features need to be satisfied) across agent groups, influences the distribution of the agent–CFE dataset, and the performance of data-driven CFE generators in generating CFEs for different agent groups.

In addition to fairness, we also investigate the personalization of CFE generation along two dimensions. 1) Agents may be interested in a subset of actions (varied access to actions) and thus restricted to CFEs that involve only specific actions. 2) Agents might prioritize different costs in the CFE generation process (varied cost preferences) and thus prefer CFE generators that optimize those specific costs in CFE generation, e.g., caloric costs over monetary ones.

### E.2.1    Fairness Based on Variability of CFEs Execution Outcome

We analyze variations in costs incurred, actions taken, features modified, and agent improvement across sensitive groups to assess the fairness of low-level CFEs compared to hl-continuous and hl-discrete CFEs.

**Variability in improvement, number of modified features, and number of actions taken.** To quantify differences in agent experiences across sensitive groups, we compute the coefficient of variation for three key variables: improvement, number of modified features, and number of actions taken. Figure 13 illustrates that in the WHR dataset, low-level CFEs exhibit substantial variation across sensitive groups

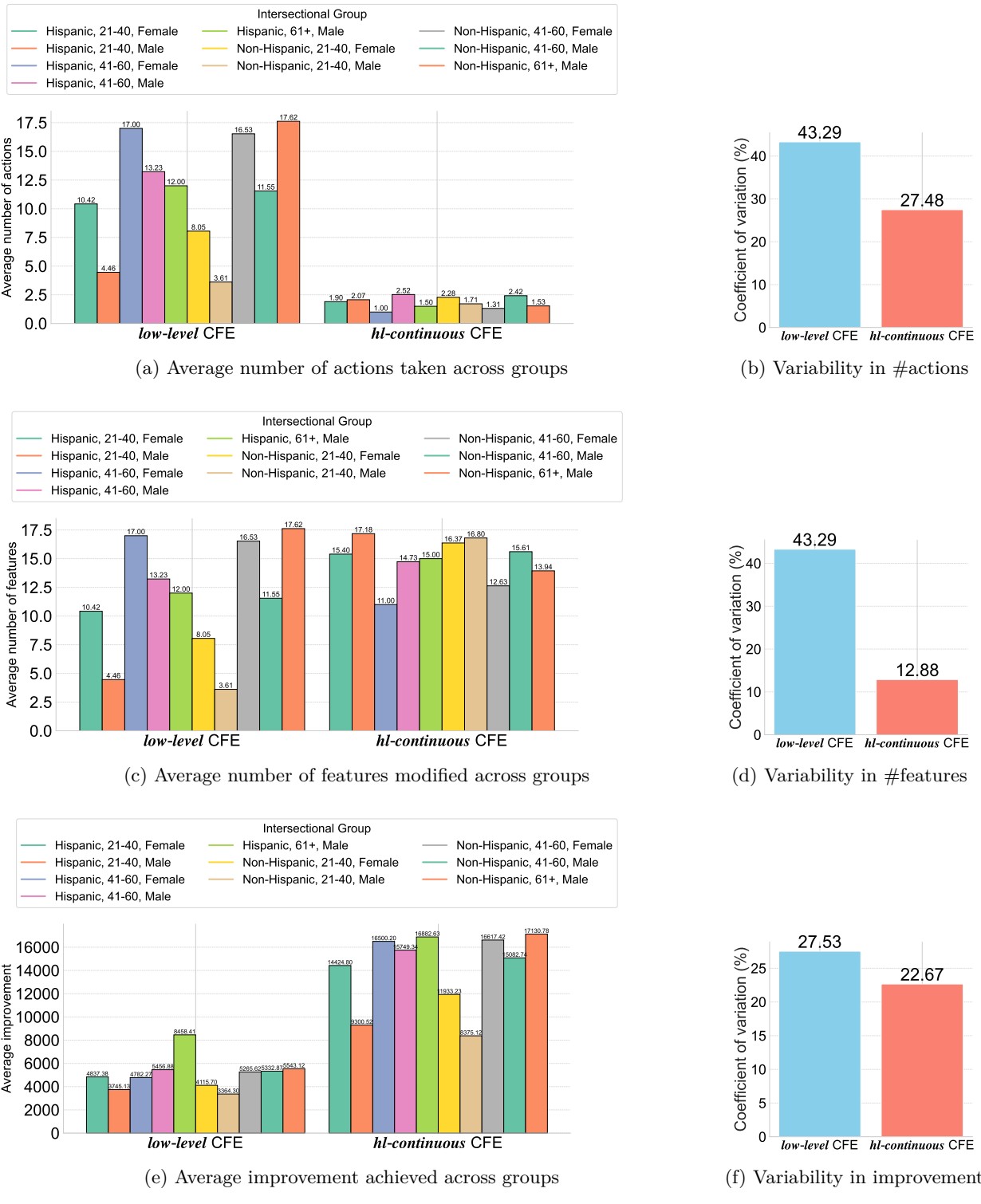

Figure 13: A comparative analysis of the average number of actions taken, the number of features modified, and the improvement achieved by agents across sensitive groups when they take low-level CFEs versus hl-continuous CFEs. (a), (c), and (e) show the raw distributions for these variables across sensitive groups while (b), (d), and (f) present the coefficients of variation that concisely illustrate the extent of dispersion around the mean for each variable. In summary, low-level CFEs are less fair than hl-continuous CFEs, which exhibit lower coefficients of variation, ensuring more comparable outcomes for agents from different groups.

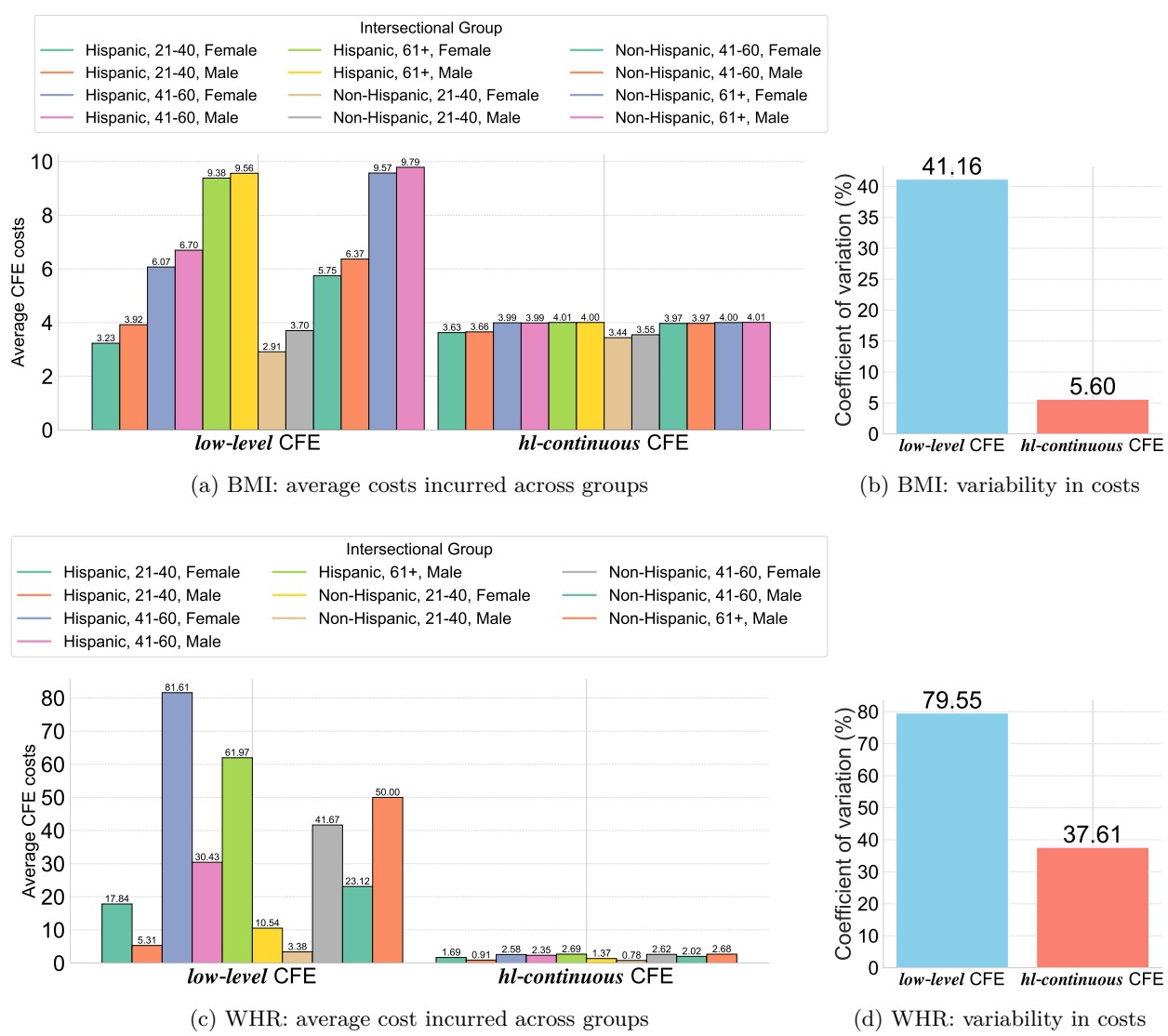

(a) BMI: average costs incurred across groups

(b) BMI: variability in costs

(c) WHR: average cost incurred across groups

(d) WHR: variability in costs

Figure 14: A comparative analysis of average cost variations among agents in sensitive groups when taking low-level CFEs versus high-level continuous CFEs. Although not comparable across CFEs, (a) and (c) show the distribution of costs between groups for each CFE, and (b) and (d) show the coefficient of variations - indicating how variable around mean the average costs in groups are. Costs across sensitive groups vary more when agents take low-level CFEs than when they take hl-continuous CFEs.

regarding agent improvement, actions taken, and features modified. Specifically, the coefficient of variation for agent improvement was 27.53% with low-level CFEs versus 22.67% otherwise. The variation in the number of actions taken was 43.29% compared to 27.48%, and for modified features, 43.29% compared to 12.88%. These findings indicate that the benefits of low-level CFEs are not distributed equally across sensitive groups, potentially favoring some over others, thus raising fairness concerns in CFE generation.

**Variability in costs incurred across sensitive groups.** Although the costs agents incur by taking low-level CFEs cannot be directly compared with taking hl-continuous CFEs because they are contextually different (feature-based vs action-based), we study how the costs of executing the same kind of CFEs varies across agents in different sensitive groups.

Our results show that the costs incurred in taking low-level CFEs vary more widely across various sensitive groups than in taking hl-continuous CFEs. For example, in Figure 14, the coefficient of variation for taking

low-level CFEs is 41.16% and 79.55% versus 5.60% and 37.61% with taking hl-continuous CFEs, on BMI and WHR datasets, respectively. Therefore, compared to taking hl-continuous CFEs, taking low-level CFEs is more biased and more likely to favor some sensitive groups.

| Features to Change | Current Value | to | Required Value |
|---|---|---|---|
| Selenium (mcg) | 45 | → | 327.7319 |
| Total monounsaturated fatty acids (gm) | 12.392 | → | 89.34236700000017 |
| Total saturated fatty acids (gm) | 10.077 | → | 2.6004500000000004 |
| Vitamin B12 (mcg) | 1.21 | → | 0.1200000000000001 |
| Total folate (mcg) | 172 | → | 1179.7380000000003 |
| Vitamin B6 (mg) | 0.482 | → | 0.21794999999999998 |
| Niacin (mg) | 8.755 | → | 85.10721500000001 |
| Vitamin C (mg) | 35.7 | → | 0.10000000000000142 |
| Copper (mg) | 0.425 | → | 0.212950000000001 |
| Zinc (mg) | 2.61 | → | 1.3895 |
| Sodium (mg) | 1326 | → | 626.65 |
| Potassium (mg) | 994 | → | 6520.550000000004 |
| Phosphorus (mg) | 488 | → | 217 |
| Magnesium (mg) | 96 | → | 57 |
| Iron (mg) | 4.08 | → | 42.72946000000002 |
| Calcium (mg) | 309 | → | 113 |
| Total polyunsaturated fatty acids (gm) | 13.999 | → | 4.40896 |
| Carbohydrate (gm) | 109.45 | → | 43.376000000000005 |
| Dietary fiber (gm) | 4.1 | → | 50.113950000000024 |

(a) low-level CFE

| | action-1 | action-2 |
|---|---|---|
| **Protein (gm)** | 1.250 | 0.000 |
| **Carbohydrate (gm)** | 3.350 | 61.500 |
| **Dietary fiber (gm)** | 3.100 | 0.200 |
| **Calcium (mg)** | 52.000 | 8.000 |
| **Iron (mg)** | 0.830 | 3.720 |
| **Magnesium (mg)** | 15.000 | 2.000 |
| **Phosphorus (mg)** | 28.000 | 5.000 |
| **Potassium (mg)** | 314.000 | 16500.000 |
| **Sodium (mg)** | 22.000 | 52.000 |
| **Zinc (mg)** | 0.790 | 0.420 |
| **Copper (mg)** | 0.099 | 0.195 |
| **Selenium (mcg)** | 0.200 | 0.200 |
| **Vitamin C (mg)** | 6.500 | 0.000 |
| **Niacin (mg)** | 0.400 | 0.000 |
| **Vitamin B6 (mg)** | 0.020 | 0.000 |
| **Total folate (mcg)** | 142.000 | 0.000 |
| **Vitamin B12 (mcg)** | 0.000 | 0.000 |
| **Total saturated fatty acids (gm)** | 0.048 | 0.000 |
| **Total monounsaturated fatty acids (gm)** | 0.004 | 0.000 |
| **Total polyunsaturated fatty acids (gm)** | 0.087 | 0.000 |

(b) hl-continuous CFE with caloric costs

| | action-1 | action-2 |
|---|---|---|
| **Protein (gm)** | 0.000 | 19.440 |
| **Carbohydrate (gm)** | 61.500 | 0.000 |
| **Dietary fiber (gm)** | 0.200 | 0.000 |
| **Calcium (mg)** | 8.000 | 17.000 |
| **Iron (mg)** | 3.720 | 1.630 |
| **Magnesium (mg)** | 2.000 | 23.000 |
| **Phosphorus (mg)** | 5.000 | 139.000 |
| **Potassium (mg)** | 16500.000 | 179.000 |
| **Sodium (mg)** | 52.000 | 247.000 |
| **Zinc (mg)** | 0.420 | 0.690 |
| **Copper (mg)** | 0.195 | 0.050 |
| **Selenium (mcg)** | 0.200 | 70.600 |
| **Vitamin C (mg)** | 0.000 | 0.000 |
| **Niacin (mg)** | 0.000 | 10.136 |
| **Vitamin B6 (mg)** | 0.000 | 0.319 |
| **Total folate (mcg)** | 0.000 | 4.000 |
| **Vitamin B12 (mcg)** | 0.000 | 2.550 |
| **Total saturated fatty acids (gm)** | 0.000 | 0.211 |
| **Total monounsaturated fatty acids (gm)** | 0.000 | 0.107 |
| **Total polyunsaturated fatty acids (gm)** | 0.000 | 0.277 |

(c) hl-continuous CFE with monetary costs

Figure 15: When given actionable features values $[29.03, 109.45, 4.1, 309., 4.08, 96., 488., 994., 1326., 2.61, 0.425, 45., 35.7, 8.755, 0.482, 172., 1.21, 10.077, 12.392, 13.999]$, in the same order as shown in (b) and (c), for a negatively classified WHR agent, the low-level CFE generator recommends a CFE (a) with a cost of 56.588. This CFE was unique to the agent. In contrast, the hl-continuous CFE generator generates two CFEs optimized for different agent's preferences. When optimizing for caloric cost, the CFE generator generates CFE (a) with a cost of 2.750. This CFE, which was also optimal for other 25 negatively classified agents, includes **action-1** (*consume endive, raw*) and **action-2** (*consume leavening agents: cream of tartar*). When optimizing for monetary cost, the CFE generator produces a CFE (b) of cost 4.010. This CFE, also optimal for other 105 agents, consists of **action-1** (*consume leavening agents: cream of tartar*) and **action-2** (*consume fish, tuna, light, canned in water, drained solids*). Lastly, while the low-level CFE (a) involves 19 actions but modifies 19 features and improve by 5679.95, the hl-continuous CFEs both involve 2 actions but modify 19 features and lead to an improvement of 16815.04 (b) and 16682.62 (c).

### E.2.2 Varied Costs Preferences

We model two types of hl-continuous CFEs: a set of Foods+monetary costs hl-continuous actions and a set of Foods+caloric costs hl-continuous actions (see Appendix B.1.1 and B.2.2). In a setting where negatively classified agents care more about monetary costs over caloric costs, and vice versa, the CFE generator adapts to these preferences and recommends the corresponding optimal CFE, as demonstrated in Figure 15.

Additionally, regardless of whether monetary or caloric costs were the desired costs by the agent, we consistently observed that taking hl-continuous CFEs involved fewer actions, resulted in more feature modifications and higher improvement when compared to low-level CFEs (see Figures 13 and Figure 15 ). Future research could investigate the data-driven CFE generation at the intersection of various settings. For instance, this could involve exploring Pareto-optimal solutions where agents seek to simultaneously optimize multiple factors, such as monetary and caloric costs.

### E.2.3 Varied Feature Satisfiability

In general, as shown in Table 2, compared to the unit threshold datasets: 20- 50- and 100-dimensional agent–CFE datasets, agents in the varied binary feature satisfiability datasets described in Appendix B.3.4 required fewer action due to the fewer features to satisfy to get a desirable classification.

Our results show that without explicit knowledge of the varied feature satisfiability when given testing set agents, the data-driven hl-discrete CFE generator trained on instances of a mixture of varied feature satisfiability agent–hl-discrete CFE datasets successfully generates the right hl-discrete CFEs for the new agents. The data-driven hl-discrete CFE generator achieves an accuracy of 99.683% on `First10`, 99.496% on `Last10`, 100% on `First5`, 100% on `Mid5`, and 100% on `Last5`, dataset variants.

### E.2.4 Varied Access to Actions

The `manual groups` agent–hl-discrete CFE datasets (described in Appendix B.3.5) are more balanced in terms of the number of actions agents take (see Figure 5(a)). The reason is agents have access to the same distribution of hl-discrete actions, i.e., although agents in each group have access to only a selected group of hl-discrete actions, all the hl-discrete actions for all groups were generated with the same probability, $p_a = 0.5$.

However, for the `probabilistic groups` agent–hl-discrete CFEs datasets (described in Appendix B.3.5), Figure 5(b) shows that as the probability of hl-discrete capabilities $p_a$ decreases, the number of hl-discrete agents require to get all the necessary capabilities to transform their states to get a positive model outcome increases. In other words, agents in certain groups only have access to more expensive and limited hl-discrete actions compared to others. For instance, agents in the `probabilistic groups` Group 0 face more difficulty (due to limited capabilities and more costly hl-discrete actions) in achieving positive classification outcomes than those in the Group 4.

| Manual Groups | | Probabilistic Groups | |
|---|---|---|---|
| Group | Accuracy | Group | Accuracy |
| Group 0 | $0.881 \pm 0.01200$ | Group 0 (0.4) | $0.880 \pm 0.04400$ |
| Group 1 | $0.871 \pm 0.01260$ | Group 1 (0.5) | $0.771 \pm 0.02081$ |
| Group 2 | $0.875 \pm 0.01249$ | Group 2 (0.6) | $0.802 \pm 0.01571$ |
| Group 3 | $0.847 \pm 0.01359$ | Group 3 (0.7) | $0.873 \pm 0.01241$ |
| Group 4 | $0.886 \pm 0.01212$ | Group 4 (0.8) | $0.931 \pm 0.00947$ |

Table 3: Group-wise accuracy of the data-driven hl-discrete CFE generator on `manual groups` & `probabilistic groups` (see Appendix B.3.5). While the accuracy on the `manual groups` was within the same range (87%), it greatly varied across the `probabilistic groups`.

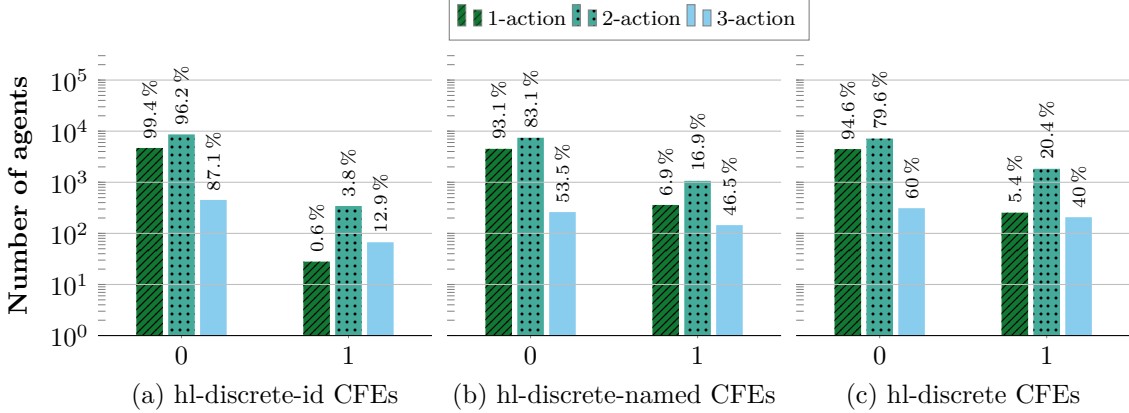

Figure 16: The data-driven hl-id CFE generator for the (**a**) hl-discrete-id CFEs, the data-driven hl-continuous CFE generator for the (**b**) hl-discrete-named CFEs, and the data-driven hl-discrete CFE generator for the (**c**) hl-discrete CFEs, achieved strong performance on the 20-dimensional `all` agent–hl-discrete CFE, varied information access, test datasets (new agents for the respective variants).

### Performance of the data-driven CFE generators

|  | all | >10 | >40 |
|---|---|---|---|
| 20-dimensional | $0.969 \pm 0.00284$ | $0.984 \pm 0.00208$ | $0.993 \pm 0.00141$ |
| 50-dimensional | $0.744 \pm 0.00608$ | $0.838 \pm 0.00534$ | $0.915 \pm 0.00458$ |
| 100-dimensional | $0.354 \pm 0.00664$ | $0.630 \pm 0.00778$ | $0.856 \pm 0.00772$ |

Table 4: A comparative analysis of the performance of the data-driven hl-discrete CFE generator across agent–hl-discrete CFE datasets with varied dimensions (20-, 50- and 100-dimensional) and varied frequency of the CFEs (`all`, `>10`, and `>40`). Results indicate that accuracy declines as dimensionality increases and CFE frequency decreases. Notably, the 20-dimensional `>40` dataset, which has the lowest dimensionality and highest CFE frequency, achieved the highest accuracy.

Since the agents in the `manual groups` agent–hl-discrete CFE datasets had more balanced access to hl-discrete actions as depicted in Figure 5(a), the data-driven hl-discrete CFE generators had almost similar accuracy (∼87%) in the generation of CFEs across all agents in different `manual groups`, as shown in Table 3 (**left**). On the other hand, since the agents in the `probabilistic groups` had access to varied hl-discrete actions, the accuracy of the data-driven hl-discrete CFE generator varied greatly across the groups, as shown in Table 3 (**right**). For instance, as expected, the CFEs for `probabilistic groups` Group 4 agents with one-action hl-discrete CFEs were more accurately generated with an accuracy of 93.06% as compared to Group 0 and Group 1 agents, generated at an accuracy of 88.04% and 77.09%, respectively.

### E.3 Data-driven Generators are Accurate, Confident and Approximate when needed

Our results show that the data-driven CFE generators are accurate and confident information-specific CFE generators. Additionally, unlike low-level CFE generators that sometimes fail to produce a CFE entirely for an agent, our data-driven CFE generators generate approximately good CFEs instead of no CFEs at all. The supplemental results in this appendix subsection are mainly for the fully-synthetic datasets.

### E.3.1 Accuracy and Confidence

The proposed data-driven CFE generators are evidenced to perform strongly on the varied datasets. As shown in Figure 16, on the 20-dimensional `all` agent–CFE dataset variants, the CFE generators achieved high accuracy at generating hl-discrete CFEs, hl-discrete-id CFEs, and hl-discrete-id CFEs. All the generators

perform best on the single-action CFE agents. Furthermore, with strong confidence, i.e., low margin error rates (see Table 4), the proposed data-driven CFE generators performed well on all datasets regardless of the data dimension or frequency of CFEs. Notably, they excelled on high-frequency datasets, that is to say, >40 datasets regardless of the data dimensions, as seen in Table 4.

### E.3.2 Approximation

Unlike ILP-based low-level CFE generators, which do not generate CFEs for agents when the ILP solution is sub-optimal or infeasible, our data-driven CFE generators alternatively produce valid CFE mistakes when suboptimal (see Figure 17). For example, of the $1.58\%, 16.23\%$ and $37.00\%$ mistakes the hl-id generator makes on the 20-, 50-, and 100-dimensional >10 agent–hl-discrete-id CFE datasets, $100\%, 99.23\%$, and $87.29\%$, respectively, were valid CFE mistakes. Similarly, the majority of the mistakes of the hl-discrete CFE generators were valid, e.g., on the 20-dimensional >10 agent–hl-discrete CFE dataset, of the $10.8\%$ mistakes the generator makes, $63.10\%$ were valid.

Additionally, the likelihood of the ILP-based low-level CFE generator's failure at generating CFEs (i.e., returns no CFEs) increases with the number of actionable features (data dimensions). On the hand, the percentage of valid mistakes from our proposed CFE generators decreases with the frequency of CFEs in the agent–CFE training set, e.g., the percentage of valid mistakes is $87.29\%$ on the >10 dataset and $57.83\%$ on the 100-dimensional all dataset.

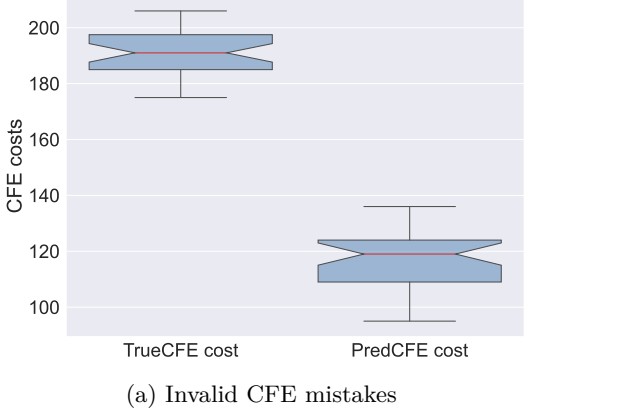
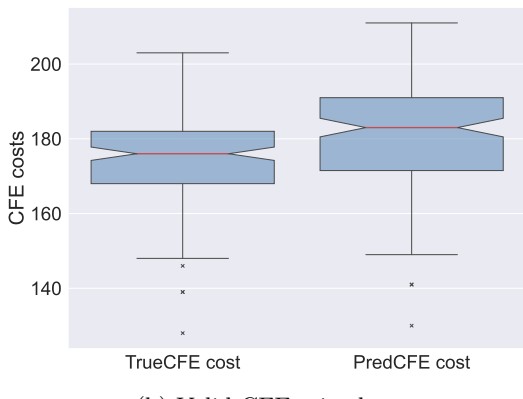

(a) Invalid CFE mistakes         (b) Valid CFE mistakes

Figure 17: A generated CFE is a mistake if the CFE doesn't match the true CFE. A valid CFE mistake transforms the agent's initial state to get a desirable model outcome. An invalid CFE mistake does not favorably transform the agent state. Distribution of costs of generated and true CFEs for (a) invalid and (b) valid CFE mistakes the data-driven hl-id CFE generator makes on 20-dimensional all agent–hl-discrete-id dataset. Valid CFE mistakes are, by definition, more expensive than the true CFEs, while invalid CFE mistakes are cheaper than the true CFEs.

### E.4 Data-driven CFE Generators are Easier to Scale and the CFEs are more Interpretable

Our results show that data-driven CFE generators, including hl-continuous, hl-discrete, and hl-id, are more scalable than low-level CFE generators. In addition, the costs and actions of hl-continuous and hl-discrete CFEs are interpretable and transparent, simplifying validation and comparison.

### E.4.1 Scalability

Unlike the overly specific feature-based actions in low-level CFEs (see Figure 15(a)), actions in hl-continuous and hl-discrete CFEs are more general, increasing the likelihood of their optimality for agents with closely similar profiles. As shown in Figure 18, while low-level CFEs were typically unique to each agent, hl-continuous and hl-discrete CFEs were often simultaneously optimal for multiple agents (see also Figure 15).

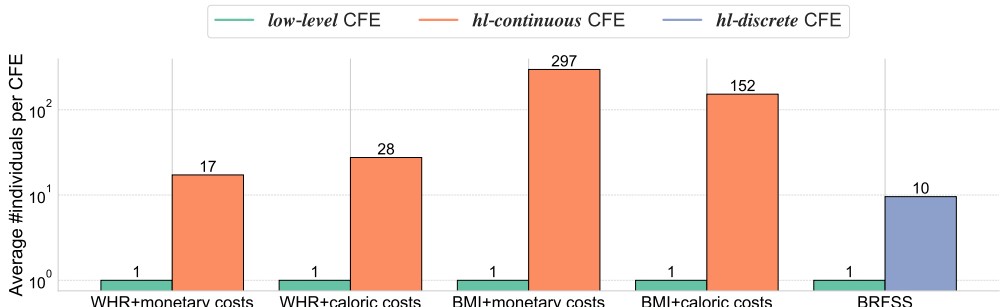

Figure 18: The average number of agents with the same CFE when agents take hl-continuous CFEs (as a set of Food+monetary and Foods+caloric costs hl-continuous actions) for WHR and BMI datasets, as well as hl-discrete CFEs on the BRFSS dataset, compared to when they take low-level CFEs for the respective datasets. Regardless of the dataset considered, on average, while low-level CFEs were unique to a given agent (individual), hl-continuous and hl-discrete CFEs were simultaneously optimal to multiple agents.

Furthermore, while our data-driven CFE generators efficiently produce CFEs for new agents without requiring re-optimization, low-level CFE generators operate on a single-agent basis, making them significantly more resource-intensive.

### E.4.2 Interpretability

The hl-continuous and hl-discrete CFEs comprise real-world-like actions (see Figures 15(b) and 15(c)), in contrast to low-level CFEs, which rely on overly specific, feature-based actions (see Figure 15(a)). Consequently, hl-continuous and hl-discrete CFEs are more intuitive, interpretable, and easier to execute and compare since they align more closely with executable actions. Moreover, the costs associated with these actions are more transparent and easier to comprehend, given a general understanding of how they were derived, an essential factor in ensuring clarity and trust in following the recommended CFE.

Looking ahead, conducting a comprehensive user study would be a valuable avenue for future work. The human-subject study could empirically validate the transparency and interpretability of the proposed high-level CFEs, serving as a test of the underlying hypotheses.

### E.5 Performance under various Information Access Constraints

In addition to other information access constraints, we investigate the effectiveness of the data-driven CFE generators under two more information access constraints. From the original agent–hl-discrete CFE datasets, we created two more information access variants, the agent–hl-discrete-named CFE dataset, and agent–hl-discrete-id CFE dataset as described in Appendix B.3.3. Given the agent–hl-discrete CFE information access datasets, we use the data-driven hl-discrete CFE generators for the hl-discrete CFEs, hl-continuous CFE generators for hl-discrete-named CFEs, and data-driven hl-id CFE generators for hl-discrete-id CFEs.

In general, all the data-driven CFE generators, regardless of information access constraints described in Appendix B.3.3, generate single-action CFEs more accurately than multi-action CFEs. For example, the hl-discrete CFE generator, as seen in Figure 16(**c**), generates one-action CFEs at an accuracy of 94.6%, two-action CFEs at an accuracy of 79.6%, and three-action CFEs at an accuracy of 60.0%.

However, in general, data-driven hl-id CFE generators were shown in Figure 16(**a**) to 16(**c**) and Table 5 to be more accurate and need less CFE frequency in the training set than the hl-continuous and hl-discrete CFE generators. For example, on the 20-dimensional `all` dataset, the data-driven hl-id CFE generator had an accuracy of 96.9%, compared to 85.4% with hl-continuous CFE generator and 83.9% with hl-discrete CFE generator.

| | Performance on 20-dimensional datasets | | |
|---|---|---|---|
| | all | >10 | >40 |
| Data-driven hl-id CFE generator | $0.969 \pm 0.00284$ | $0.984 \pm 0.00208$ | $0.993 \pm 0.00141$ |
| Data-driven hl-continuous CFE generator | $0.854 \pm 0.00581$ | $0.886 \pm 0.00531$ | $0.940 \pm 0.00411$ |
| Data-driven hl-discrete CFE generator | $0.839 \pm 0.00605$ | $0.892 \pm 0.00518$ | $0.937 \pm 0.00420$ |

Table 5: Accuracy of the CFE generators on 20-dimensional: `all`, `>10`, and `>40` datasets. All the data-driven CFE generators demonstrate consistently high accuracy across all datasets, with particularly strongest performance on high CFE frequency datasets (`>40`).

### E.6  Challenges and Proposed Solutions in Designing Data-driven CFE Generators

We identify several challenges in data-driven CFE generators: the infrequent occurrence of CFEs, the large number of actionable features, and the significant dependence on the complexity of the CFE generator model. In this work, we thoroughly examine these challenges, propose plausible solutions, and suggest avenues for future research to explore these issues in greater depth.

#### E.6.1  Negatively Affected by High Number of Actionable Features

As the number of actionable features increases, agents CFEs have more actions (see Table 2). For instance, in the 100-dimensional agent–hl-discrete CFE dataset, 54.4% of agents' hl-discrete CFEs had three actions, and in the 20-dimensional dataset, 33.3% of agents needed only one action in their CFE and only 3.6% had three.

Beyond an increase in the number of actions in the CFEs, the uniqueness of CFEs also rises as the number of actionable features grows. The average frequency of CFEs in the `all` agent–hl-discrete CFE training set dropped from 46.64% in the 20-dimensional dataset to 21.75% in the 50-dimensional dataset and further to 8.09% in the 100-dimensional dataset. Additionally, 18.115%, 20.797%, and 31.072% of CFEs in the 20-, 50-, and 100-dimensional `all` datasets, respectively, were unique (i.e., optimal for only one agent). This low frequency of CFEs in the agent–CFE datasets led to discrepancies after train/test splits, where some CFEs appeared in one split but not the other. Specifically, for the 20-, 50-, and 100-dimensional datasets, there were 52, 154, and 708 unique CFEs in the testing set that were absent from the training set.

As a result, data-driven CFE generators become less accurate as the number of actionable features increases. As shown in Table 4, the data-driven hl-id CFE generator consistently performed worse on higher-dimensional datasets. For example, while it achieved 96.9% accuracy on the 20-dimensional `all` dataset, its accuracy dropped to 74.4% on the 50-dimensional `all` dataset and was lowest on the 100-dimensional `all` dataset.

#### E.6.2  Negatively Affected by Low Frequency of CFEs

We created the varied frequency of CFEs agent–CFE datasets: `all`, `>10`, and `>40` (see Appendix B.3.2), to examine how CFE frequency in the agent–hl-discrete CFE dataset impacts the robustness of data-driven CFE generators. After the train/test split, the `>40` dataset ensured that at least 20 agents shared the same CFE in the training set. By definition, the `>40` dataset had the highest CFE frequency, while `all` had the lowest. This frequency also varied with data dimensionality, as illustrated in Appendix E.6.1.

The low frequency of CFEs in the agent–hl-discrete CFE training sets negatively impacted CFE generation across all datasets, regardless of data dimensionality. However, this effect became more pronounced as data dimensions increased. For instance, as shown in Table 4, the accuracy of CFE generators on the 20-dimensional dataset was highest when CFEs had a frequency of at least 20 in the training set (`>40`) and lowest on the `all` dataset, where some CFEs appeared in the test set but not in the training set. Specifically, the data-driven hl-id CFE generator achieved an accuracy of 99.3% on the 20-dimensional `>40` dataset, compared to 96.9% on the 20-dimensional `all` dataset. In contrast, CFE generation accuracy on the 20-dimensional dataset was significantly higher than on the 100-dimensional dataset. This difference

highlights that the negative impact of low CFE frequency in the training set becomes more severe as data dimensionality increases.

Additionally, the minimum frequency of CFEs required for a strong CFE generator increases with the number of actionable features. While the frequency of at least 20 in the training set ensured an accuracy of 99.3% of the CFE generator on the 20-dimensional dataset (see Table 4), a higher frequency is needed for the 50- and 100-dimensional datasets (see Table 4 and Figure 19).

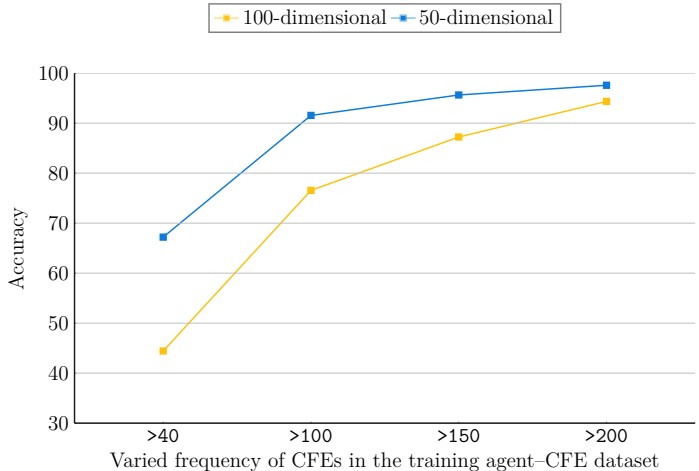

Figure 19: Impact of CFE frequency in agent–CFE training datasets for 50 and 100-dimensional datasets on the accuracy of the data-driven CFE generators. As the frequency of CFEs (the number of agents sharing the same optimal CFE) increases, the accuracy of data-driven CFE generators improves. This improvement occurs more rapidly in the lower-dimensional (50-dimensional) dataset.

**Data augmentation.** We examine the impact of increasing the frequency of CFEs through data augmentation (Algorithm 1) on the performance of the data-driven hl-discrete CFE generator. Algorithm 1 is specific for the agent–hl-discrete CFE datasets with all kinds of threshold classifiers. To generate new agents for which a given hl-discrete CFE is the most optimal, we ensure that no other hl-discrete CFE within the complete set of CFEs can achieve the transformation at a lower cost.

Therefore, given an agent state, we find all possible worse-off agent states, such that the current optimal hl-discrete CFE is still the best CFE for the worse-off agent states. Worse-off agent states are those such that the features where the hl-discrete CFE adds more capabilities than required to transform the agent state favorably are made worse, i.e., for $i$ such that $x_i^\star > t_i, aug_i < x_i$. Specific to the threshold classifier we use in the experiments, a hl-discrete CFE is adding more capabilities than required to feature $i$ of $\mathbf{x}$, if by after the action, the transformed feature $x_i^\star$ is such that $x_i^\star > t_i$. The derived worse-off agent state (augment) **aug** is valid if $\mathbf{x}$'s hl-discrete CFE is also the optimal CFE.

---

**Algorithm 1:** The agent–hl-discrete CFE dataset augmentation

---

**Input:** an agent $\mathbf{x}$ and their hl-discrete CFE $I$, and the threshold classifier $\mathbf{t}$
**Output:** valid derived augmentations of agent $\mathbf{x}$, $\mathbf{x}_{\text{augs}}$ with the same CFE
**Data:** indices of features $ids$ where the hl-discrete CFE when taken, adds more than needed
      capabilities to $\mathbf{x}$
augs $\leftarrow 2^{|ids|}$ possible worse-off agents;
**foreach aug** *in augs* **do**
    **if aug** *is valid* **then**
        $\mathbf{x}_{\text{augs}} \leftarrow \mathbf{x}_{\text{augs}} \cup \{\mathbf{aug}\}$;
    **end**
**end**

---

| | Effect of data augmentation | | |
| --- | --- | --- | --- |
| | 20-dimensional | 50-dimensional | 100-dimensional |
| Before data augmentation | $0.969 \pm 0.00284$ | $0.744 \pm 0.00608$ | $0.354 \pm 0.00664$ |
| After `AG1` | $0.965 \pm 0.00303$ | $0.760 \pm 0.00595$ | $0.505 \pm 0.00694$ |
| After `AG2` | $0.982 \pm 0.00218$ | $0.845 \pm 0.00504$ | $0.790 \pm 0.00565$ |

Table 6: Effects of `AG1` and `AG2` augmentation on the accuracy of the data-driven CFE Generator. Data augmentation mitigates the negative impact of low frequency of CFEs and improves the accuracy of data-driven CFE generators on the 20-, 50-, and 100-dimensional: `all` datasets.

**Data augmentation reduces negative impact of low frequency of CFEs.** Using Algorithm 1, we augment the agent–hl-discrete CFE training set to ensure that each CFE appears at least twice (`AG1`) and to increase the frequency of CFEs with fewer than 20 occurrences (`AG2`). As a result, the number of hl-discrete CFEs with fewer than 20 agents significantly decreased from 813 to 638, 2676 to 2005, and 9043 to 7144 for the 20- 50- and 100-dimensional datasets, respectively.

Experimental results show an improvement in the accuracy of the CFE generators on the test samples after data augmentation, e.g., on the 100-dimensional dataset, the accuracy of the data-driven CFE generator increases from 35.37% before data augmentation to 50.54 after `AG1`, and 78.99% after `AG2` (see Table 6). The findings show that data augmentation can improve the robustness of CFE generators in cases where the frequency of CFEs in the agent–CFE training datasets is low.

### E.7 Performance Heavily Depends on Complexity of CFE Generator Model

Given the agent–hl-discrete-id CFE 20-dimensional, `>40` dataset variant, we compare the effectiveness of the neural network-based CFE generator against the Hamming distance-based CFE generator. As shown in Figure 20, the neural network-based CFE generator demonstrates greater accuracy in generating CFEs for new agents. Interesting for future works is an exploration of the effectiveness of CFE generators based on more advanced and alternative methods, e.g., multi-chain neural networks, reinforcement learning, and transformer models.

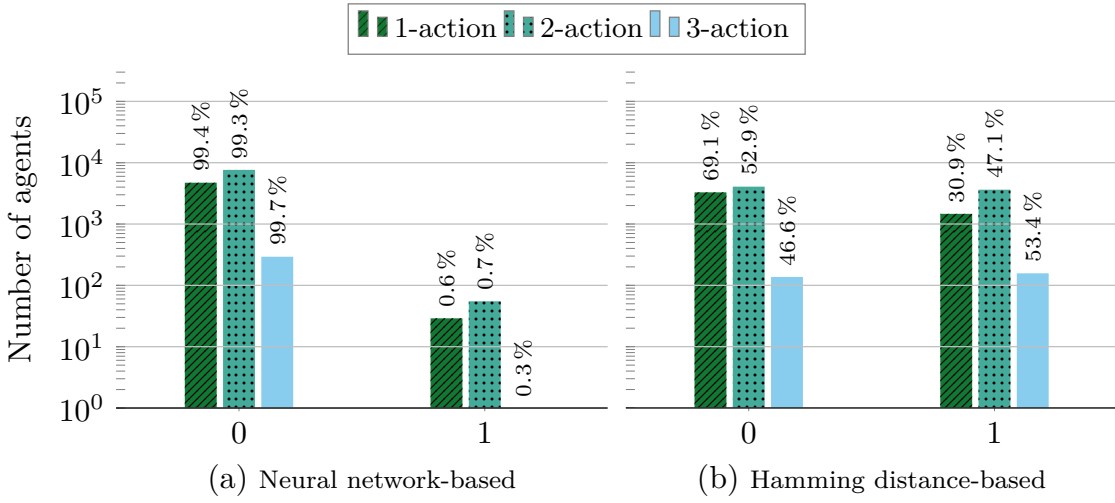

Figure 20: A comparison of accuracy of two data-driven hl-discrete CFE generators on the 20-dimensional `>40` dataset. The neural network model consistently performs better than the hamming distance model.

