# OpenReview forum: "Learning Actionable Counterfactual Explanations in Large State Spaces"
_TMLR — Accepted by TMLR_

### Review · Reviewer_hnCE · 2025-04-10

**Summary Of Contributions:**

The paper proposes new types of recourse relying on pre-defined actions that are supposed to improve actionability as well as other important aspects such as robustness, fairness, etc. The recourse is generated by a learned function that maps individuals (agents) to recourse actions.

With the paper addressing the issue of actionable recourse, I think its topic would be valued by the XAI community.
In particular, two points are interesting:
1. Replacing optimization-based recourse generators by a learned model. This has many benefits as outlined by the authors. However, it may be difficult to build in practice.
2. Highlighting the usefulness and potential of considering pre-defined action sets. In particular, the impact on robustness and fairness.

The contributions are interesting, yet I see a few limitations (please see the next sections).
Regarding the supportive evidence, I feel a bit mixed. On the one hand, extensive empirical evaluations support the claims, but on the other hand, the paper is limited to linear classifiers, and I also have some concerns about the comparison to existing methods (please see the next sections)

**Audience:**

Yes

**Claims And Evidence:**

Yes

**Requested Changes:**

I have three major concerns that need to be discussed and potentially addressed/changed:
1. It seems that the entire work focuses on linear classifiers. While linear classifiers do exist in the real world, other non-linear classifiers such as tree-based models and deep neural networks are very common as well. I think this limitation should be stated more clearly, and it should be stated how well the proposed methods and concepts could be extended to non-linear classifiers. Because of the empirical focus of the paper, just considering linear classifiers looks a bit weak to me.
2. The main benefit seems to come from more interpretable and actionable actions (with the side effect of being more robust, etc.). This is done by assuming knowledge of all actions. Is this really realistic? What if we only know about a subset of actions? Is recourse guaranteed for every individual/query?
3. I am a bit unsure how fair the comparison to existing generators is. The proposed methods rely on manually created actions. I am wondering if those could be made available to existing generators.

A few other comments and suggestions that I had while I was reading the paper:

Introduction:
- The GDPR is already "quite old". You may want to consider citing the more recent EU AI act (https://artificialintelligenceact.eu) -- I found this website to be very informative and helpful for navigating the AI act.
- Better explain "recourse". Personally, I am very familiar with counterfactuals and (computational) recourse, but I think the paper could be made more accessible by briefly explaining what is meant by "recourse" and how it relates to counterfactual explanations.
- Figure 1 looks very appealing; however, it seems that the proposed recourse generator comes with a lot of prior (domain) knowledge and assumptions (e.g., knowing about the Google AI residency program and how it affects the attributes). I think this should be made clear because it seems to be very domain-specific.

Background:
- Such crazy specific recommendations (e.g., "43.376000000000005") are unrealistic in practice. In practice, people would round numbers instead of reporting that many decimal places. I get the author's argument, but rounding is not the problem here.
- "Moreover, low-level CFE generation becomes very limited, sometimes impractical, in scenarios where access to critical information, such as sufficient training data, classifier parameters, or a comprehensive list of actions and their costs, is restricted or inaccessible." <- this statement is a bit provocative and needs references. While those aspects can be an issue for some generators, there also exist generators that do not need any training data or access to model internal parameters. I recommend being a bit more specific here and less general.
- "Lastly, low-level CFE generators operate on a single-agent basis" this is not completly true since group counterfactuals and multi-instance counterfactual exists, which are valid for more than a single individual. For instance, Kanamori et al. (2022) proposed counterfactual explanation trees, which (after they have been learned) can be applied to any given instance.

Section 3:
- Eq. 2: You assume a linear binary classifier, while Eq. does not make this assumption. Please state this clearly in the beginning; right now it is a bit hidden at the bottom.
- The same for Eq. 3. But here the classifier itself is a bit unusual -- can you give examples of such a threshold classifier?
- Are Eq. 2 and 3 really novel? To me looks like applying an action vector to a given input, which is quite common in counterfactual reasoning. Distinguishing between discrete and continuous domains does not seem that novel to me. Can you please clarify?

Section 4:
- How does it relate to Section 3. To me it looks like that you "just" learn a generator for computing counterfactuals. Besides assuming a training data set of counterfactuals, it remains unclear if you can state any guarantees on the computed actions. How do you ensure feasibility? Are you still restricted to linear classifiers? Shouldn't this be possible for arbitrary classifier types?

Experiments:
- Why only a single train-test split? I recommend to use smth. like cross validation to get statistically meaningful results.
- Eq. 6: I am unsure how reasonable it is to compare "the ground truth counterfactual". Since counterfactuals are known to be not unique, I was wondering why this is not considered here. Intuitively, I would just check for the correctness (i.e., change in prediction) and maybe some other quantities. Can you please clarify this?

Limitations:
- Only linear classifiers are considered in this work. As already stated, I consider this a major limitation that should be discussed as well.

Minor:
- GitHub repository: Please add a REQUIREMENTS.txt listing all needed packages -- the Prerequisites listed in the README are not sufficient
- Appendix: Add line numbers to Algorithm 1
- Please carefully check the references. For instance, "Sandra Wachter, Brent D. Mittelstadt, and Chris Russell. Counterfactual Explanations without Opening the Black Box: Automated Decisions and the GDPR." has been published in a journal which should be cited instead of the arXiv preprint.

**Strengths And Weaknesses:**

The paper addresses an important problem (actionable resource that is robust, fair, etc.) and performs extensive empirical evaluations.
However, in my opinion, it suffers from a few limitations:
- Only linear classifiers are considered
- Knowledge of pre-defined actions is assumed. It remains unclear to me how fair the comparison is to methods that do not have access to those pre-defined actions.

However, it could also be the case that I did not understand this correctly. Happy to discuss this. Please see next section for a more detailed review.

---

> ### Comment · Action_Editor_Y3oG · 2025-04-10
> **Thank you**
>
> Hi Reviewer hnCE,
>
> Thank you for getting this review completed in a timely manner. I believe that you have provided a wonderful set of possible improvements for the authors to consider as well as offered many points of discussion to start an informative dialogue.
>
> One aspect of your review that I think could stand to be expanded is by providing more context surrounding the contributions of the work. Do you find that these contributions and the supporting evidence to be compelling? Would members of the ML community working in related areas appreciate this work?
>
> If you could give concrete feedback about what the paper does well along these questions, that would also be helpful to the authors so they have a strong basis of understanding what resonated from the paper, as currently written.

---

> > ### Comment · Reviewer_hnCE · 2025-04-10
> >
> > Sure, just added a short paragraph to the summary of my review.

---

> > > ### Comment · Action_Editor_Y3oG · 2025-04-10
> > >
> > > I appreciate you providing additional context. This is great.

---

> > > > ### Author Response · Authors · 2025-04-15
> > > > **Response to the reviewer**
> > > >
> > > > Thank you, Editor Y3oG and Reviewer hnCE, for handling our work.
> > > > We thank the reviewer for their prompt, thorough evaluation of our work and the thoughtful, detailed, and constructive feedback provided. Below, we address each of the requested changes. We have uploaded an edited manuscript with revisions (in blue) in response to the reviewer’s suggestions. To keep this response concise, we generally refrain from repeating the revised text in full here and instead indicate where we made changes. We are happy to clarify any aspect of our response that may remain unclear.
> > > >
> > > >
> > > > > “It seems that the entire work focuses on linear classifiers.”
> > > > > “Limitations: Only linear classifiers are considered in this work. As already stated, I consider this a major limitation that should be discussed as well.”
> > > >
> > > > Linear classifiers are primarily employed in Equations (1) and (2) for generating low-level CFEs and single-agent hl-continuous CFEs, respectively. In contrast, single-agent hl-discrete CFE generation relies on feature thresholds, which can be part of varied classifier types.\
> > > > Meanwhile, the data-driven CFE generators introduced in Section 4 leverage neural networks with diverse architectures. These generators are agnostic to the underlying classifiers and rely mainly on agent-CFE datasets.\
> > > > Thus, aside from classification tasks involving the BMI and WHR datasets and the single-agent low-level and hl-continuous CFE generation, the remaining parts of our work utilize a broader range of model types.
> > > >
> > > > We would also like to note that although the linear models may not always be optimal, they can outperform non-linear models in some contexts. When non-linear classifiers are preferred, one practical alternative is to approximate them by linear models, enabling CFE generation with the proposed single-agent hl-continuous CFE generator.  Extending this generator to produce exact CFEs for non-linear models in a scalable and computationally efficient manner remains a challenging but promising avenue for future research.
> > > >
> > > > We have updated several parts of the limitations section, and the second paragraph, in particular, addresses the reviewer’s feedback.
> > > >
> > > >
> > > > > “The main benefit seems to come from more interpretable and actionable actions (with the side effect of being more robust, etc.). This is done by assuming knowledge of all actions. Is this realistic? What if we only know about a subset of actions? Is recourse guaranteed for every individual/query?”
> > > >
> > > > While equations in Section 3 (single-agent CFE generation) require access to the actions, their effects on features, and associated costs, the data-driven CFE generators described in Section 4 do not. Moreover, the proposed data-driven CFE generators (Section 4) help address several challenges related to limited information access. For example, having an exhaustive action list may not be essential for effective CFE generation.
> > > >
> > > > Although reliance on explicit action information in Section 3 is a limitation, the new formulations reduce the dependency on the classifier’s training data to determine the action set and corresponding costs. As noted by Ustun et al. (2019), relying on training data may not accurately reflect the difficulty of recourse, especially when the data lacks sufficient representation from the target population or fails to capture its distribution.
> > > >
> > > > We also observed that when the action space is big,  the low-level CFE generator more frequently fails to produce feasible CFEs compared to the single-agent hl-continuous and hl-discrete CFE  generators.
> > > > Additionally, we observed that when the data-driven CFE generator didn’t generate the correct CFE (least cost CFE that favorably flips the model outcome), it often generated a valid (favorably flips the model outcome) but more costly CFE.
> > > >
> > > >
> > > > > “I am a bit unsure how fair the comparison to existing generators is. The proposed methods rely on manually created actions. I am wondering if those could be made available to existing generators.”
> > > >
> > > > Reconceptualizing CFEs from feature-based actions and costs to more real-world-like actions and costs offers significant advantages, as demonstrated in Section 6 and Appendix D.\
> > > > To the best of our knowledge, this reformulation is novel and valuable for the recourse and CFE generation community, as it directly addresses long-standing challenges. For instance, Ustun et al. (2019) point out that percentile-based cost functions often fail to capture the difficulty of recourse. Similarly, one of the highlighted key hidden assumptions behind CFEs by Barocas et al.  is that "features do not clearly map to actions."
> > > >
> > > > While some existing CFE generators might be plug-and-play adapted to support the proposed higher-level actions, we do not consider this an unfair comparison. Instead, we see it as an exciting future research direction: to explore which pre-existing CFE generation approaches can be more seamlessly adapted to this new paradigm and identify the challenges that may emerge.

---

> > > > > ### Author Response · Authors · 2025-04-15
> > > > > **Response to the reviewer(2)**
> > > > >
> > > > > > “Introduction:  The GDPR is already "quite old", Better explain "recourse".  I think this should be made clear because it seems to be very domain-specific.”
> > > > >
> > > > > We have updated the first paragraph of the introduction to reflect this feedback, and added a note on domain-specificity.
> > > > >
> > > > > > “Background: I get the author's argument, but rounding is not the problem here. I recommend being a bit more specific here and less general. This is not completely true since group counterfactuals and multi-instance counterfactuals exist.”
> > > > >
> > > > > Thank you for pointing us to this related work. We have updated the last paragraph of the background and the second paragraph of related works sections to reflect the reviewer's feedback.
> > > > >
> > > > >
> > > > > > “Section 3: Please state this clearly in the beginning; right now it is a bit hidden at the bottom. Also, can you give examples of such a threshold classifier? Are Eq. 2 and 3 really novel? To me it looks like applying an action vector to a given input, which is quite common in counterfactual reasoning. Distinguishing between discrete and continuous domains does not seem that novel to me. Can you please clarify?”
> > > > >
> > > > > We have revised Sections 3.1 and 3.2 to explicitly reference the underlying models used to assess whether an outcome is desirable.
> > > > >
> > > > > Feature thresholds can be the last form of any classifier type. For instance, consider the feature healthBMI in *App2, Figure 1*, where based on a given classification model outcome, a specific threshold to qualify for healthy or unhealthy BMI is used to determine feature eligibility ({0,1}).\
> > > > > The key distinction between hl-discrete and hl-continuous CFEs is how they affect features, where the former caters to feature eligibility and the latter numerically modifies feature values.
> > > > > Using the same example of the feature healthBMI, while hl-discrete CFE would ensure BMI is past a desired threshold, thus ensuring healthBMI = 1, the hl-continuous CFE would affect the actual feature values, such as reducing BMI from 30 to 22.3.
> > > > > These two forms of CFEs are suited for different contexts. The hl-discrete CFEs are particularly effective in rule-based systems, such as eligibility checks, wellness evaluations, or quality and safety assessments. In contrast, hl-continuous CFEs are better suited for scenarios where fine-grained numerical adjustments are meaningful, typically in environments where low-level CFEs are effective.
> > > > > While it might be easier to go from hl-continuous CFEs to hl-discrete CFEs, the reverse is less trivial and may require additional considerations.
> > > > >
> > > > > Although applying an action vector to an input is not new in the CFE generation literature, incorporating real-world-like actions into the formulation while ensuring the generated CFEs are valid and plausible is a useful and novel contribution to the CFE/recourse community.
> > > > >
> > > > > > “Section 4: How does it relate to Section 3. To me it looks like that you "just" learn a generator for computing counterfactuals. Besides assuming a training data set of counterfactuals, it remains unclear if you can state any guarantees on the computed actions. How do you ensure feasibility? Are you still restricted to linear classifiers? Shouldn't this be possible for arbitrary classifier types?”
> > > > >
> > > > > Section 4 is conceptually distinct from Section 3. While Section 3 presents a method for generating high-level CFEs using a single-agent approach, Section 4 tackles a different question: Can we, from agent-CFE data (instances of agents and their optimal CFEs), learn a CFE generator that accurately generates CFEs for new agents?
> > > > > Once learned, the data-driven CFE generator generates optimal CFEs that are the least cost and favorably flip the model outcome.
> > > > > Importantly, these learned data-driven CFE generators are model-agnostic and do not require access to the underlying classification model. Instead, they rely solely on the agent-CFE examples to learn how to generate correct CFEs (least cost and favorably flips the model outcome) for future agents.

---

> > > > > > ### Author Response · Authors · 2025-04-15
> > > > > > **Response to reviewer(3)**
> > > > > >
> > > > > > > “Experiments: Why only a single train-test split? I recommend using something like cross-validation to get statistically meaningful results. Eq. 6: I am unsure how reasonable it is to compare "the ground truth counterfactual". Since counterfactuals are known to be not unique, I was wondering why this is not considered here. Intuitively, I would just check for the correctness (i.e., change in prediction) and maybe some other quantities. Can you please clarify this?”
> > > > > >
> > > > > > Although we do not explicitly create a separate validation set during the initial data split (only training and testing), we use "validation_split" when training the generator models and apply "GridSearchCV" to hyperparameter tune the classification models.
> > > > > > Additionally, we perform statistical evaluations (Appendix C.2) to establish the statistical significance of the observed patterns (Section 6.1, Appendix Figure 11).
> > > > > >
> > > > > > Since the proposed data-driven CFE generators are supervised models, one way to evaluate their performance is by assessing the correctness of the generated CFEs. A generated CFE is *correct* if it matches the *true* CFE, defined as the one that incurs the least cost while favorably flipping the model’s prediction.\
> > > > > > In Appendix D.3.2, we further explore cases where the generated CFE may be *valid*, that is, flips the model outcome favorably but not *optimal*, meaning it does not represent the minimum-cost solution.\
> > > > > > Therefore, while our evaluation framework may be strict, it helps us validate if the data-driven generated CFE is similar to one that would be instance-level or local CFE generated.\
> > > > > > We would also like to note that one of the strengths of our data-driven approach is the reliance on the similarity between agent profiles to generate CFEs and the removal of the need to have query access to the classifier.
> > > > > >
> > > > > >
> > > > > >
> > > > > > > “Minor: Please add a REQUIREMENTS.txt listing all needed packages -- the Prerequisites listed in the README are not sufficient. Appendix: Add line numbers to Algorithm 1. Please carefully check the references” arxiv vs peer-review venue.
> > > > > >
> > > > > > Thank you for the valuable feedback. We have updated the GitHub repository to include the requirements.txt file. Additionally, we have revised the manuscript by adding line numbers to the algorithm and improving the formatting of the references.

---

> > > > > > > ### Comment · Reviewer_hnCE · 2025-04-15
> > > > > > >
> > > > > > > >Although we do not explicitly create a separate validation set during the initial data split (only training and testing), we use >"validation_split" when training the generator models and apply "GridSearchCV" to hyperparameter tune the classification >models. Additionally, we perform statistical evaluations (Appendix C.2) to establish the statistical significance of the observed >patterns (Section 6.1, Appendix Figure 11).
> > > > > > >
> > > > > > > I think my point is that I am always sceptical when somebody shows me a bar plot of averages and claims that from this we see that method A is different from method B (e.g., Figure 3). The average is quite sensitive to outliers and, therefore, inspecting higher moments, such as the variance, to get a better and more reliable impression of what is going on.
> > > > > > > While you do provide additional evaluations in the appendix, including variability of quantities (mainly for the fairness though), I am still wondering if there are any for those in Figure 3.
> > > > > > > Did I miss smth.? Can you argue against showing variance or other higher-order moments?
> > > > > > >
> > > > > > > >Since the proposed data-driven CFE generators are supervised models, one way to evaluate their performance is by >assessing the correctness of the generated CFEs. A generated CFE is correct if it matches the true CFE, defined as the one >that incurs the least cost while favorably flipping the model’s prediction.
> > > > > > > >In Appendix D.3.2, we further explore cases where the generated CFE may be valid, that is, flips the model outcome favorably >but not optimal, meaning it does not represent the minimum-cost solution.
> > > > > > > >Therefore, while our evaluation framework may be strict, it helps us validate if the data-driven generated CFE is similar to one >that would be instance-level or local CFE generated.
> > > > > > > >We would also like to note that one of the strengths of our data-driven approach is the reliance on the similarity between agent >profiles to generate CFEs and the removal of the need to have query access to the classifier.
> > > > > > >
> > > > > > > I see. Another question that comes to my mind is uniqueness. I think your point is valid if optimal counterfactuals are unique. Now, in general, as you may know, counterfactuals are often not unique -- even if they are unique w.r.t. to the minimum cost, there might exist different counterfactuals with the same minimum cost.
> > > > > > > Are the optimal counterfactuals in your experiments unique?
> > > > > > >
> > > > > > >
> > > > > > > >Thank you for the valuable feedback. We have updated the GitHub repository to include the requirements.txt file. Additionally, >we have revised the manuscript by adding line numbers to the algorithm and improving the formatting of the references.
> > > > > > >
> > > > > > > Looks better, I just spotted one error: Susanne Dandl, ... "\textunderscore31" in DOI

---

> > > > > > ### Comment · Reviewer_hnCE · 2025-04-15
> > > > > >
> > > > > > >We have updated the first paragraph of the introduction to reflect this feedback, and added a note on domain-specificity.
> > > > > >
> > > > > > Thanks! Looks much better now.
> > > > > >
> > > > > > >Thank you for pointing us to this related work. We have updated the last paragraph of the background and the second paragraph of related works sections to reflect the reviewer's feedback.
> > > > > >
> > > > > > Thanks! Looks much better now.
> > > > > >
> > > > > > >We have revised Sections 3.1 and 3.2 to explicitly reference the underlying models used to assess whether an outcome is desirable. ...........
> > > > > >
> > > > > > Thanks for clarifying this. I do not know if it just me, but maybe adding this example and clarification to the paper would help other readers as well. Maybe better to wait for opinions from the other reviewers.
> > > > > >
> > > > > > >Section 4 is conceptually distinct from Section 3. While Section 3 presents a method for generating high-level CFEs using a single-agent approach, Section 4 tackles a different question: Can we, from agent-CFE data (instances of agents and their optimal CFEs), learn a CFE generator that accurately generates CFEs for new agents? Once learned, the data-driven CFE generator generates optimal CFEs that are the least cost and favorably flip the model outcome. Importantly, these learned data-driven CFE generators are model-agnostic and do not require access to the underlying classification model. Instead, they rely solely on the agent-CFE examples to learn how to generate correct CFEs (least cost and favorably flips the model outcome) for future agents.
> > > > > >
> > > > > > I think the generator is *supposed* to compute optimal actions, but can you guarantee this? I think not. Another issue might be missing uniqueness, similar to my other reply regarding the evaluation of the computed counterfactuals.
> > > > > > To me, it feels like we have to discuss this uniqueness issue and then figure out how you can rephrase the statements in your paper. Let me know what you think.

---

> > > > > ### Comment · Reviewer_hnCE · 2025-04-15
> > > > >
> > > > > >Meanwhile, the data-driven CFE generators introduced in Section 4 leverage neural networks with diverse architectures. These generators are agnostic to the underlying classifiers and rely mainly on agent-CFE datasets.
> > > > >
> > > > > Right, but are you evaluating your proposed method for linear models only? -- because you can compute optimal counterfactuals using MILPs and then compare to them? I am struggling a bit with seeing this in the paper. Can you please clarify?
> > > > >
> > > > > >We would also like to note that although the linear models may not always be optimal, they can outperform non-linear models in some contexts. When non-linear classifiers are preferred, one practical alternative is to approximate them by linear models, enabling CFE generation with the proposed single-agent hl-continuous CFE generator. Extending this generator to produce exact CFEs for non-linear models in a scalable and computationally efficient manner remains a challenging but promising avenue for future research.
> > > > >
> > > > > I know xD No need to worry about that. I think my point is that this does not become clear from the title or abstract, only when reading the paper. So it might be necessary to change the title/abstract in that regard. But first let's wait what the other reviewers think about this.
> > > > >
> > > > > >We have updated several parts of the limitations section, and the second paragraph, in particular, addresses the reviewer’s feedback.
> > > > >
> > > > > Thanks! Reads much better now.
> > > > >
> > > > > >Although reliance on explicit action information in Section 3 is a limitation, the new formulations reduce the dependency on the classifier’s training data to determine the action set and corresponding costs. As noted by Ustun et al. (2019), relying on training data may not accurately reflect the difficulty of recourse, especially when the data lacks sufficient representation from the target population or fails to capture its distribution.
> > > > > >
> > > > > >We also observed that when the action space is big, the low-level CFE generator more frequently fails to produce feasible CFEs compared to the single-agent hl-continuous and hl-discrete CFE generators. Additionally, we observed that when the data-driven CFE generator didn’t generate the correct CFE (least cost CFE that favorably flips the model outcome), it often generated a valid (favorably flips the model outcome) but more costly CFE.
> > > > >
> > > > > I think there are always limitations, no matter what is assumed. Fine with me as long as it is discussed properly.
> > > > >
> > > > >
> > > > > >Reconceptualizing CFEs from feature-based actions and costs to more real-world-like actions and costs offers significant advantages, as demonstrated in Section 6 and Appendix D.
> > > > > To the best of our knowledge, this reformulation is novel and valuable for the recourse and CFE generation community, as it directly addresses long-standing challenges. For instance, Ustun et al. (2019) point out that percentile-based cost functions often fail to capture the difficulty of recourse. Similarly, one of the highlighted key hidden assumptions behind CFEs by Barocas et al. is that "features do not clearly map to actions."
> > > > > >
> > > > > >While some existing CFE generators might be plug-and-play adapted to support the proposed higher-level actions, we do not consider this an unfair comparison. Instead, we see it as an exciting future research direction: to explore which pre-existing CFE generation approaches can be more seamlessly adapted to this new paradigm and identify the challenges that may emerge.
> > > > >
> > > > > I agree that features do not clearly map to actions, but I do not understand if it is not possible to make those sets of possible actions available to existing methods? For instance, an evolutionary algorithm should be able to handle those, or not? I probably did not fully understand your reply -- can you please clarify?

---

> > > > > > ### Author Response · Authors · 2025-04-16
> > > > > > **Response to reviewer(1)**
> > > > > >
> > > > > > Thank you for the response and the thoughtful and constructive feedback!
> > > > > >
> > > > > > > I think my point is that I am always sceptical when somebody shows me a bar plot of averages and claims that from this we see that method A is different from method B (e.g., Figure 3). The average is quite sensitive to outliers and,....
> > > > > >
> > > > > > We supplement Figure 3 with Figure 10 and Figure 11 (new) in the appendix. Figure 10 presents the density of the raw distribution of the difference in the number of modified features and agent improvement achieved. Figure 11 (new) compares the difference in the number of modified features and the agent improvement when each negatively classified WHR agent takes a low-level CFE (P) versus a high-level continuous CFE (Q). A look at the density plots and the raw values plots reveal patterns consistent with those observed in Figure 3.
> > > > > >
> > > > > > >Are the optimal counterfactuals in your experiments unique? .. To me, it feels like we have to discuss this uniqueness issue and then figure out how you can rephrase the statements in your paper. Let me know what you think.
> > > > > >
> > > > > > In the agent-CFE datasets, each agent is associated with the least-cost CFE that successfully flips the model’s prediction. However, the dataset also contains other valid CFEs (leading to a favorable prediction change) but at a higher cost, as evidenced by Figure 17 .
> > > > > > This highlights that while multiple valid CFEs may exist, the optimal CFE in terms of minimum cost is unique.
> > > > > >
> > > > > > For the hl-continuous and hl-discrete CFEs, which are represented as sets of hl-continuous and hl-discrete actions, respectively, we verify the correctness of the data-driven generated action sets (CFE) by ensuring not only that the generated CFE (set of actions) lead to a prediction flip, but also that they correspond to the ground truth least-cost CFE. This dual validation ensures that the generated CFEs are both valid and optimal.
> > > > > >
> > > > > > Finally, since the proposed data-driven CFE generators are capable of identifying the optimal action set corresponding to the least-cost CFE, a natural and promising extension would be to adapt the approach to generate a set of valid CFEs rather than a set of least-cost actions, an avenue future work can actively explore.
> > > > > >
> > > > > > > Looks better, I just spotted one error: Susanne Dandl, ... "\textunderscore31" in DOI
> > > > > >
> > > > > > Thank you! We have fixed this error in the revised manuscript
> > > > > >
> > > > > > > Thanks for clarifying this. I do not know if it is just me, but maybe adding this example and clarification to the paper would help other readers as well….
> > > > > >
> > > > > > We have added the example to Appendix A to supplement Figure 1.
> > > > > >
> > > > > > > Right, but are you evaluating your proposed method for linear models only? -- because you can compute optimal counterfactuals using MILPs and then compare them? I am struggling a bit with seeing this in the paper. Can you please clarify?
> > > > > >
> > > > > > For the fully synthetic agent-CFE datasets, we synthetically generated the agent states and actions and used ILP to generate the CFEs used in our empirical evaluations (see Section 5.3 and Appendices B.3, E.3, and E.4). In this case, as well as for the semi-synthetic hl-discrete CFEs, the evaluation is not restricted to linear models. Only the low-level CFEs and semi-synthetic hl-continuous CFEs rely on linear classification models.
> > > > > >
> > > > > > > So it might be necessary to change the title/abstract in that regard. …
> > > > > >
> > > > > > Thank you, we are open to revisiting this
> > > > > >
> > > > > > >..but I do not understand if it is not possible to make those sets of possible actions available to existing methods? For instance, should an evolutionary algorithm be able to handle those, or not? I probably did not fully understand your reply -- can you please clarify?
> > > > > >
> > > > > > We think that while it may be possible to make the proposed sets of actions available to existing CFE generation methods, at this stage, we're not entirely certain which pre-existing methods would be most suitable, how seamlessly they could be adapted, or what challenges might arise. For instance, for evolutionary algorithms that you suggest, generating an exact CFE with the new paradigm could be computationally expensive,  especially in large action spaces, potentially resulting in only approximate or Pareto-optimal solutions. We hope future research explores how to effectively adapt existing CFE generation methods to this new paradigm and uncover any associated interesting challenges.

---

> > > > > > > ### Comment · Reviewer_hnCE · 2025-04-16
> > > > > > >
> > > > > > > Thanks for your quick reply.
> > > > > > >
> > > > > > > >We have added the example to Appendix A to supplement Figure 1.
> > > > > > >
> > > > > > > Thanks.
> > > > > > >
> > > > > > > >We supplement Figure 3 with Figure 10 and Figure 11 (new) in the appendix.
> > > > > > >
> > > > > > > Thanks for the additional figure; this helps. In Figure 11 b), you seem to have a few outliers with a huge negative value, making it difficult to see the blue bars. Not sure how, but maybe think about a different y-scale. But this is not super important.
> > > > > > >
> > > > > > > > We think that while it may be possible to make the proposed sets of actions available to existing CFE generation methods, at this stage, we're not entirely certain which pre-existing methods would be most suitable, .....
> > > > > > >
> > > > > > > Yeah, I also do not know which method would be best suited. I was just thinking whether this would be a possibility. I think, at least for me, it would help to just elaborate on this aspect in the paper. But let's see what the other reviewers have to say.
> > > > > > >
> > > > > > > > For the fully synthetic agent-CFE datasets, we synthetically generated the agent states and actions and used ILP to generate the CFEs used in our empirical evaluations (see Section 5.3 and Appendices B.3, E.3, and E.4). In this case, as well as for the semi-synthetic hl-discrete CFEs, the evaluation is not restricted to linear models. Only the low-level CFEs and semi-synthetic hl-continuous CFEs rely on linear classification models.
> > > > > > >
> > > > > > > I see. Let's see what the other reviewers think. Maybe they agree and you should change title/abstract, or maybe it's just me who is a bit too picky here.
> > > > > > >
> > > > > > > >  However, the dataset also contains other valid CFEs (leading to a favorable prediction change) but at a higher cost, as evidenced by Figure 17
> > > > > > >
> > > > > > > Not sure if I understand this correctly. Are you plotting all valid counterfactuals or just the ones that you generate with the model? For binary features, there exists a finite number of counterfactuals, I agree on that. But for other feature types, there might exist infinitely many counterfactuals.
> > > > > > >
> > > > > > > > In the agent-CFE datasets, each agent is associated with the least-cost CFE that successfully flips the model’s prediction. .... the optimal CFE in terms of minimum cost is unique. ......
> > > > > > >
> > > > > > > Not sure if there is a misunderstanding. I agree that the counterfactuals are unique w.r.t. to the cost, but my question was whether it is possible that two different and valid counterfactuals have the same minimal cost. Do I understand your response correctly, that you say that this is not the case?
> > > > > > >
> > > > > > > > .... by ensuring not only that the generated CFE (set of actions) lead to a prediction flip, but also that they correspond to the ground truth least-cost CFE. .....
> > > > > > >
> > > > > > > I was thinking about my own question. I think my point is that in general, counterfactuals are non-unique, and therefore it does not make much sense for me to try to compute a specific counterfactual. To me, it would be more reasonable to check if the cost of the computed counterfactual is correct (i.e., flipping the classifier's prediction) and has minimal cost. Furthermore, I think uniqueness highly depends on the set of possible actions A(x) -- maybe it is impossible to make general statements without making any assumptions about A(x), although you often assume A(x) to include binary actions for each feature? What do you think? Am I missing smth. that you are already doing?

---

> ### Author Response · Authors · 2025-04-16
>
> Thank you!
>
> On uniqueness,
>
> We ensured that, in the agent-CFE dataset, there is one valid CFE with the lowest cost. While there is the only optimal CFE, other valid CFEs that successfully flip the prediction but incur the same, slightly higher costs may also exist. It could especially be the case where we generate a set of actions (either high-level continuous or discrete), as different combinations of actions can be valid yet more expensive.
>
> We check for both validity (flipping the prediction) and minimal cost.
> Figure 17 displays the CFEs (valid with minimal cost or valid with higher cost) generated by the CFE generator, not the full space of all possible valid CFEs. While Figure 17 focuses on hl-discrete actions (i.e., binary actions), we expect a similar pattern to emerge with hl-continuous actions.
>
> The number of unique valid CFEs, defined as distinct action sets with unique costs, depends on the agent's initial state, the comprehensive list of actions available, and the associated costs of those actions.
> For instance, an agent starting at state [1, 0, 1, 1] that needs to reach [1, 1, 1, 1]  to flip the prediction would have fewer unique CFEs than an agent starting from [0, 0, 0, 0] trying to reach the same state.
>
> Since the data-driven generators effectively produce the proposed CFEs, their use can extend to generating sets of valid CFEs rather than optimal action set. Instead of constructing agent-CFE datasets with a unique, valid, low-cost CFE, encode agent–validCFE datasets, where each validCFE represents a varied set of valid CFEs. Then, at evaluation, a generated CFE is checked to determine whether it flips the classifier prediction and exhibits minimal cost compared to other possible CFEs.

---

> > ### Comment · Reviewer_hnCE · 2025-04-17
> >
> > Thanks.
> >
> > > We ensured that, in the agent-CFE dataset, there is one valid CFE with the lowest cost.
> >
> > Sounds good, but how did you do this?
> >
> > > The number of unique valid CFEs, defined as distinct action sets  ....
> >
> > Agreed. Would you say that this could influence your proposed method? It might be interesting to see how your method behaves in such cases.
> >
> > > Since the data-driven generators effectively produce the proposed CFEs, their use can extend to generating sets of valid CFEs rather
> >
> > Okay. I agree and think it would be an interesting and relevant direction for future work, and definitely strengthen the proposed generator. Are you already discussing this in the paper? Did not find it right now, but also did not look too closely.

---

> > > ### Author Response · Authors · 2025-04-22
> > >
> > > Thank you!
> > >
> > > > Sounds good, but how did you do this?
> > >
> > > The cost associated with fulfilling the eligibility of each feature was unique, resulting in diverse overall hl-discrete action costs. Consequently, the likelihood of agents having multiple lowest cost-equivalent, valid CFEs was very low. In the rare instances where agents had non-unique optimal CFEs, i.e., different valid CFEs with identical minimal costs, we excluded those agents from the agent-CFE dataset.
> > >
> > > > Okay. I agree and think it would be an interesting and relevant direction for future work, and definitely strengthen the proposed generator. Are you already discussing this in the paper? Did not find it right now, but also did not look too closely.
> > >
> > > Thank you, we have added it in Appendix C.

---

> > > > ### Comment · Reviewer_hnCE · 2025-04-23
> > > >
> > > > >The cost associated with fulfilling the eligibility of each feature was unique, resulting in diverse overall hl-discrete action costs. Consequently, the likelihood of agents having multiple lowest cost-equivalent, valid CFEs was very low. In the rare instances where agents had non-unique optimal CFEs, i.e., different valid CFEs with identical minimal costs, we excluded those agents from the agent-CFE dataset.
> > > >
> > > > I see. Thanks for the clarification. I think this is also smth. that should go into the appendix.
> > > >
> > > > Other than that, I do not have any other questions.

---

> > > > > ### Author Response · Authors · 2025-04-23
> > > > >
> > > > > Thank you, we will include it in the final version

---

> > > > > > ### Comment · Reviewer_hnCE · 2025-05-07
> > > > > >
> > > > > > Coming back to one of my original concerns:
> > > > > >
> > > > > > >> Right, but are you evaluating your proposed method for linear models only? -- because you can compute optimal >>counterfactuals using MILPs and then compare them? I am struggling a bit with seeing this in the paper. Can you please clarify?
> > > > > > >
> > > > > > >For the fully synthetic agent-CFE datasets, we synthetically generated the agent states and actions and used ILP to generate the CFEs used in our empirical evaluations (see Section 5.3 and Appendices B.3, E.3, and E.4). In this case, as well as for the semi-synthetic hl-discrete CFEs, the evaluation is not restricted to linear models. Only the low-level CFEs and semi-synthetic hl-continuous CFEs rely on linear classification models.
> > > > > >
> > > > > > After reading the other reviews and given the fact that I have to make a final recommendation, I would like to request one last change. Can you please highlight the fact that parts of your contributions rely on linear models? I think a short statement in the abstract will be sufficient.
> > > > > > After that, I will be happy to recommend "accept".
> > > > > >
> > > > > > Thanks!

---

> > > > > > > ### Author Response · Authors · 2025-05-07
> > > > > > >
> > > > > > > Thank you, we have made the changes to the revised manuscript

---

> > > > > > > > ### Comment · Reviewer_hnCE · 2025-05-07
> > > > > > > >
> > > > > > > > Looks good. Thanks!

---

> > > > > > > > > ### Author Response · Authors · 2025-05-09
> > > > > > > > >
> > > > > > > > > Thank you, Reviewer hnCE!

---

### Review · Reviewer_Vkih · 2025-04-17

**Summary Of Contributions:**

This paper defines three high-level counterfactual explanation (CFE) or recourse methods, i.e. hl-continuous, hl-discrete, and hl-id, that provide recommendations for individuals or agents to modify their input features to achieve a desired outcome. Compared to low-level CFEs, the high-level CFEs are more actionable and interpretable, without requiring users to understand precise numerical adjustments. The authors formulate single-agent CFE generation as optimization problems and further propose data-driven generators that learn from historical agent–CFE pairs to rapidly produce recourses for new agents. Experimental results on multiple datasets demonstrate that the single-agent CFE generators require few actions while yielding greater improvements compared to low-level CFEs, and that the data-driven generators accurately replicate the optimal recourses computed by the single-agent generator while maintaining computational efficiency.

**Audience:**

Yes

**Claims And Evidence:**

Yes

**Requested Changes:**

The writing would benefit from greater clarity, as certain aspects of the problem challenges and the details of the proposed methods remain somewhat unclear.
- Could the authors explicitly discuss the difference between low-level and high-level CFEs? Figure 1 was helpful in comparing them through concrete examples, but it is not immediately clear why estimating high-level CFEs is challenging and needs specific techniques. Technically, is this because multiple actions are selected simultaneously in a high-level CFE?
- According to Figure 1 and the following descriptions, the ultimate goal is to provide a descriptive recommendation using natural language. However, these recommendations still need to be represented as numerical values when learning the recourse. How is a numerical action translated into a descriptive sentence?
- The definitions for single-agent hl-continuous actions and hl-discrete actions are not clearly presented. What are the range and dimensions of these actions? If I understand correctly, a hl-continuous action is $\in \mathbb{R}^{J}$ for some high-dimensional $J$, and a hl-discrete action is $\in \{0, 1\}^{J}$. Besides, what is the form of the cost and effect, also $\in \mathbb{R}$? Further, examples are usually not presented within the definition itself but serve as explanations that follow the definition.
- For the single-agent hl-continuous CFE generator in Section 3.1, what is the range of $v_j$? Further, it should be clearly stated that $v_j$ represent the change in the feature $x$. Additionally, it is better to clearly define "negatively classified agents" as the $x$ with $c^T x + b < 0$.
- For the single-agent hl-discrete CFE generator in Section 3.2, it is mentioned that each action is represented by a vector $v_j$, yet $v_j$ does not appear in equation (3). Furthermore, I assume that a hl-discrete action only operates on binary states $x$, and that a value of one indicates a better health state than a value of zero in $x$. This has not been explained explicitly.
- A discussion of the differences between the problem settings for hl-continuous and hl-discrete actions should be provided. For example, if I understand correctly, the definition of the state $x$, the action $v$, and the form of the classifier are all different.
- Why does equation (4) use a cross-entropy loss, while equation (5) is computed only over positive identifications? Additionally, in Section 4.2, why is the objective function defined as a mean-squared error, if—according to my understanding—this should still be treated as a classification problem?
- Could the authors explain why the coefficient of variation for the high-continuous CFE is much lower in Figure 3(b)? Specifically, which part of the proposed method (perhaps the constraint in equation (2)?) guarantees this lower variation?
- Section 6.1 discusses single-agent generators, but this is not explicitly stated in the text.

**Strengths And Weaknesses:**

The paper proposes high-level recourse methods that are actionable and interpretable, and further introduces techniques to estimate the optimal recourse for each agent. More interestingly, it presents a data-driven method that learns a direct mapping from the features to the CFEs, which is generalizable across different agents. The proposed approach has been extensively evaluated on multiple real-world and synthetic datasets.

However, the requirements for training both the single-agent CFE generators and the data-driven CFE generator are stringent. The single-agent CFE generators depend on the existence of some accurate classifiers. In the simulation experiments, the authors first trained a classifier before training the single-agent generator. Since it is usually difficult to obtain a perfect classifier (in the examples of BMI and WHR in the paper, the test accuracies are 72.78% and 85.18%), the performance of the generator is likely to be compromised. Although the data-driven CFE generator is claimed not to rely on direct access to the classifier, it depends on the existence of the true optimal CFEs, which is a stronger assumption than that required by a classifier. In the simulation experiments, the true label is obtained through the single-agent generators. However, since these generators are not perfect, the data-driven generators are likely to be further compromised. This long chain (training a classifier, then a single-agent generator, and finally a data-driven generator) may be fragile. I wonder whether there exists a method that directly finds the actions to generate the desired outcome without the intermediary steps.

---

> ### Author Response · Authors · 2025-04-22
> **Response to reviewer(1)**
>
> Thank you, reviewer VKih for reviewing our work!  Below are the responses to the points the reviewer specifies under the “requested changes” subheading. Please refer to the revised manuscript for accurate references of sections used in the response below.  We have uploaded an edited manuscript with revisions (in blue) in response to the suggestion of the reviewers. To keep this response concise, we generally refrain from repeating the revised text in full here and instead indicate where we made changes.
>
> > Could the authors explicitly discuss the difference between low-level and high-level CFEs? Figure 1 was helpful in comparing them through concrete examples, but it is not immediately clear why estimating high-level CFEs is challenging and needs specific techniques. Technically, is this because multiple actions are selected simultaneously in a high-level CFE?
>
> Both the low-level and high-level CFE generators find the least-cost set of actions (not just one action) that favorably flip the prediction. \ Although generating the new proposed forms of CFEs presents new technical challenges because the actions are not feature-based as is the case in low-level CFEs, and associated costs are not computed from the classification training data (e.g., maximum percentile shift in Ustun et al. 2019), this was not the major motivation for the proposed new forms of recourse (high-level CFEs). We include the detailed motivation for the new proposed forms of recourse in the second and third-last paragraphs of the Background section.
>
> > According to Figure 1 and the following descriptions, the ultimate goal is to provide a descriptive recommendation using natural language. However, these recommendations still need to be represented as numerical values when learning the recourse…
>
> The hl-continuous and hl-discrete actions by design encode information on how they affect features, but the agents don't need to know these details to execute the CFE. Figure 1 and its caption, for example, says, "the hl-continuous CFE adjusts multiple features numerically, e.g., corresponding to features in the profile (app3), ca1 = [8.0, 61.5, 0.195, 0.2, 3.72, 2.0, · · · ], ca2 = [17.0, 0.0, 0.05, 0.0, 1.63, 23.0, · · · ] but the agent does not need to understand the exact changes to follow the CFE."  While the actions encode information on how they affect features which is important for the single agent CFE generation (Section 3), agents don't need to know this information to execute the CFEs. Additionally, the proposed data-driven CFE generators perform well in the case that this information is limited or restricted.
>
> > What are the range and dimensions of these actions? What is the form of the cost and effect, also ? … For the single-agent hl-continuous CFE generator in Section 3.1, what is the range of ?... …
>
> Definitions 1 and 2 are specifically meant to be general. Their dimensionality and range depend on the actionable features they affect. For instance, while the hl-discrete actions have binary effects (elaborate example with agent state given in definition 2), the hl-continuous actions like low-level CFEs operate over real-valued ranges. Equations (2) and (3) and their descriptions include details on formats of the actions and usage in the single-agent CFE generation.
>
> As noted in the text following Equations (2) and (3), the index $j\in J$ refers to the index of specific action in a list of actions. For example, with five possible actions, the indices would be {$\{1, 2, 3, 4, 5\}$}. The cost associated with each action, denoted $cost_j$, is a positive real number (e.g., 1, 0.1, 1.15, etc.).
>
> In Equation (3), we solve a weighted set cover problem. We find the least cost set of actions (problem objective) that fulfill the eligibility of all the required features (problem constraints). The formulation implicitly involves $v_j$, which governs the contribution of each action to the overall solution. The hl-discrete action ensures the feature $i$ meets the defined feature threshold $ t_i$ where if met, it is 1 and 0 otherwise.

---

> > ### Author Response · Authors · 2025-04-22
> > **Response to reviewer(2)**
> >
> > > A discussion of the differences between the problem settings for hl-continuous and hl-discrete actions should be provided. For example, if I understand correctly, the definition of the state , the action , and the form of the classifier are all different.
> >
> > The key distinction between hl-discrete and hl-continuous CFEs is how they affect features, where the former caters to feature eligibility and the latter numerically modifies feature values.
> > Using the example of feature healthBMI in App2, Figure 1, while the hl-discrete CFE would ensure BMI is past a desired threshold, thus ensuring healthBMI = 1, the hl-continuous CFE would affect the actual feature values, such as reducing BMI from 30 to 22.3.  These two forms of CFEs are suited for different contexts. The hl-discrete CFEs are particularly effective in rule-based systems, such as eligibility checks, wellness evaluations, or quality and safety assessments. In contrast, hl-continuous CFEs are better suited for scenarios where fine-grained numerical adjustments are meaningful, typically in environments where low-level CFEs are effective.
> >
> > We distinguish between hl-continuous and hl-discrete CFEs in Figure 1 and paragraph 3 of the introduction. We have also included a supplementary explanation in appendix A of the revised uploaded manuscript.
> >
> > > Why does equation (4) use a cross-entropy loss, while equation (5) is computed only over positive identifications? Additionally, in Section 4.2, why is the objective function defined as a mean-squared error, if—according to my understanding—this should still be treated as a classification problem?
> >
> > The data-driven CFE generators are implemented using supervised learning models, with different training strategies. The data-driven hl-continuous CFE generator generates the least-cost set of hl-continuous actions for a given agent. We trained the model using binary cross-entropy loss (Eq. 4), where the loss is summed over all actions for each agent, treating each action as an independent binary classification problem. This setup is effectively equivalent to multi-label classification. \
> > The choice of the loss function for the varied data-driven hl-discrete CFE generators depended on the model architectures and dataset. When each output bit represents an independent binary decision (i.e., the probability of being 1), we use binary cross-entropy. However, when the goal is to produce a binary vector that closely matches the target, especially in sparse cases, we use mean squared error (MSE) that captures the overall similarity between the predicted and target vectors.
> > The data-driven hl-id CFE generator outputs a single hl-id CFE for each agent, selected from a set of possible CFEs. This model is trained with categorical cross-entropy loss (Eq. 5), aligning with the multi-class classification framework, as each agent selects one CFE (hl-id CFE) from multiple candidates.
> >
> > > Could the authors explain why the coefficient of variation for the high-continuous CFE is much lower in Figure 3(b)? ….
> >
> > The coefficient of variation, a normalized measure of dispersion, is calculated as the ratio of the standard deviation to the mean of a given variable. Figure 13(c) shows the number of modified features across different sensitive groups for the two forms of recourse, offering insights into why the coefficient of variation for hl-continuous CFEs in Figure 3(b) (or Figure 13(d)) is substantially lower than that of low-level CFEs. Additionally, Section 6.1, Paragraph 3, and Appendix Figure 12 provide further context that explores the relationships between various variables and helps explain the patterns observed in Figure 3(b).
> > In the revised manuscript, we have edited the caption for Figure 3 to make it easier for readers to connect Figure 3(b) to Appendix Figure 13(c).
> >
> > > Section 6.1 discusses single-agent generators, but this is not explicitly stated in the text
> >
> > In Section 5.4, "Evaluation and Comparative Analysis Metrics," we clarify that the comparative analysis of the actual CFEs primarily focuses on CFEs generated by single-agent CFE models. While the first paragraph of Section 6.1 explicitly mainly focuses on comparing low-level CFEs to high-level CFEs, the second paragraph blends evaluating the differential pros and cons between the CFEs and highlighting the advantages provided by data-driven CFE generators.

---

> > > ### Author Response · Authors · 2025-04-22
> > > **Response to reviewer(3)**
> > >
> > > >  However, the requirements for training both the single-agent CFE generators and the data-driven CFE generator are stringent. The single-agent CFE generators depend on the existence of some accurate classifiers. In the simulation experiments, the authors first trained a classifier before training the single-agent generator. Since it is usually difficult to obtain a perfect classifier (in the examples of BMI and WHR in the paper, the test accuracies are 72.78% and 85.18%), the performance of the generator is likely to be compromised. Although the data-driven CFE generator is claimed not to rely on direct access to the classifier, it depends on the existence of the true optimal CFEs, which is a stronger assumption than that required by a classifier. In the simulation experiments, the true label is obtained through the single-agent generators. However, since these generators are not perfect, the data-driven generators are likely to be further compromised. This long chain (training a classifier, then a single-agent generator, and finally a data-driven generator) may be fragile. I wonder whether there exists a method that directly finds the actions to generate the desired outcome without the intermediary steps.
> > >
> > > CFE generators, whether single-agent, global, or data-driven, can be explicitly or implicitly compromised by changes in the classifier or by relying on a less-than-accurate classifier. Feature-based low-level CFEs, such as those proposed by Ustun et al. (2019), are particularly very vulnerable because they are so specific that even minor changes in the classifier may render them invalid. Data-driven CFE generators, in particular, may be susceptible to model drift due to their reliance on agent-CFE datasets. These datasets are often shaped, either implicitly or explicitly, by the aggregation approach and the dependence of the aggregation method on the underlying classification model.
> > >
> > > To address issues related to model errors and changes, robust CFE generation methods (e.g., Guo et al., 2023; Jiang et al., 2024) aim to account for model and distribution drift during the CFE generation process which improves the overall performance of all CFE generators, whether single-agent, global, or data-driven.
> > >
> > > Looking ahead, a promising direction for future research is to investigate the robustness of data-driven CFE generators in the presence of model drift. In particular, it would be valuable to examine whether the proposed high-level CFEs offer greater resilience to changes or errors in the classification model compared to other forms of recourse, such as low-level CFEs. Additionally, future work could explore methods for the valuation of agent-CFE data instances and developing robust aggregation strategies (e.g., federated learning) to enhance the overall performance of data-driven CFE generators.
> > >
> > > - [Guo et al. 2023] "Rocoursenet: Robust training of a prediction-aware recourse model," in In Proceedings of the 32nd ACM International Conference on Information and Knowledge Management, CIKM ’23 page 619–62
> > > - [Jiang et al. 2024] "Robust counterfactual explanations in machine learning: a survey," In Proceedings of the Thirty-Third International Joint Conference on Artificial Intelligence, IJCAI ’24 2024

---

> > > > ### Author Response · Authors · 2025-05-09
> > > >
> > > > Hello Reviewer VKih, thank you for reviewing our work!
> > > > Since the author-reviewer discussion period is ending soon, we wanted to check if we have sufficiently addressed all your concerns or if any remain.

---

> > > > > ### Comment · Reviewer_Vkih · 2025-05-12
> > > > >
> > > > > Thank you for the thoughtful responses. I appreciate the clarifications and would like to see them incorporated into the revised manuscript. Additionally, please clarify in the abstract and introduction that the “predefined real-world–like actions” refer to a pre-specified set of high-level actions, each of which influences multiple features, with the effect on each feature assumed to be known.

---

> ### Author Response · Authors · 2025-05-13
> **Response to reviewer VKih**
>
> Thank you, Reviewer VKih! We incorporated all the suggested changes for previously raised concerns in the uploaded revised manuscript. For example, see page 26 on "Robust data-driven CFE generators" for the comment on classifier errors.
>
> On clarifying that the "predefined real-world–like actions" refer to a pre-specified set of high-level actions, each of which influences multiple features, with the effect on each feature assumed to be known.
> We have now edited the third paragraph of the introduction to include the suggestion. The change is as follows:  "Specifically, the hl-continuous CFEs involve general and predefined real-world-like actions that modify multiple features simultaneously through numerical adjustments. Similarly, hl-discrete CFEs also stem from real-world-like actions, *each of which might affect several features, with the effect on each feature assumed to be known*. However, unlike hl-continuous CFEs, which adjust feature values, hl-discrete CFEs modify feature eligibility."
>
>
> To avoid making the abstract longer, we hope that the change in the introduction sufficiently address reviewer VKih's concerns.
> Please let us know if we have sufficiently addressed all your concerns or if any remain.

---

> > ### Comment · Reviewer_Vkih · 2025-05-13
> >
> > Thanks for addressing my concerns!

---

### Review · Reviewer_rVLP · 2025-04-18

**Summary Of Contributions:**

This paper considers algorithmic recourse, where the goal is provide users with actionable insights on how to change their features in order to change a model's prediction (e.g. flipping negative to positive). These are known as "counterfactual explanations" or CFE. Previous work focused on low-level CFE requiring precise adjustments to feature values which can be impractical. In contrast, this work considers high-level CFE corresponding to real-world implementable actions.

The paper proposes three types of high-level CFEs: hl-continuous, hl-discrete, and hl-id. It then describes how to obtain the CFEs for a single example (or "agent" in the paper's phrasing) by formulating integer programs for hl-continuous and hl-discrete. Lastly, it describes a data-driven approach to learn a model that maps an agent's current feature to the optimal CFE.

Experiments were conducted on health-related datasets including BMI (body mass index) and WHR (waist-to-hip ratio), and using Foods dataset to define the food items as the real-world implementable actions.

**Audience:**

Yes

**Claims And Evidence:**

No

**Requested Changes:**

Major issues
- The title is misleading. The phrase "large state spaces" occurs nowhere else in the paper. "State space" has a very specific meaning in machine learning and reinforcement learning, and the authors are encouraged to reconsider the wording.
- In Sec 3 when introducing "Single-agent CFE Generators", it is unclear where the action space comes from. Later in experiments it seems to be derived from some auxiliary dataset, but this may need to be explained earlier in the paper.
- Sec 3 only introduces "Single-agent hl-continuous CFE Generation" and "Single-agent hl-discrete CFE Generation". What about hl-id?
- Eqn (2)(3) are written in a different format compared to Eqn (1) which makes it confusing to interpret.
  - They also use new notation that are not introduced in the definitions of hl-continuous action and hl-discrete action (Def 1 and 2).
  - What are the optimization variables of these optimization problem? This is not specified in the equation.
- Eqn (2) assumes a linear classifier, whereas Eqn (3) assumes a threshold classifier. However, before Sec 3, the framing of the paper is addressing algorithmic recourse in binary classification (Sec 2) without placing assumption on the model. This discrepancy means the proposed approach does not fully address the problem setting considered.
- Def 3 hl-discrete action only allows changing a binary feature from 0 to 1, but does not discuss the opposite, nor does it discuss categorical features (the paper only says "the formulation is extensible to more general case" without explaining which cases and how.
- Experiments:
  - In CFE generation the minimize problem only returns one optimal, but in reality there might be multiple actions that all achieve the desired outcome (of flipping predictions). The current evaluation on accuracy does not seem to reflect this.
  - In Sec 6.1 & Fig 3(a), there is no comparison on the cost achieved by low-level vs high-level CFE, it only compares the number of actions, but each low-level action and high-level action costs are different so they are not directly comparable.
  - Fig 3(b) compares across sensitive groups but did not specify what sensitive groups were used.
  - Sec 6.2 & Table 1 shows results for the data-driven CFE, but does not show any data for it being "resource-efficient".
  - Comparing single agent low-level CFE to data-driven high-level CFE is unfair. Why can't we make data-driven low-level CFE?
  - The paper also makes claims about the proposed approach being "more transparent and easier to interpret" "making them cheaper and more desirable" (Page 11) but this is only based on the three examples shown in Fig 1. Such conclusions would require user study to justify.

Minor issues
- Incorrect citation style on page 1: "A popular form of recourse, actionable recourse Ustun et al. (2019), ..."
- Page 2: "Finally, we conduct extensive experiments on 11+ agent–CFE datasets" - What does 11+ datasets mean? There should be an exact number of datasets used in the experiments.

**Strengths And Weaknesses:**

Strengths:
- Addresses an interesting problem with potential practical importance.

Weaknesses:
- Proposed methods appear to be limited to certain problems e.g., model must be a binary linear classifier, requires prior knowledge of the set of possible high-level actions.
- Notation are not defined clearly and appear to overcomplicate the ideas.
- Experiments do not fully support the claims e.g., on resource-efficiency.

---

> ### Author Response · Authors · 2025-04-22
> **Response to reviewer(1)**
>
> Thank you, reviewer rVLP for reviewing our work!  Below are the responses to the points the reviewer specifies under the “requested changes” subheading. Please refer to the revised manuscript for accurate references of sections used in the response below.  We have uploaded an edited manuscript with revisions (in blue) in response to the suggestion of the reviewers. To keep this response concise, we generally refrain from repeating the revised text in full here and instead indicate where we made changes.
>
> >The title is misleading. The phrase "large state spaces" occurs nowhere else in the paper. "State space" has a very specific meaning in machine learning and reinforcement learning, and the authors are encouraged to reconsider the wording.
>
> Interestingly, this problem can be viewed as a Markov Decision Process, where a policy (i.e., CFE generator) navigates the state space to propose actionable changes for a negatively classified agent. Starting from an initial state $x$, defined by a feature vector including attributes such as age and calcium (mg), the goal of the CFE generator is to find the least cost set of actions that the agent can take to reach a new state $x'$ where the classifier yields a favorable prediction. \
> More generally, we view the state space as the set of all possible configurations of agent states, that is, the set of all possible feature instantiations as we use “state” and “features” interchangeably, which we believe to be consistent with the use of the term “state space” both within machine learning and beyond. We have clarified this definition in the revised manuscript’s background section. That said, we would be happy to consider an alternative phrasing.
>
> >In Sec 3 when introducing "Single-agent CFE Generators", it is unclear where the action space comes from. Later in experiments it seems to be derived from some auxiliary dataset, but this may need to be explained earlier in the paper.
>
> The action space is determined by the specific application domain. Sections 3 and 4 provide domain-agnostic definitions and methodologies for the proposed single-agent and data-driven CFE generators. Sections 6 and 7 demonstrate the application of the proposed CFE generation approaches to particular domains, where the nature of the action space, such as the hl-continuous and hl-discrete actions space described in Section 5.1, is tailored to the domain's requirements. These sections (6 and 7) serve as proof-of-concept, illustrating both the adoption and effectiveness of the proposed CFE  generation approaches.
>
> >Sec 3 only introduces "Single-agent hl-continuous CFE Generation" and "Single-agent hl-discrete CFE Generation". What about hl-id?
>
> The hl-id CFEs provide a more abstract representation than the hl-continuous and hl-discrete CFEs, as illustrated in Figure 1 and discussed in the introduction. Unlike hl-continuous and hl-discrete CFEs, which include explicit references to actions, their associated costs, and their direct effects on features, hl-id CFEs omit this level of detail. Instead, their generation is primarily based on agent profile similarity. This marks a shift toward a data-driven methodology, moving away from action-specific reasoning. Accordingly, in Section 4, we detail how hl-id CFEs are generated using data-driven techniques that leverage similarities across agents.
>
> >Eqn (2)(3) are written in a different format compared to Eqn (1) which makes it confusing to interpret. They also use new notations that are not introduced ... What are the optimization variables of these optimization problems?...
>
> In equations. (1), (2), and (3), our goal is to identify the least-cost set of actions that favorably flip the prediction, however, equations. (2) and  (3) each have distinct sets of constraints and variables compared to (1). For instance, equation(3) represents a solution to a weighted set cover problem. The notation for each of the equations. (1), (2), and (3) are defined in the paragraphs following the equations, beginning with "where." In each equation, the expression to the right of "minimize" defines the problem objective of the problem, while the expressions following "s.t." outline the problem constraints.

---

> ### Author Response · Authors · 2025-04-22
> **Response to reviewer(2)**
>
> >Eqn (2) assumes a linear classifier, whereas Eqn (3) assumes a threshold classifier. However, before Sec 3, the framing of the paper is addressing algorithmic recourse in binary classification (Sec 2) without placing assumptions on the model. This discrepancy means the proposed approach does not fully address the problem setting considered.
>
> Both classifier types are binary, assigning agents to either a positive or negative class.  We define the threshold classifier in the second sentence of the paragraph preceding Section 4 where we state: “The threshold classifier $\mathbf{t} = \{t_{1}, t_{2},\cdots, t_{n} \}$ over \(n\) features classifies an agent state $\mathbf{x}$  positive if $x_{i} \geq t_{i}, \ \forall i \in [n]$, and negative otherwise.” Consequently, the proposed CFE generation approaches generate recourse for the negatively classified agents, and the approach is thus well-aligned with the problem setting considered.
>
> >Def 3 hl-discrete action only allows changing a binary feature from 0 to 1, but does not discuss the opposite, nor does it discuss categorical features (the paper only says "the formulation is extensible to more general case" without explaining which cases and how.
>
> Figure 1 (App2) illustrates an example of the hl-discrete action and its effect on features, while Definition 2 formally defines the action. This paper focuses on scenarios where hl-discrete actions ensure feature eligibility, constrained to binary values $\{0,1\}$. However, the proposed formulation is extensible to more general settings (e.g., categorical), which would require modifying the problem constraints and where Mixed Integer Programming (MIP) may offer a more suitable approach to solving the problem.
>
> >In CFE generation the minimize problem only returns one optimal, but in reality there might be multiple actions that all achieve the desired outcome (of flipping predictions). The current evaluation on accuracy does not seem to reflect this.
> The proposed CFE generators, both the single-agent CFE generators and the data-driven CFE generators generate  CFEs, which are a set of the least cost actions that favorably flip the prediction. These CFEs have a varied number of actions in them.
>
> Therefore, we don’t return one action but rather a CFE with a set of the least cost actions that flip the prediction.
> In the data-driven CFE generation, the evaluation formulation (Eq. 6) assesses the correctness of the generated CFEs. A generated CFE is correct if it matches the true CFE, defined as the one that incurs the least cost while favorably flipping the model’s prediction.
>
> >In Sec 6.1 & Fig 3(a), there is no comparison on the cost achieved by low-level vs high-level CFE, it only compares the number of actions, but each low-level action and high-level action costs are different so they are not directly comparable.
>
> No, we do not compare the specific actions taken but rather the number of actions, which provides a consistent and comparable metric across the two CFEs. In contrast, the cost metrics used for the two CFEs are not directly comparable. For the low-level CFEs, Ustun et al. (2019) define cost as the maximum percentile shift (Equation 3 in Ustun et al. (2019)).
> In contrast, the costs of the proposed actions, as outlined in Section 5.1, are expressed in terms of monetary or caloric costs.
>
> >Fig 3(b) compares across sensitive groups but did not specify what sensitive groups were used.
>
> The sensitive groups are race, age, and gender. We consider the intersectional sensitive groups (e.g., Hispanic, 21-40, Female). We have also revised the caption on Figure (3b) and Figure 3 and the text following it to make it more explicit. Thank you for pointing this out.
>
> >Sec 6.2 & Table 1 shows results for the data-driven CFE, but does not show any data for it being "resource-efficient".
>
> Although we don't explicitly use the term "resource efficient," the ideas of strong performance under diverse informational challenges and eliminating the need for re-optimization inherently reflect resource efficiency. The first paragraph of section 6.2 mentions that "our results demonstrate that the proposed data-driven CFE generators, operating under various information... and accurately and efficiently produce CFEs, without requiring re-optimization of the generator ..."

---

> ### Author Response · Authors · 2025-04-22
> **Response to reviewer(3)**
>
> >Comparing single agent low-level CFE to data-driven high-level CFE is unfair. Why can't we make data-driven low-level CFE?
>
> A key characteristic of data-driven CFE generation is its reliance on the similarity between agent profiles to identify suitable CFEs. As noted in the background section, specifically in the sentence beginning with "Second, although a CFE..." low-level CFEs tend to be highly specific. Furthermore, as demonstrated in Section 6.1 and Figure 12 in the Appendix, unlike high-level CFEs, low-level CFEs are generally unique to each individual. We also observe a statistically significant perfect positive correlation between the number of modified features and the actions taken. Given these observations, we opted not to use data-driven low-level CFEs. Additionally, we compare the data-driven CFE generation to single-agent CFE generation, and in the background section, particularly the last paragraph, we motivate the proposed CFE generation approach.
>
> >The paper also makes claims about the proposed approach being "more transparent and easier to interpret" "making them cheaper and more desirable" (Page 11) but this is only based on the three examples shown in Fig 1. Such conclusions would require user study to justify.
>
> The high-level CFEs (hl-continuous and hl-discrete) have general, predefined actions and costs that closely resemble real-world ones, making them more interpretable and transparent than the overly specific, feature-based low-level CFEs.
> Specifically, high-level CFEs offer actionable insights that are more directly executable, with costs that better approximate real-world values. As a result, agents would expend less effort interpreting and translating recommendations into action and can more easily compare CFEs.
>
> In contrast, low-level CFEs require additional costs and effort to compare and operationalize, as the feature-based changes often lack clear real-world analogs and cost clarity. As Barocas et al. observes, a key hidden assumption in many CFEs is that "features do not clearly map to actions." Similarly, Ustun et al. (2019) highlight that the cost estimates associated with low-level CFEs "may not correctly reflect the difficulty of recourse."
> Looking ahead, in line with the reviewer's suggestion, conducting a comprehensive user study would be a valuable avenue for future work. The human-subject study could empirically validate the transparency and interpretability of the proposed high-level CFEs, serving as a test of the underlying hypotheses.
>
> >Incorrect citation style on page 1: "A popular form of recourse, actionable recourse Ustun et al. (2019), ..."
>
> Thank you for pointing out the error, we have corrected it in the revised uploaded manuscript.
>
> >"Finally, we conduct extensive experiments on 11+ agent–CFE datasets" - What does 11+ datasets mean? There should be an exact number of datasets used in the experiments.
>
> Thank you, In the revised manuscript that has been uploaded, we have specified the exact number of semi-synthetic and fully synthetic datasets, 30 and 34, respectively.
>
> ----
> If you come across any claims that appear inconsistent with the evidence, please do not hesitate to let us know.

---

> ### Comment · Reviewer_rVLP · 2025-05-04
>
> > "state space" terminology
>
> OK. With the revised manuscript, this makes more sense.
>
> > where to get action space
>
> OK.
>
> > hl-id CFE in Sec 3
>
> I'm still not convinced by the explanation. It seems that the difference between hl-discrete/hl-continuous vs hl-id is that in hl-id we can have mixed feature types (categorical and continuous) that the high-level action can modify. Is it impossible to write an optimization problem for hl-id in the same way that hl-discrete/hl-continuous, and if so, why? This should be explained at the beginning of Sec 3. If that's not the case, then I believe this formulation should be included, otherwise the paper is not addressing the task it set out to address completely.
>
> > Eqn (2)(3)
>
> The expressions still do not clearly show the optimization variables (in the defined notation) of each optimization problem. For Eqn (2), are the optimization variables only $a_j$, or both $a_j$ and $\epsilon_j$?
>
> > linear/threshold classifier
>
> I agree both are binary classifiers. However, they are placing assumptions on the type of models. For example, the proposed approach cannot be applied to non-linear classifier (or does it?), and this limits the scope of the contributions. Furthermore, this is not clearly stated in the abstract or introduction.
>
> > Def 3
>
> OK.
>
> > CFE accuracy evaluation
>
> OK.
>
> > Comparing number of actions
>
> Can you elaborate on (1) Why is the "number" of actions the right metric, regardless of the "size" of the actions? (2) Why can't there be a cost metric that can be applied to both low-level and high-level CFEs?
>
> > sensitive groups
>
> OK. Please make sure to include this information in the revised manuscript.
>
> > Resource-efficient
>
> The phrase "resource-efficient" appears in Sec 6.2 heading so it is explicitly used in the paper. Thus, I believe this is a claim that should be supported with data.
>
> > [remaining questions]
>
> All good.

---

> ### Author Response · Authors · 2025-05-05
> **Response to reviewer(1)**
>
> Thank you Reviewer rVLP18!
> Below are the responses to the reviewer's concerns.
>
> > I'm still not convinced by the explanation. It seems that the difference between hl-discrete/hl-continuous vs hl-id is that in hl-id we can have mixed feature types (categorical and continuous) that the high-level action can modify. Is it impossible to write an optimization problem for hl-id in the same way that hl-discrete/hl-continuous, and if so, why? This should be explained at the beginning of Sec 3. If that's not the case, then I believe this formulation should be included, otherwise the paper is not addressing the task it set out to address completely.
>
> No, the difference between hl-discrete/hl-continuous CFEs and hl-id CFEs **is not**  "that hl-id we can have mixed feature types (categorical and continuous) that the high-level action can modify."
>
> The actual difference lies in the level of abstraction and the specificity of actions. The hl-discrete and hl-continuous CFEs include explicit references to actions, associated costs, and their direct effects on features. In contrast, hl-id CFEs are more abstract: they do not specify constituent (sub)actions or how those actions directly impact features. In the introduction and Figure 1, we highlight this difference.
>
> In the single-agent CFE generation approaches (Eqs 1, 2, 3), based on the optimization frameworks, we must have access to a well-defined action set that defines the actions, their costs, and their effects on features. In contrast, generating hl-id CFEs relies primarily on agent profile similarity and domain knowledge and does not require explicit access to actions, costs, or their direct feature effects.
>
> In the revised manuscript, we added a clarifying sentence at the beginning of Section 4.3 to make this more explicit.
>
>
> > The expressions still do not clearly show the optimization variables (in the defined notation) of each optimization problem. For Eqn (2), are the optimization variables only $a_j$, or both $a_j$ and $\epsilon_j$?
>
> In Eq2, $a_j \in $ {0, 1} is a binary decision variable indicating whether action $ j \in J$ is selected ($1$) or not ($0$). The binary $\epsilon_j \in \in $ {0, 1} controls the sign or direction of influence for action. Both $a_j$ and $\epsilon_j$ are optimization variables.
> The goal of the objective function, $minimize \sum_{j \in J} cost_{j} a_{j}$,  is to minimize the total cost of selected actions. Each action has a predefined cost $\text{cost}_j$, which only counts if the jth action is selected, that is, $a_j = 1$.
> The first line of constraint ensures that the generated CFE, which is a set of least-cost actions in the solution set, favorably flips the prediction, and the second one ensures that every action is in at least one selected set.
>
> Equation 3 is a typical weighted set cover problem formulation. See, for example, these lecture notes (https://www.cs.cmu.edu/~15451-f16/lectures/lec21-approx2.pdf)
>
> At the end of each formulation (Equations 1, 2 and 3),  with the sentence that begins with "where",  we specify what variables are predefined and which are not.
>
> > I agree both are binary classifiers. However, they are placing assumptions on the type of models. For example, the proposed approach cannot be applied to non-linear classifier (or does it?), and this limits the scope of the contributions. Furthermore, this is not clearly stated in the abstract or introduction.
>
> Linear classifiers are primarily employed in Equations (1) and (2) for generating low-level CFEs and single-agent hl-continuous CFEs, respectively. In contrast, single-agent hl-discrete CFE generation relies on feature thresholds, which can be part of varied classifier types.
>
> On the other hand, the data-driven CFE generators introduced in Section 4 leverage neural networks with diverse architectures. These generators are agnostic to the underlying classifiers and rely mainly on agent-CFE datasets.
>
> Therefore, aside from classification tasks involving the BMI and WHR datasets and the single-agent low-level and hl-continuous CFE generation, a majority of our work utilizes a broader range of model types.
>
> Lastly, for the few cases where we employ linear classifiers, we would like to note that although the linear models may not always be optimal, they can outperform non-linear models in some contexts. Additionally, in the case that non-linear classifiers are preferred, one practical alternative is to approximate them by linear models, enabling CFE generation with the proposed single-agent hl-continuous CFE generator.

---

> ### Author Response · Authors · 2025-05-05
> **Response to reviewer(2)**
>
> > Can you elaborate on (1) Why is the "number" of actions the right metric, regardless of the "size" of the actions? (2) Why can't there be a cost metric that can be applied to both low-level and high-level CFEs?
>
> The metric “number of actions” offers a consistent and comparable measure across both hl-discrete/hl-continuous and low-level CFEs. Although these two types of CFEs differ conceptually, i.e., while low-level CFEs involve feature-based action, the hl-discrete/hl-continuous CFEs involve real-world-like actions, the two CFE types are still comparable by the number of actions taken within each CFE. The focus is on how many actions does each CFE have? How does this compare with other CFEs?
>
> However, a direct comparison of costs incurred by hl-discrete/hl-continuous and low-level CFEs is not straightforward. The cost of low-level CFEs is defined in terms of the maximum percentile shift across features (as in Equation 3 of Ustun et al., 2019), whereas we use predefined real-world costs (e.g., monetary or caloric values). These cost measures are inherently different and thus not directly comparable.
>
> While the number of actions provides a shared measurement scale, aligning the cost metrics would require converting both cost types into a common framework or uniform scale for direct comparison. We are uncertain how feasible this is at the moment.
>
> > OK. Please make sure to include this information in the revised manuscript.
>
> We included it in the revised manuscript, thank you!
>
> > The phrase "resource-efficient" appears in Sec 6.2 heading so it is explicitly used in the paper. Thus, I believe this is a claim that should be supported with data.
>
> We have results (data) on resource efficiency. Specifically, our results show that the proposed data-driven CFE generators are resource-efficient in terms of both limited information access and low computational overhead. These generators, without requiring re-optimization (low computational overhead), accurately and efficiently generate CFEs for new agents after being trained under restrictive constraints such as no query access to the classifier or, in the case of the data-driven hl-id CFE generator, without knowledge of the cost and impact of actions on agent states (efficient when resources (information) are restricted).
>
> We have revised the first sentence of Section 6.2 to more explicitly include the notion of resource efficiency in relation to information, computational overload and accuracy.
>
> ----
> *Let us know if the responses address your concerns and if you come across any claims that appear inconsistent with the evidence, please do not hesitate to let us know.*

---

> > ### Author Response · Authors · 2025-05-09
> >
> > Hello Reviewer rVLP, thank you for reviewing our work and the correspondence!
> > Since the author-reviewer discussion period is ending soon, we wanted to check if we have sufficiently addressed all your concerns or if any remain.

---

> > > ### Comment · Reviewer_rVLP · 2025-05-16
> > >
> > > Thanks for the updates. All of my major concerns have been addressed.

---

### Comment · Action_Editor_Y3oG · 2025-05-03
**Please respond to author responses**

Hello reviewers,

We're now entering the phase of this paper's submission to where you will be prompted for your final recommendation.

I appreciate your efforts in reviewing this paper but we are not done yet. I will not acknowledge any reviewer's recommendation who has not responded to the authors' follow-ups. We owe them this as it is a shared expectation for any of our own work. Please, at your earliest convenience, discuss with the authors whether their clarifications have addressed your concerns or if there are still outstanding issues you have with their work.

Thanks,
AE

---

### Decision · Action_Editor_Y3oG · 2025-05-25

**Recommendation:** Accept as is

**Comment:**

The reviewers collectively agree, after a rich discussion with the authors that this paper is adequately prepared for publication. The authors were very engaged in the process and submitted timely revisions of the paper in addressing reviewer comments. As such, it is felt that this paper is ready for publication as is. There are no outstanding major issues that the authors need to address before the camera ready deadline.

I would recommend that the authors do take a final fine-grained pass through the reviews and their agreed upon changes to the paper to be sure that they've included everything.

**Audience:**

The paper addresses the issue of actionable recourse, which the reviewers collectively felt will be valued by the XAI community. There are some limitations due to the specific assumptions to develop the proposed approach, however this will be a nice stepping stone for the community to build upon in future work.

**Claims And Evidence:**

This paper considers algorithmic recourse, where the goal is provide users with actionable insights on how to change their features in order to change a model's prediction (e.g. flipping negative to positive). These are known as "counterfactual explanations" or CFE.

The paper proposes three types of high-level CFEs: hl-continuous, hl-discrete, and hl-id. The paper then introduces a data-driven approach to learn a model that maps an agent's current feature to the optimal CFE.

Experiments were conducted on health-related datasets including BMI (body mass index) and WHR (waist-to-hip ratio), and using Foods dataset to define the food items as the real-world implementable actions which extensively support the claims made about the development of the proposed CFE approach. The results demonstrate that the single-agent CFE generators require few actions while yielding greater improvements compared to low-level CFEs, and that the data-driven generators accurately replicate the optimal recourses computed by the single-agent generator while maintaining computational efficiency.